# Mesolimbic dopamine adapts the rate of learning from action

Luke T. Coddington[1 ✉], Sarah E. Lindo[1] & Joshua T. Dudman[1 ✉]

Recent success in training artificial agents and robots derives from a combination of direct learning of behavioural policies and indirect learning through value functions[1–3]. Policy learning and value learning use distinct algorithms that optimize behavioural performance and reward prediction, respectively. In animals, behavioural learning and the role of mesolimbic dopamine signalling have been extensively evaluated with respect to reward prediction[4]; however, so far there has been little consideration of how direct policy learning might inform our understanding[5]. Here we used a comprehensive dataset of orofacial and body movements to understand how behavioural policies evolved as naive, head-restrained mice learned a trace conditioning paradigm. Individual differences in initial dopaminergic reward responses correlated with the emergence of learned behavioural policy, but not the emergence of putative value encoding for a predictive cue. Likewise, physiologically calibrated manipulations of mesolimbic dopamine produced several effects inconsistent with value learning but predicted by a neural-network-based model that used dopamine signals to set an adaptive rate, not an error signal, for behavioural policy learning. This work provides strong evidence that phasic dopamine activity can regulate direct learning of behavioural policies, expanding the explanatory power of reinforcement learning models for animal learning[6].

Biological and artificial agents learn how to optimize behaviour through experience with an environment. Reinforcement learning theory describes the algorithms that allow an agent to iteratively improve its success through training[3]. Experience with the environment can be evaluated either by the success of an agent's behavioural 'policy' that directly determines the actions performed ('policy learning') or by an agent's subjective expectations of reward that indirectly guide action ('value learning'). Over the past several decades much work has explored how midbrain dopamine neuron (mDA) activity matches the predicted update signals (reward prediction errors (RPEs)[7]) for value learning[4]. However, mDA activity also reflects a heterogeneous mix of signals and functions that may not be completely addressed by the predictions of value learning models[8–12]. Phasic mDA activity can be intertwined with the production and monitoring of action[10,13–18] and is determined at least in part by inputs from areas involved in determining behavioural policy[19]. This calls for an exploration of how broadening the scope of considered reinforcement learning algorithms might inform our understanding of phasic mDA signals in biological agents.

Direct policy learning specifically offers untapped potential[5,20] to provide 'computational and mechanistic primitives'[6] that explain the functions of dopamine, especially in the context of novel task acquisition by animals. First, direct policy learning methods have achieved substantial success in embodied learning problems in robotics that resemble problems faced by a behaving animal[1]. Second, under a wide set of conditions policy learning is the most parsimonious reinforcement learning model that explains learned behaviour[5]. Third, policy learning can be directly driven by behavioural performance error (PE) signals, in lieu of, or in addition to, RPEs[21,22], connecting them to diverse observations of learning in dopamine-recipient brain areas[23,24]. Finally, policy learning methods facilitate explicit modelling of meaningful variability[25] in individual behavioural learning trajectories as a search through the space of policy parameterizations[1].

It can in fact be a criticism of policy search that learning trajectories can be too variable; although conducive to modelling individual differences, this feature can produce suboptimal learning[26,27]. A powerful solution is to set an optimal update size for each trial according to some heuristic for how useful each trial could be for learning[2]. Doing so independently of the performance feedback that directs learning can enhance useful variability while suppressing noise[1,26,28]. Such 'adaptive learning rates' have led to fundamental advances in machine learning[28], and can also make models of animal learning more accurate[29]. Thus, insights from policy learning lead to an intriguing hypothesis for phasic mDA activity that has not, so far, been explored. Phasic mDA activity could be a useful adaptive learning rate signal, given its correlations to novel and salient stimuli[12], upcoming actions[13] and prediction errors[7], all of which are useful heuristics for identifying key moments during which learning rates should be elevated. Alternatively, mDA activity correlates with PEs during avian song learning[30], suggesting that in mammals it could also dictate error-based updates to behavioural policies—a role more analogous to conveying RPEs for value learning. The establishment of policy learning models of canonical animal behavioural tasks is required to distinguish among these possibilities.

Here we develop a policy learning account of the acquisition of classical trace conditioning in which behaviour is optimized to minimize the

[1]Howard Hughes Medical Institute, Janelia Research Campus, Ashburn, VA, USA. ✉e-mail: coddingtonl@hhmi.org; dudmanj@janelia.hhmi.org

latency to collect reward once it is available, inspired by observations of this process in naive mice. A multidimensional dataset of behavioural changes during acquisition could be seen to drive improvements in reward collection performance, and a novel policy learning model quantitatively accounted for the diverse learned behaviour of individual animals. mDA activity predicted by the component of this model that sets an adaptive learning rate closely matched fibre photometry recordings of mDA activity made continuously throughout learning. Individual differences in initial phasic mDA responses predicted learning outcome hundreds of trials later in a manner consistent with dopamine modulating learning rate. Optogenetic manipulation of ventral tegmental area (VTA) dopamine neurons was calibrated to physiological signals and triggered in closed-loop with behaviour to provide a key test of the hypothesis that phasic mDA activity modulates learning rate as a distinct alternative to signalling signed errors. Together, these results define a novel function for mesolimbic dopamine in adapting the learning rate of direct policy learning (summarized in Extended Data Fig. 10).

## Task design and learning trajectories

We tracked multiple features of behavioural responses to classical trace conditioning in thirsty mice that had been acclimated to head fixation but had received no other 'shaping' or pre-training. Sweetened water reward was 'cued' by a 0.5-s auditory cue (10-kHz tone) followed by a 1-s delay, except on a small number of randomly interleaved 'uncued' probe trials (about 10% of total trials). Although reward was delivered irrespective of behaviour, mice still learned to optimize reward collection, as assayed by monotonic decreases in latency to collect reward across training (Fig. 1a–c). We measured multiple features of behaviour to understand how idiosyncratic learning across individual mice subserved performance improvements: an accelerometer attached to the movable basket under the mice summarized body movements[9], while high-resolution video was used to infer lick rate, whisking state, pupil diameter and nose motion. We reasoned that reward collection performance could be improved along two dimensions: preparation for reward delivery and reaction to its sensory components (Fig. 1d–f). 'Preparatory' behaviour was assayed across lick, body, whisker and pupil measurements as the total amount of activity during the delay period between cue and reward. 'Reactive' behaviour was assayed across nose, body and whisker measurements as the latency to initiate following reward delivery.

Although preparatory and reactive components of learned behaviour exhibited roughly monotonic trajectories on average (Fig. 1c), this belied heterogeneity in the dynamics of learning across individuals (Extended Data Fig. 1a–c). To assess the relationship between learned behaviours and reward collection performance on an individual basis, we built generalized linear models (GLMs) to predict reward collection latency across training in each mouse (Fig. 1d). GLMs using preparatory and reactive behavioural measures as predictors captured much of the variance in reward collection efficiency over training ($r^2 = 0.69 \pm 0.11$; $r^2$ with shuffled responses = $0.01 \pm 3 \times 10^{-4}$). Each predictor's weighting could vary widely from mouse to mouse, with preparatory licking having the most consistent relation to reward collection latency (Fig. 1d). However, both preparatory and reactive variables were necessary to most accurately predict reward collection latency (Fig. 1d; Friedman's: $P = 0.0003$; preparatory alone $r^2 = 0.51 \pm 0.24$, versus full model $P = 0.004$; reactive alone $r^2 = 0.46 \pm 0.20$, versus full model $P = 0.002$). Consistent with direct policy updates by a PE related to reward collection latency, our observations showed that updates to both the reactive and preparatory behaviour on each current trial were significantly related to reward collection latency on the previous trial (Fig. 1e). The significantly different time courses of preparatory and reactive learning (Fig. 1f) further confirm that these two learning components are dissociable processes.

We thus describe updates to the behavioural policy for each mouse as a trajectory through an abstract 'learning space' spanned by two components (preparation and reaction) that together explain improvements in reward collection performance optimized by minimizing reaction times and maximizing preparation (Fig. 1i and Extended Data Fig. 1b,c).

## ACTR policy learning model

The above data suggest that naive acquisition of trace conditioning could be considered as a problem of optimizing an effective control policy for reward collection through direct policy learning rather than indirectly through value learning. To formalize this comparison, we took an exemplar of a low-parameter value learning model that accounts for variable learning rates across individuals[31] and implemented a matched direct policy learning algorithm of the REINFORCE class[21] with equal free parameters (Methods). We next compared the negative log likelihood (−LL) and Akaike information criterion for the data given the optimal parameterization of each model class as carried out previously[32]. We found that the policy learning variant achieved significantly better fits (lower −LL and Akaike information criterion) when comparing optimally parameterized versions across nine mice ($\Delta LL = -252.6 \pm 70.8$; $P < 0.01$, signrank; Extended Data Fig. 1d,e). We also found that the policy learning formulation was markedly less brittle ($\Delta LL_{median} = -3.8 \times 10^3 \pm 0.8 \times 10^3$; $P < 0.01$, signrank; Extended Data Fig. 1f).

This model comparison indicates that for these broad algorithmic classes, a policy learning instantiation is a better descriptor of learning behaviour, as observed previously in human dynamic foraging[32] and sensorimotor adaptation tasks[33]. However, these low-parameter models enable only limited comparisons to behavioural and neurophysiological measurements. First, our experimental data clearly indicated two dissociable components of learning (reactive and preparatory) that have no clear analogy to abstract policy or value learning models in large part because there is no explicit modelling of the control of behaviour. Second, although model comparison reveals that policy learning algorithms in general may be superior, this is a broad class of algorithm that prescribes properties of the learning rule, but depending on model structure can be computed in many different ways[21]. Thus, we next sought to implement a circuit-inspired policy learning model that might facilitate direct comparisons to neurophysiological measurements.

To address these issues, we first specified a behavioural 'plant' (Extended Data Fig. 2d and Fig. 1h) that captured the statistics of rodent licking behaviour as a state model that transitions between quiescence and a licking state that emits a physiological lick frequency. A continuous control policy ($\pi(t)$) determined the forward transition rate to active licking. The reverse transition rate reflects a bias towards quiescence that decreases in the presence of water such that licking is sustained until collection is complete (Methods). The control policy was learned as the additive combination of output from a recurrent neural network (RNN) modelling preparatory learning and a feedforward sensorimotor pathway modelling reactive learning (Fig. 1h; see Methods for model details and code). Notably, optimal policies for speeding reward collection (identified by a search through the space of potential RNNs; Extended Data Fig. 2a,b) required preparatory cued licking that depends on sustained dynamics in the RNN output.

The PE used to train the model was proportional to the difference between performance, as measured by the latency to collect the water reward, and a correlate of expected performance, the activity of the output unit at the time of reward delivery (Methods). Both reactive and preparatory learning occurred in proportion to this PE, but they were implemented at different positions within the network. Reactive learning was modelled as changes to feedforward weights from

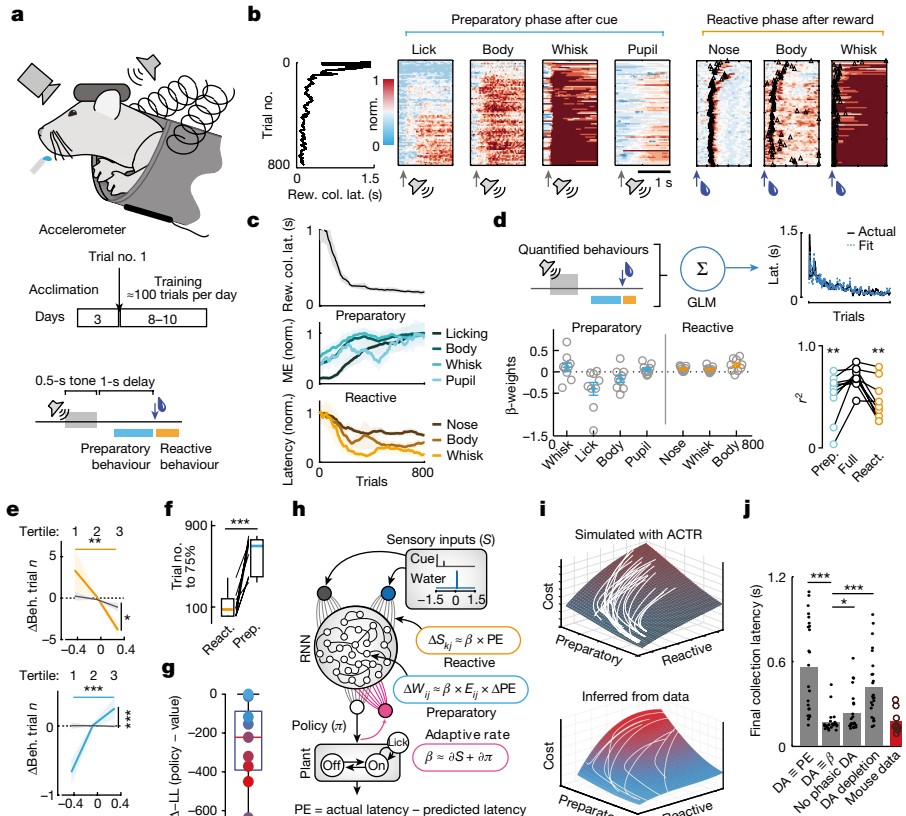

**Fig. 1 | Changes to behavioural policy correlate with improved reward collection performance. a**, Experimental design. **b**, Ten-trial binned behavioural quantification across the first 800 training trials. Reward collection latency (Rew. col. lat.; leftmost column) compared to normalized (Norm.) heat maps of preparatory (middle four columns; grey arrows: cue start) and reactive behaviour (right three columns; blue arrows: reward delivery; black triangles: mean first response). **c**, One-hundred-trial moving means of reward collection latency (top), and normalized preparatory (middle, motion energy (ME)) and reactive (bottom, latency) measures ($n = 9$ mice). **d**, Top: behavioural measures predicted reward collection latency in a GLM for each mouse. Bottom left: GLM predictor weights for each of nine mice. Bottom right: preparatory (Prep.; blue) or reactive (React.; orange) predictors alone performed worse than the full model. Significance testing: Friedman's. **e**, Trialwise reactive (top, orange) and preparatory (bottom, blue) behaviour (Beh.; binned into tertiles of PE magnitudes) correlated with inferred PE on the previous trial (black lines: shuffled control of trialwise PE for all other mice). $n$, trial number. Significance testing: two-way ANOVA; Tukey–Cramer post hoc. **f**, Trials to reach 75% maximum learned performance for reactive (orange) and

preparatory (blue) behaviours ($n = 9$ mice). $P < 0.001$; two-tailed rank sum test. **g**, Difference between fits (in negative log likelihood (−LL)) for versions of ACTR model (grey bars (smaller number equals better fit) for each mouse (coloured circles). **h**, The ACTR model learned a lick plant control policy ($\pi$) as the output from an RNN receiving sensory inputs following cue onset/offset (purple) and reward delivery (red). $e_{ij}$, eligibility trace for node perturbation at the synapse between the $i$th neuron and the $j$th neuron. Learning rules (blue and orange boxes) updated the weights of sensory inputs ($S_{kj}$) and internal connections ($W_{ij}$) using a mDA-like adaptive learning rate ($\beta$, pink). **i**, Top: cost surface calculated from ACTR model, overlaid with trajectories from individual initializations (white). Bottom: cost surface fitted from mouse data, overlaid with individual trajectories (white). **j**, Final performance for versions of ACTR model (grey bars; individuals as dots; $n = 24$) with the indicated differences in dopamine function (see main text) compared to observed performance in mice (red bar; individuals as circles; $n = 9$). Significance testing: rank sum. All error bars denote ±s.e.m. Box plots represent the median at their centre bounded by the 25th and 75th percentile of the data, with whiskers to each extreme. *$P < 0.05$; **$P < 0.01$; ***$P < 0.001$.

sensory inputs to behavioural policy output ($S_{kj}$; Fig. 1h), to replicate the optimization of behavioural responses to reward delivery in both cued and uncued trials (Extended Data Fig. 2e). Preparatory learning was modelled as changes to internal weights in the RNN ($W_{ij}$; Fig. 1h), and was proportional to the relative change in PE (customary in many policy learning algorithms[21]). The combination of reactive and preparatory learning robustly converged on stable, near-optimal policies that led to marked reductions in the latency to collect reward over several hundreds of training trials (Fig. 1i,j). To stabilize policy across a range of model initializations, an adaptive learning rate for each trial was controlled by a feedback unit (pink output unit in Fig. 1h); activity of the feedback unit was the sum of the state change in the behavioural plant (akin to an efference copy of reward-related action initiation commands) and the change in behavioural policy at the time of reward delivery (akin to reward-related sensory evidence informing behaviour[9]). This feedback scheme has a direct and intentional

parallel to the phasic activity of mDA neurons in this task, which is well described as the sum of action- and sensory-related components of reward prediction[9,13] and occurs in parallel to direct sensorimotor outputs[34]. Notably, this scheme closely reproduces mDA activity across naive learning as measured by both somatic spiking[9] and $Ca^{2+}$-sensor dynamics (Fig. 4). Overall, our approach adds an adaptive rate component inspired by supervised learning optimization methods[26,28,35] to an unsupervised, biologically plausible rule for training RNNs[36] that itself drew inspiration from node perturbation methods and the classic direct policy optimization method whose acronym is REINFORCE[3,21]. Hence, we refer to the complete model as ACTR (for adaptive rate, cost of performance to REINFORCE).

ACTR successfully reproduced many meaningful aspects of mouse behavioural learning data. For repeated ACTR simulations ($n = 24$) with a range of initializations, latency to collect rewards declined comparably to observed mouse behaviour over training (Fig. 1i–j, including an

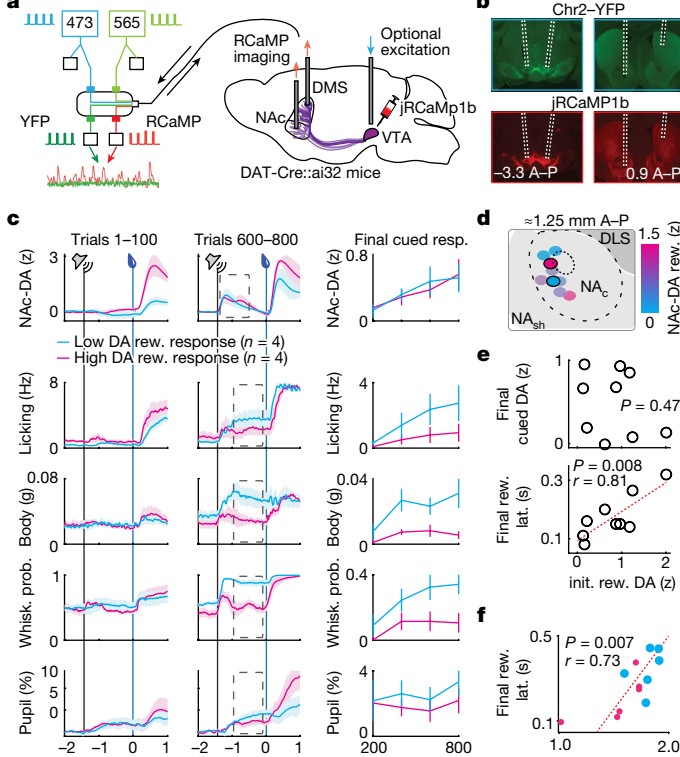

**Fig. 2 | Individual differences in mesolimbic dopamine signals correlate with learned behavioural policy. a**, Schematic for fibre photometry with optional simultaneous optogenetic stimulation of midbrain dopamine neurons. **b**, Fibre paths and virus expression from an example experiment. A–P, anterior–posterior. **c**, Left and middle: NAc–DA, licking, body movements, whisking probability and pupil diameter measurements for the mean of animals with lowest (blue, $n = 4$) and highest (pink, $n = 4$) NAc–DA reward signals over the initial 100 trials, shown for trials 1–100 (left) compared to trials 600–800 (middle). Right: means of responses (resp.) in the analysis windows indicated at left (dashed grey boxes) across training. **d**, Illustration of fibre locations for each mouse ($n = 9$), colour-coded according to the size of their initial NAc–DA reward signals. DLS, dorsolateral striatum; NA$_{sh}$, nucleus accumbens shell. **e**, Initial (Init.) NAc–DA reward responses (trials 1–100) were correlated with final latency to collect reward (bottom, Pearson's $r = 0.81$, $P = 0.008$), but not with final cued NAc–DA response (top, Pearson's $P = 0.47$) ($n = 9$ mice). **f**, ACTR simulations with low (small pink dots, $n = 6$) or high (large blue dots, $n = 6$) initial reward-related sensory input exhibited a significant correlation between initial (trials 1–100) predicted mDA reward response and final reward collection latency. a.u., arbitrary units. All error bars denote ±s.e.m.

equivalent cued performance gain (Extended Data Fig. 2e)). Learning trajectories and cost surfaces calculated from a range of model initializations compared well qualitatively and quantitatively to those inferred from mouse data (Fig. 1i, Extended Data Fig. 2c and Methods). Finally, modelling the adaptive rate term ($\beta$) after phasic mDA activity (see Fig. 4 for comparison of modelled to actual) was well supported by comparing end performance of the full model to modified versions (Fig. 1j) in which: the mDA-like feedback unit signalled PE instead of rate (Extended Data Fig. 2f; significantly worse performance, rank sum $P < 2 \times 10^{-7}$); learning rate was globally reduced (akin to dopamine depletion[37]; significantly worse performance, rank sum $P < 2 \times 10^{-6}$); a basal learning rate was intact but there was no adaptive component (akin to disruption of phasic mDA reward signalling[38]; significantly worse performance, rank sum $P = 0.02$). Thus, naive trace conditioning is well described as the optimization of reward collection behaviour, and best approximated when mDA-like signals act not as signed errors

directing changes to the policy, but instead adapting the size of the learned update on each trial.

## Change in dopamine activity over learning

We measured mDA activity in the above mice, which were DAT-Cre::ai32 mice that transgenically expressed Chr2 under control of the dopamine transporter promoter, by injecting a Cre-dependent jRCaMP1b virus across the ventral midbrain[9] (Fig. 2a,b). This combined optogenetic–fibre photometry strategy also allowed for calibrated dopamine manipulations in later experiments. Optical fibres were implanted bilaterally over the VTA, and unilaterally in the nucleus accumbens core (NAc), and in the dorsomedial striatum (DS; Fig. 2a). We recorded jRCaMP1b signals from the NAc ('NAc–DA') in all mice, with some additional simultaneous recordings from ipsilateral VTA ($n = 3$, 'VTA–DA') or contralateral DS ($n = 6$, 'DS–DA'). NAc–DA reward responses became better aligned to reward delivery across training but did not decrease significantly (trials 1–100: $0.82 \pm 0.21$ z, trials 700–800: $1.16 \pm 0.23$ z, signed rank $P = 0.13$), even as cue responses steadily increased (Extended Data Fig. 3a,b). These dynamics recapitulated our previous observations from somatic activity[9] and indeed closely resembled simultaneously recorded VTA–DA signals (Extended Data Fig. 3a–e). By contrast, DS–DA developed cue and reward responses only on further training, matching previous reports[39,40] (Extended Data Fig. 3a–e), and indicating that mesolimbic (for example, VTA-to-NAc) reward signals are of specific interest during the initial learning period studied here.

We thus proceeded to examine how individual differences in mesolimbic reward signals were related to the individual differences in behavioural learning. We found substantial inter-animal variance in initial NAc–DA responses in the first 100 trials that was not related to anatomical location of fibres (Fig. 2c,d; initial NAc–DA reward: anterior–posterior: $P = 0.5$, medial–lateral: $P = 0.4$, dorsal–ventral: $P = 0.5$; multiple linear regression all axes, $P = 0.7$). Unexpectedly, initial NAc–DA reward signals were negatively correlated with the amount of preparatory behaviour at the end of training (Fig. 2c and Extended Data Fig. 3f; NAc–DA reward$_{trials\ 1–100}$ versus preparatory index$_{trials\ 700–800}$, $r = -0.85$, $P = 0.004$), as well as the speed of reward collection (Fig. 2e; NAc–DA reward$_{trials\ 1–100}$ versus reward collection latency$_{trials\ 700–800}$, $r = 0.81$, $P = 0.008$). This relationship was specific for preparatory behaviours (Extended Data Fig. 3g; NAc–DA reward$_{trials\ 1–100}$ versus reactive index$_{trials\ 700–800}$, $P = 0.24$), and was robust as each mouse's initial dopamine reward signals could be accurately predicted from quantifications of final preparatory behaviour (Extended Data Fig. 3h; actual versus predicted $r = 0.99$, $P < 0.0001$).

The negative correlation between dopaminergic reward signaling and behavioural learning is not consistent with the magnitude of phasic mDA activity determining or correlating with the error used for a learning update. However, it is potentially consistent with phasic mDA activity reflecting action-related and sensory-related components of the control policy. At initialization of the ACTR model, no preparatory actions have been learned, so the dopamine signal is dominated by the initial reactive response to sensory input at reward delivery. Adjusting the strength of this sensory input at model initialization scales the initial dopamine reward response magnitudes similarly to the range observed in mice (Fig. 2f). These initialization differences in ACTR simulations predicted a delayed collection latency at the end of training (Extended Data Fig. 3i; $r = 0.73$, $P = 0.007$) due to a reduced development of a preparatory licking policy (Extended Data Fig. 3j), mirroring our results in mice and demonstrating that stronger reactive responses to sensory information can impair the learned development of preparatory responses (and thus ultimately impair performance). These insights from the ACTR model suggest that, in real mice, initial sensitivity to reward-related sensory stimuli is reported by increased mDA reward signalling, and this initial condition can explain meaningful individual differences in the courses of future learning.

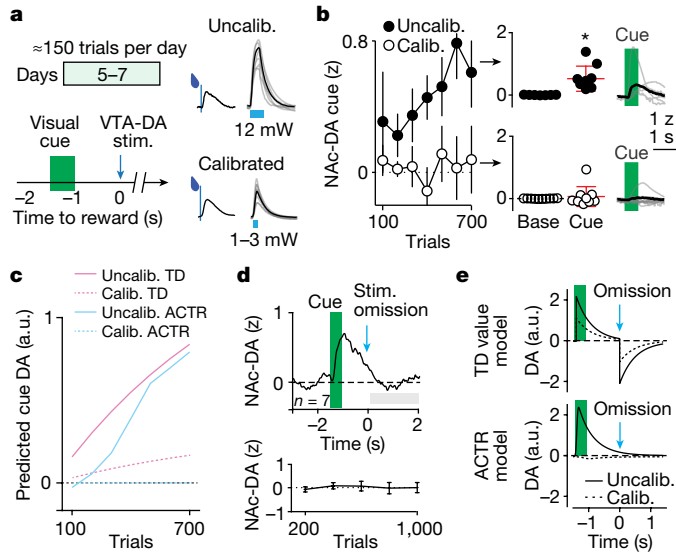

**Fig. 3 | Large mesolimbic dopamine manipulations drive value-like learning. a**, Left: experimental design for VTA–DA stimulation (stim.) predicted by a 0.5-s light flash at the front of the behavioural chamber. Right: mean uncued NAc–DA reward responses versus VTA–DA stimulation responses (right, individuals in grey, mean in black) for mice that received either large, uncalibrated (Uncalib.) stimulation (top; 30 Hz, 12 mW for 500 ms) or stimulation calibrated to reward responses (bottom; 30 Hz, 1–3 mW for 150 ms). **b**, Left: jRCaMP1b NAc–DA cue responses across training for mice that received large stimulations (5× the size of reward responses; filled circles, $n = 7$) or calibrated (Calib.) stimulations (1× the size of reward responses; open circles, $n = 10$). Right: quantified and raw data for mean NAc–DA traces after 750 training trials (5 sessions) with uncalibrated (top, two-tailed signed rank versus baseline, $*P = 0.02$, $n = 7$) and calibrated (bottom, two-tailed signed rank versus baseline $P = 0.8$, $n = 10$) stimulation. **c**, Predicted dopamine cue responses for the experiment in **a** simulated with a TD value learning model (light pink) versus ACTR (light blue). **d**, Top: mean NAc–DA signal after seven sessions of uncalibrated stimulation training, on probe trials with omitted stimulation that were delivered 10% of the time, with quantification over training (bottom). **e**, Predicted dopamine responses by TD (top) and ACTR (bottom) models, for uncalibrated (bold line) and calibrated (dotted line) dopamine stimulation. All error bars denote ±s.e.m.

## Calibrated dopamine stimulation

Work exploring direct roles of dopamine in movement[10] or motivation[41] suggests that phasic cue responses provoke or invigorate preparatory behaviour. Indeed, learned NAc–DA cue responses were correlated with cued licking across mice (Extended Data Fig. 4c). However, at the end of regular training some mice experienced an extra session in which VTA–DA stimulation was triggered on cue presentation on a random subset of trials (Extended Data Fig. 4). Increasing mesolimbic cue responses in this way had no effect on cued licking in the concurrent trial (control 2.3 ± 1.1 Hz, stimulation 2.3 ± 1.0 Hz, $P > 0.99$). Thus, within this context (although not necessarily others[42]), the magnitude of NAc–DA cue signals correlates only with learned changes in behavioural policy but does not seem to directly regulate preparatory behaviour in anticipation of reward delivery[9,43].

Notably, individual differences in initial NAc–DA reward signals were not correlated with the learning of NAc–DA cue signals (reward$_{trials 1–100}$ versus cue$_{trials 700–800}$, $P = 0.5$; Fig. 2e). This could argue that dopamine reward signals are not a driving force in the learning of cue signals. This is surprising given that results in rodents[43,44] and monkeys[45] provide specific evidence for value learning effects following exogenous VTA–DA stimulation. However, reward-related mDA bursting is brief (≤0.2 s) in our task[9] as well as in canonical results across species[4],

raising the question of whether high-power, longer-duration stimulation recruits the same learning mechanisms as briefer, smaller reward-sized responses. We next used our ability to simultaneously manipulate and measure mesolimbic dopamine[9] to examine the function of brief dopamine transients calibrated to match reward responses from our task.

Following initial training on the trace conditioning paradigm (Supplementary Table 1), we introduced mice to a novel predictive cue—a 500-ms flash of light directed at the chamber wall in front of the mouse. After ten introductory trials, this visible cue stimulus was paired with exogenous VTA–DA stimulation after 1 s delay for five daily sessions (about 150 trials per session; Fig. 3a). One group of randomly selected mice received VTA–DA stimulation calibrated to uncued reward responses (150 ms at 30 Hz and 1–3 mW steady-state power, stimulation response = 1.4 ± 0.3 uncued reward response, $n = 10$), whereas the complement received larger, uncalibrated stimulations (500 ms at 30 Hz and 10 mW steady-state power, stimulation response = 5.5 ± 0.8 uncued reward response, $n = 7$; Fig. 3a,b). After five sessions, the group receiving calibrated, reward-sized stimulation did not exhibit NAc–DA cue responses above baseline (0.0 ± 0.2 z, $P = 0.8$), whereas the large, uncalibrated stimulation group exhibited substantial NAc–DA cue responses (0.5 ± 0.2 z, $P = 0.02$; Fig. 3b).

The emergence of cue signals following uncalibrated dopamine stimulation was captured in ACTR by introducing a nonlinearity in which larger, more sustained dopamine activation was modelled as a large modulation of learning rate coupled with a change in PE encoding (Fig. 3c–e and Methods). This coupled effect in the model enhanced cue encoding in a manner similar to the predictions of value learning; however, it was also distinct in that this change in the encoding of cues was not accompanied by the suppression of dopamine activity on omission of the laser stimulus expected from value learning models (Fig. 3d,e). In contrast to previous observations of inhibition following omission of expected stimulation in the context of consistent, overt behavioural responses to cues[17,46], only a brief bout of body movement accompanied cue learning in the current paradigm (Extended Data Fig. 5). This suggests that dopamine inhibition observed following omission of expected rewards may depend on concurrent control[9,13] or evaluation[17] of action.

In separate experiments, calibrated and uncalibrated VTA–DA stimulations had a similar spatial profile of response magnitude across the medial prefrontal cortex and the dorsal-to-ventral axis of the striatum, suggesting that the recruitment of value-like cue learning by uncalibrated stimulation was related to the magnitude or duration of the uncalibrated signal rather than an increased spatial spread (Extended Data Fig. 6). Together, these data provide further evidence that phasic mDA reward responses of the magnitude measured in our task are not sufficient to drive value-like learning of predictive cue responses, but larger stimulation flooding the same downstream regions with higher dopamine concentrations are sufficient to teach phasic responses to a cue that predicts dopamine stimulation.

## Dopamine sets an adaptive learning rate

We next elaborate on the role of dopamine reward signals in performance-driven direct policy learning. NAc–DA signals predicted by ACTR and temporal difference (TD) value learning can be visualized by convolving their dopamine signals (the adaptive rate signal in ACTR and the RPE signal from the optimized TD model parameters; Extended Data Fig. 1) with a temporal kernel matched to the kinetics of jRCaMP1b (Fig. 4 and Methods). ACTR's predicted phasic mDA photometry signal corresponds closely to experimentally measured NAc– and VTA–DA activity across training (Fig. 4a–d and Extended Data Fig. 3a). Notably, ACTR's modelling of mDA activity as the sum of action and sensory components in a control policy reproduces the emergence of differences between expected and unexpected rewards, despite the

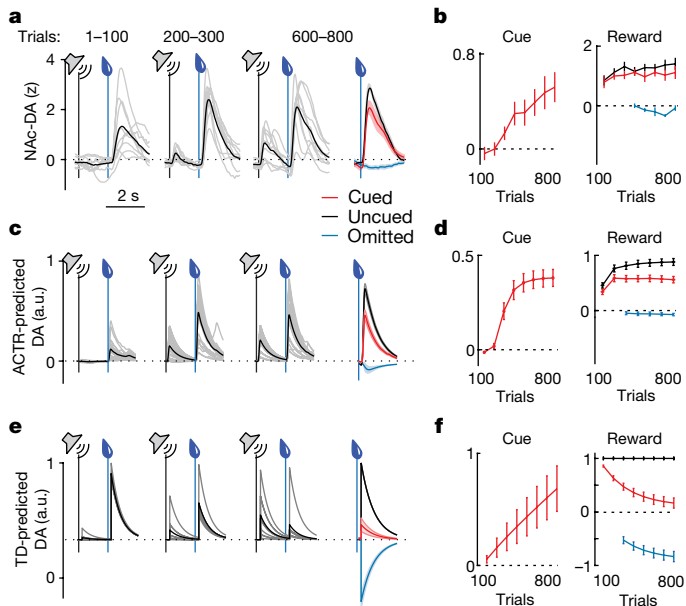

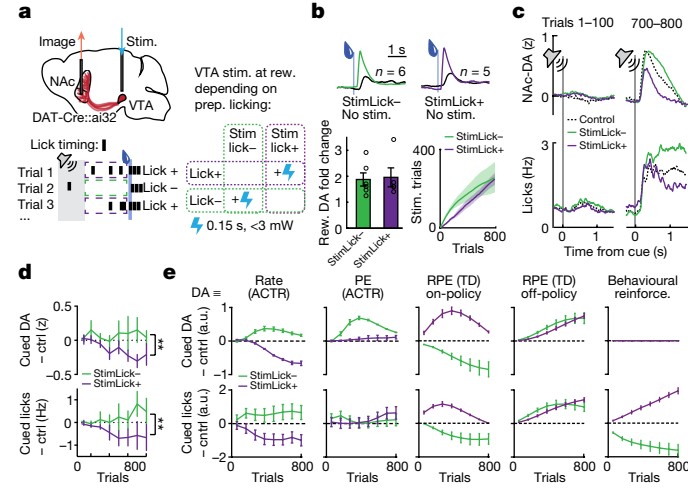

**Fig. 4 | Dopaminergic adaptive rate signals correlate with RPEs. a**, Measured NAc–DA responses for mice across training, including mean responses to reward on cued (red), uncued (black) and omission (blue, aligned to point at which reward should have been delivered) trials. **b**, Mean cue and reward responses binned in 100-trial blocks across training (*n* = 9). **c,d**, Same as **a,b** except for predicted dopamine responses in the ACTR model (*n* = 9 model initializations). **e,f**, As in **a,b** except for predicted dopamine responses in the TD learning model (*n* = 9 model initializations) using parameters that gave the best fit to behavioural learning (from Extended Data Fig. 1). All error bars denote ±s.e.m.

**Fig. 5 | Closed-loop manipulation of mesolimbic dopamine distinguishes learning rate signalling from error signalling. a**, Top: simultaneous measurement and manipulation of mesolimbic dopamine. Bottom: closed-loop experiment design, in which different groups of mice received bilateral VTA stimulation (0.15 s, 30 Hz, power calibrated to roughly double control reward response) concurrent with reward delivery on either lick− trials ('stimLick−') or lick+ trials ('stimLick+'). **b**, Top: mean NAc–DA reward responses across training (trials 1–800) for each mouse, with (coloured traces) and without (black traces) exogenous stimulation, for stimLick− (left and green, *n* = 6) and stimLick+ (right and purple, *n* = 5) cohorts of mice. Bottom: fold increase in NAc–DA reward signals on stimulated trials and cumulative sum of stimulated trials across training for stimLick− (green) and stimLick+ (purple). **c**, NAc–DA (top) and licking (bottom) during early (trials 1–100, left) and late (trials 700–800, right) training for control (black), stimLick− (green) and stimLick+ (purple) animals. **d**, NAc–DA cue responses (top) and cued licking (bottom) for stimLick− (green, *n* = 6) and stimLick+ (purple, *n* = 5) mice across training, shown as the difference from control mice. Two-way ANOVA, **P = 0.002. **e**, Modelling results for the difference of dopamine cue responses (top) and cued licking (bottom) from control in stimLick− and stimLick+ contingencies, compared for five possible functions of phasic dopamine signals. Left to right: the adaptive rate term (*β*) in the ACTR policy learning model, biasing the PE term in the ACTR, the RPE signal in on-policy and off-policy TD value learning models, or reinforcing contingent behaviour without affecting learning of dopamine signals (*n* = 9 simulations for each condition (control, stimLick− and stimLick+) for each model version (Methods)). All error bars denote ±s.e.m.

lack of an explicit RPE computation (Fig. 4b and Extended Data Fig. 7). This scheme also predicts that mDA reward signals should reflect the evolution of reward collection policy across learning. Although animals' policies are not directly observable, the presence or absence of preparatory licking on a given trial of behaviour is a noisy correlate of differences in underlying behavioural policy. Indeed, both ACTR and mouse data exhibited differential reward responses on trials with ('lick+') or without ('lick−') preparatory licking as learning progressed (Extended Data Fig. 8a–d).

A close examination of the learning signals on lick− versus lick+ trials indicates that those trial types are capturing different distributions of PEs, as estimated from reward collection latency and anticipatory licking (Methods). Although lick+ trials are critical for optimal performance, generally lick+ trials are associated with negative PEs and lick− trials are associated with positive PEs (Extended Data Fig. 8f). Our modelling provides some insight into why this effect is expected and can be most easily illustrated by considering the limiting cases. If the policy is optimal then trials are lick+ and stochasticity in the licking plant or fluctuations in the policy can result only in deviations towards worse than expected reward collection latencies (that is, resulting in a bias towards negative PEs). By contrast, if the policy is as bad as possible, trials are lick− and stochastic initiation of the lick plant right after reward delivery or stochastic fluctuations in the underlying policy (even without inducing a pre-reward lick) can only improve reward collection relative to expected latency of a poor policy (that is, resulting in a bias towards positive PEs).

It then follows that disrupting the balance of dopamine signalling between lick+ and lick− trials should systematically affect learning given the biases in PEs between the two trial types. Owing to their average negative PE, increasing the learning rate exogenously (by increasing the dopamine signal) only on lick+ trials should bias away

from robust preparatory policies and decrease final learned preparatory licking. The converse is true for lick− trials, for which selective amplification of positive PEs should bias to more preparatory licking. These paradoxical effects of enhanced rates impairing learning of the contingent behaviour can be demonstrated in ACTR simulations, as trial-type-dependent enhancement of the dopaminergic learning rate signal indeed produced opposite signed effects on preparatory licking behaviour (Extended Data Fig. 8e). Furthermore, this closed-loop stimulation paradigm offers the unique ability to distinguish between many competing models of dopamine function: when dopamine reward signals are modelled as signed errors (PEs in ACTR or RPEs in value learning models) or as a simple behavioural reinforcement signal, the same closed-loop stimulation paradigm biases dopamine cue signals and preparatory licking in opposite directions for at least one of the two trial types (Fig. 5e).

We thus performed this experiment in mice, selectively increasing dopamine reward signals through optogenetic stimulation in the VTA contingent on preparatory cued behaviour. Separate groups of animals experienced each of the following stimulation contingencies: 'stimLick+' animals received VTA–DA stimulation at the moment of reward delivery on trials in which we detected licking in the 750 ms

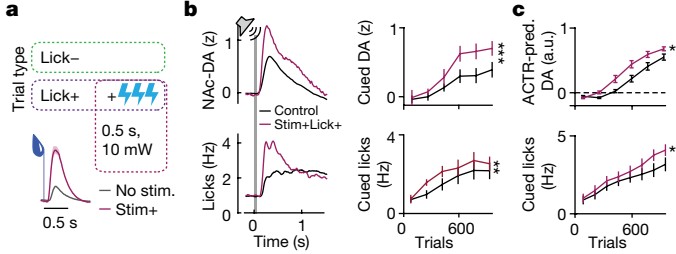

**Fig. 6 | Supraphysiological dopamine stimulation signals error. a**, Top: experimental design for new group of mice that experienced a 'stim+Lick+' contingency: they received large, uncalibrated VTA–DA stimulation on lick+ trials. Bottom: NAc–DA reward responses on stimulated (magenta) and unstimulated (grey) trials. **b**, NAc–DA cue responses (top) and cued licking (bottom) for control (black, $n = 9$) and stim+Lick+ (light purple, $n = 4$) mice. Two-way ANOVA, $**P = 0.01$, $***P = 0.001$. **c**, Predicted cue responses (top) and cued licking (bottom) in a version of the ACTR model altered for the large-amplitude stimulations to bias PE in addition to setting the adaptive learning rate ($n = 9$ simulations). Two-way ANOVA, $*P < 0.05$. All error bars denote ±s.e.m.

preceding reward delivery, whereas 'stimLick−' animals received the same stimulation on trials in which no licking was detected during the delay interval (Fig. 5a). Crucially, stimulation was brief (150 ms) and low power (3 mW), approximately doubling the endogenous NAc–DA reward response (Fig. 5b). To account for the large discrepancy in stimulated trials that would arise between the two stimulation groups due to eventual predominance of lick+ trials, stimLick+ animals were limited to having a maximum of 50% of total trials stimulated in a given session. This resulted in a comparable number of stimulated trials between the two groups by the end of the training period (Fig. 5b). We also confirmed post hoc that stimulation captured systematically different ($P = 0.004$, rank sum test) positive and negative PEs for each condition as expected (Extended Data Fig. 8g).

Calibrated enhancement of reward-related activity in mesolimbic projections in this way had opposite effects on emerging delay-period behaviour across the two stimLick contingencies (Fig. 5b–d and Extended Data Fig. 8h). As in the ACTR model, behaviour was biased in opposite directions for each contingency, with stimLick+ animals exhibiting lower and stimLick− animals exhibiting higher preparatory licking (trials 600–800, stimLick+ 1.0 ± 0.7, stimLick− 0.6 ± 0.1, analysis of variance (ANOVA) $F_{1,72} = 10.5$, $P = 0.002$; Fig. 5d). Furthermore, NAc–DA cue signals were biased in matching directions, with the stimLick− group also exhibiting higher NAc-DA cue responses versus stimLick+ (trials 600–800, stimLick+ 0.3 ± 0.1 z, stimLick− 2.6 ± 0.7 z, ANOVA $F_{1,72} = 10.1$, $P = 0.002$; Fig. 5d). Baseline licking examined just before trials began across training showed no correlation with the extent of learning ($P = 0.9$) or initial NAc–DA magnitude ($P = 0.8$), confirming that preparatory licking learning was indeed driven by the predictive cue (Extended Data Fig. 9).

The differences in effects of calibrated and uncalibrated stimulations (Fig. 3) suggest that uncalibrated stimulation could paradoxically reverse the effect on suppression of cued licking seen in the stimLick+ condition above. To test this possibility, we repeated the stimLick+ experiment with a new set of mice, but this time augmented rewards on lick+ trials with large, uncalibrated VTA–DA stimulation (500 ms, at 30 Hz and about 10 mW power; Fig. 6a). Indeed, this new larger exogenous stimulation contingency ('stim+Lick+') now resulted in increased NAc–DA cue responses (Fig. 5b, two-way ANOVA, stimulation group $F_{1,66} = 11.7$, $P = 0.001$) and increased cued licking (Fig. 5b, two-way ANOVA, stimulation group $F_{1,60} = 7.1$, $P = 0.01$), reversing the sign of the effects of calibrated stimLick+ stimulation (Fig. 5d). These effects were well predicted by a modified version of the ACTR model in which large dopamine stimulations biased towards positive

errors in addition to modulating learning rate (Fig. 6c), exactly as in the previous experiment in which large uncalibrated stimulation caused the emergence of NAc–DA responses to a predictive cue (Fig. 3a–c).

## Discussion

The discovery that the phasic activity of mDA neurons in several species correlated with a key quantity (RPE) in value learning algorithms has been a marked and important advance suggesting that the brain may implement analogous processes[4,7]. At the same time, reinforcement learning constitutes a large family of algorithms[1,3] that include not only learning about expected values of environmental states, but also directly learning parameterized policies for behavioural control. Close analysis of our trace conditioning paradigm indeed revealed that behavioural learning was better explained by direct policy learning as compared to value learning. The fact that signed RPEs are not required for policy learning opened up the possibility that mDA activity could map onto a different quantity. This led us to develop a biologically plausible network-based formulation of policy learning that is consistent with many aspects of individual behavioural trajectories, but also closely matches observed mDA neuron activity during naive learning. This is distinct from standard 'actor–critic' models of dopamine function in the basal ganglia[47] in multiple ways, including in the function of dopamine in the model and the proposition that the ventral striatum is part of the 'actor' that determines policy, rather than the 'critic'. However, both emphasize the need to understand how policy is implemented in dopamine-recipient circuits as an abstracted action-control signal.

Regardless of the specific reinforcement learning algorithm favoured, our analyses and experiments discriminate between two potential biological functions of dopamine: a signed error signal that governs the direction of learned changes and an unsigned adaptive rate signal that governs how much of the error is captured on a given trial. A role in modulating learning rate as opposed to signalling an error predicts that stimulation of mDA neurons will often enhance learning as previously observed, but could paradoxically slow learning in some contingencies. Slowed learning would be paradoxical if dopamine activity functioned as an error or as a reinforcer of past action. Stimulation of mDA neurons without respect to ongoing behaviour, as routinely carried out, fails to distinguish between these possibilities and thus a new experiment was required—calibrated manipulation of mDA activity in closed-loop operation with behaviour (inferred policy state). We found a remarkable agreement between policy learning model-based predictions and experimental observations. Intriguingly, we also discovered that uncalibrated mDA stimulation 3–5 times stronger than endogenous mDA activity (but with parameters common in the field at present) was well explained by our model as a bias in a signed error in addition to modulating learning rate. This suggests that dopamine-dependent attribution of motivational value to cues[46,48,49] is at least partially dissociable from the regulation of policy learning rate within the same mesolimbic circuits. Such parallel functions could be complementary, intriguingly mirroring the system of parallel policy and value learning networks implemented in AlphaGo[50], a landmark achievement in modern artificial intelligence.

Value-like error signalling following higher-power, longer stimulation may depend on specific receptor recruitment within a circuit[51] (as suggested by similar input–output relationships across tested regions (Extended Data Fig. 6)), and/or differential recruitment of diverse[8,52–54] dopaminergic circuits. This predicts that recipient areas should exhibit distinct electrophysiological responses to supraphysiological stimulation, which can be tested in future experiments. Our data indicate that both strongly enhanced dopamine signalling and oversensitivity to sensory input can bias towards value-like learning that leads an animal to exhibit excessively strong reactive responses to

cues at the expense of optimal behaviour. This may be akin to the excessive acquired and innate sensitivity to drug-predictive cues thought to underlie the development of addiction[55], and connects our results to previous observations of correlations between the magnitude of phasic dopamine signalling and individual differences in reward-related behaviours[49]. This suggests that policy learning, and specifically the reactive component in our ACTR model, may be a useful way to model the acquisition of incentive salience[41] (although not its expression, as phasic dopamine signals could be shown to modulate only learning, and not apparent incentive salience on the current trial (Extended Data Fig. 4)). Our results promote the practice of matching exogenous manipulations of neuromodulators to physiological signals[9,56], and support the modelling of addictive maladaptive learning[57] with extended, high-magnitude mDA stimulation.

There are many opportunities to extend the current ACTR model formulation to capture more biological reality and evaluate the biologically plausible, but so far incompletely tested, cellular and circuit mechanisms of posited ACTR learning rules. There is prior evidence for the capacity of mDA activity to capture eligibility traces and modulate synaptic plasticity[58,59]; however, our behavioural data and modelling call for further examination of multiple coordinated learning rules governing reactive-like and preparatory-like learning. Given that adaptive control over the magnitude of learning rate can be a key determinant of machine learning performance in deep neural networks[2,28] and RNNs[35], studying how adaptive control of learning rates is implemented in animal brains, and especially across diverse tasks, may provide additional algorithmic insights to those developed here. Recent evidence also suggests that other neuromodulators in the brain may play distinct, putatively complementary roles in controlling the rate of learning[60]. Here we effectively identify a key heuristic apparent in phasic mDA activity that adapts learning rates to produce more stable and performant learning; however, we focused on a single behavioural learning paradigm and dopamine is known to be critical for a broad range of putative behavioural policies. Our work provides a perspective for future work to expand on and identify other aspects of mDA activity that may be critical for the adaptive control of learning from action.

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

# Methods

## Animals

All protocols and animal handling procedures were carried out in strict accordance with a protocol (no. 19-190) approved by the Janelia Institutional Animal Care and Use Committee and in compliance with the standards set forth by the Association for Assessment and Accreditation of Laboratory Animal Care.

For behaviour and juxtacellular recordings, we used 24 adult male DAT-Cre::ai32 mice (3-9 months old) resulting from the cross of DAT$^{IREScre}$ (The Jackson Laboratory stock 006660) and Ai32 (The Jackson Laboratory stock 012569) lines of mice, such that a Chr2-EYFP fusion protein was expressed under control of the endogenous dopamine transporter *Slc6a3* locus to specifically label dopaminergic neurons. Mice were maintained under specific-pathogen-free conditions. Mice were housed on a free-standing, individually ventilated (about 60 air changes hourly) rack (Allentown Inc.). The holding room was ventilated with 100% outside filtered air with >15 air changes hourly. Each ventilated cage (Allentown) was provided with corncob bedding (Shepard Specialty Papers), at least 8 g of nesting material (Bed-r'Nest, The Andersons) and a red mouse tunnel (Bio-Serv). Mice were maintained on a 12:12-h (8 am-8 pm) light/dark cycle and recordings were made between 9 am and 3 pm. The holding room temperature was maintained at 21 ± 1 °C with a relative humidity of 30% to 70%. Irradiated rodent laboratory chow (LabDiet 5053) was provided ad libitum. Following at least 4 days recovery from headcap implantation surgery, animals' water consumption was restricted to 1.2 ml per day for at least 3 days before training. Mice underwent daily health checks, and water restriction was eased if mice fell below 75% of their original body weight.

## Behavioural training

Mice were habituated to head fixation in a separate area from the recording rig in multiple sessions of increasing length over ≥3 days. During this time they received some manual water administration through a syringe. Mice were then habituated to head fixation while resting in a spring-suspended basket in the recording rig for at least two sessions of 30+ min each before training commenced. No liquid rewards were administered during this recording rig acclimation; thus, trial 1 in the data represents the first time naive mice received the liquid water reward in the training environment. The reward consisted of 3 μl of water sweetened with the non-caloric sweetener saccharin delivered through a lick port under control of a solenoid. A 0.5-s, 10-kHz tone preceded reward delivery by 1.5 s on 'cued' trials, and 10% of randomly selected rewards were 'uncued'. Matching our previous training schedule[9], after three sessions, mice also experienced 'omission' probe trials, in which the cue was delivered but not followed by reward, on 10% of randomly selected trials. Intertrial intervals were chosen from a randomly permuted exponential distribution with a mean of about 25 s. Ambient room noise was 50-55 dB, and an audible click of about 53 dB accompanied solenoid opening on water delivery and the predictive tone was about 65 dB loud. Mice experienced 100 trials per session and one session per day for 8-10 days. In previous pilot experiments, it was observed that at similar intertrial intervals, behavioural responses to cues and rewards began to decrease in some mice at 150-200 trials. Thus, the 100 trials per session limit was chosen to ensure homogeneity in motivated engagement across the dataset.

Some animals received optogenetic stimulation of VTA-DA neurons concurrent with reward delivery, contingent on their behaviour during the delay period (see technical details below). Mice were randomly assigned to stimulation group (control, stimLick−, stimLick+) before training. Experimenter was not blinded to group identity during data collection. Following trace conditioning with or without exogenous dopamine stimulation, five mice experienced an extra session during which VTA-DA neurons were optogenetically stimulated concurrently with cue presentation (Extended Data Fig. 4). Mice were then randomly assigned to groups for a new experiment in which a light cue predicted VTA-DA stimulation with no concurrent liquid water reward (5-7 days, 150-200 trials per day). The light cue consisted of a 500-ms flash of a blue light-emitting diode (LED) directed at the wall in front of head fixation. Intertrial intervals were chosen from randomly permuted exponential distributions with a mean of about 13 s. Supplementary Table 1 lists the experimental groups each mouse was assigned to in the order in which experiments were experienced.

## Video and behavioural measurement

Face video was captured at 100 Hz continuously across each session with a single camera (Flea 3, FLIR) positioned level with the point of head fixation, at an approximately 30° angle from horizontal, and compressed and streamed to disk with custom code written by J. Keller (available at https://github.com/neurojak/pySpinCapture). Dim visible light was maintained in the rig so that pupils were not overly dilated, and an infrared LED (model#) trained at the face provided illumination for video capture. Video was post-processed with custom MATLAB code available on request.

Briefly, for each session, a rectangular region of interest (ROI) for each measurement was defined from the mean of 500 randomly drawn frames. Pupil diameter was estimated as the mean of the major and minor axis of the object detected with the MATLAB regionprops function, following noise removal by thresholding the image to separate light and dark pixels, then applying a circular averaging filter and then dilating and eroding the image. This noise removal process accounted for frames distorted by passage of whiskers in front of the eye, and slight differences in face illumination between mice. For each session, appropriateness of fit was verified by overlaying the estimated pupil on the actual image for about 20-50 randomly drawn frames. A single variable, the dark/light pixel thresholding value, could be changed to ensure optimal fitting for each session. Nose motion was extracted as the mean of pixel displacement in the ROI *y* axis estimated using an image registration algorithm (MATLAB imregdemons). Whisker pad motion was estimated as the absolute difference in the whisker pad ROI between frames (MATLAB imabsdiff; this was sufficiently accurate to define whisking periods, and required much less computing time than imregdemons). Whisking was determined as the crossing of pad motion above a threshold, and whisking bouts were made continuous by convolving pad motion with a smoothing kernel. Licks were timestamped as the moment pixel intensity in the ROI in between the face and the lick port crossed a threshold.

Body movement was summarized as basket movements recorded by a triple-axis accelerometer (Adafruit, ADXL335) attached to the underside of a custom-designed three-dimensionally printed basket suspended from springs (Century Spring Corp, ZZ3-36). Relative basket position was tracked by low-pass filtering accelerometer data at 2.5 Hz. Stimulations and cue deliveries were coordinated with custom-written software using Arduino Mega hardware (https://www.arduino.cc). All measurement and control signals were synchronously recorded and digitized (at 1 kHz for behavioural data, 10 kHz for fibre photometry data) with a Cerebus Signal Processor (Blackrock Microsystems). Data were analysed using MATLAB software (Mathworks).

## Preparatory and reactive measures and abstract learning trajectories

To describe the relationship between behavioural adaptations and reward collection performance, for each mouse in the control group a GLM was created to predict reward collection latency from preparatory and reactive predictor variables on each trial. Preparatory changes in licking, whisking, body movement and pupil diameter were quantified by measuring the average of each of those signals during the 1-s delay period preceding cued rewards. The nose motion signal was not included as it did not display consistent preparatory changes. Reactive responses in the whisking, nose motion and body movement were

measured as the latency to the first response following reward delivery. For whisking, this was simply the first moment of whisking following reward delivery. For nose motion, the raw signal was convolved with a smoothing kernel and then the first response was detected as a threshold crossing of the cumulative sum of the signal. For body movement, the response was detected as the first peak in the data following reward delivery. On occasional trials no event was detected within the analysis window. Additionally, discrete blocks of trials were lost owing to data collection error for mouse 3, session 7; mouse 4, session 5; and mouse 9, session 4. To fit learning curves through these absent data points, missing trials were filled in using nearest-neighbour interpolation.

Trial-by-trial reward collection latencies and predictor variables (preparatory licking, whisking, body movement and pupil diameter; and reactive nose motions, whisking and body movement) were median filtered (MATLAB medfilt1(signal,10)) to minimize trial-to-trial variance in favour of variance due to learning across training. Collection latency was predicted from $z$-scored predictor variables using MATLAB glmfit to fit $\beta$-values for each predictor. The unique explained variance of each predictor was calculated as the difference in explained variance between the full model and a partial model in which $\beta$-values were fitted without using that predictor.

Preparatory and reactive predictor variables were used to define abstract learning trajectories that were plots of collection latency against the inferred reactive and preparatory variables for each of the first 800 cue–reward trials of training. Reactive and preparatory variables were calculated as the first principal component of the individual reactive and preparatory variables used in the GLM fits. For visualization, we fitted a parametric model to all three variables (single exponential for preparatory, double exponentials for reactive and latency using the MATLAB fit function). Quality of fits and choice of model were verified by visual inspection of all data for all mice. An individual mouse's trajectory was then visualized by plotting downsampled versions of the fit functions for latency, reactive and preparatory. Arrowheads were placed at logarithmically spaced trials.

To quantify the total amount of preparatory behaviour in each mouse at a given point in training (final prep. behav., Extended Data Fig. 3f), each preparatory measure (pupil, licking, whisking and body movement) was $z$-scored and combined across mice into a single data matrix. The first principal component of this matrix was calculated and loading onto PC1 was defined as a measure of an inferred underlying 'preparatory' component of the behavioural policy. This created an equally weighted, variance-normalized combination of all preparatory measures to allow comparisons between individual mice. An analogous method was used to reduce the dimensionality of reactive variables down to a single 'reactive' dimension that captures most variance in reactive behavioural variables across animals (final reactive behav., Extended Data Fig. 3g). Initial NAc–DA signals were predicted from trained behaviour at trials 700–800 by multiple regression (specifically, pseudoinverse of the data matrix of reactive and preparatory variables at the end of training multiplied by data matrix of physiological signals for all animals).

**Combined fibre photometry and optogenetic stimulation**
In the course of a single surgery session, DAT-Cre::ai32 mice received: bilateral injections of AAV2/1-CAG-FLEX-jRCaMP1b in the VTA (150 nl at the coordinates −3.1 mm anterior–posterior (A–P), 1.3 mm medial–lateral (M–L) from bregma, at depths of 4.6 and 4.3 mm) and in the substantia nigra pars compacta (100 nl at the coordinates −3.2 mm A–P, 0.5 mm M–L, depth of 4.1, mm); custom 0.39-NA, 200-µm fibre cannulas implanted bilaterally above the VTA (−3.2 mm A–P, 0.5 mm M–L, depth of −4.1 mm); and fibre cannula implanted unilaterally in the DS (0.9 mm A–P, 1.5 mm M–L, depth of 2.5 mm) and NAc (1.2 mm A–P, 0.85 mm M–L, depth of 4.3 mm). Hemisphere choice was counterbalanced across individuals. A detailed description of the methods has been published previously[56].

Imaging began >20 days post-injections using custom-built fibre photometry systems (Fig. 2a)[56]. Two parallel excitation–emission channels through a five-port filter cube (FMC5, Doric Lenses) allowed for simultaneous measurement of RCaMP1b and eYFP fluorescence, the latter channel having the purpose of controlling for the presence of movement artefacts. Fibre-coupled LEDs of 470 nm and 565 nm (M470F3, M565F3, Thorlabs) were connected to excitation ports with acceptance bandwidths of 465–490 nm and 555–570 nm, respectively, with 200-µm, 0.22-NA fibres (Doric Lenses). Light was conveyed between the sample port of the cube and the animal by a 200-µm-core, 0.39-NA fibre (Doric Lenses) terminating in a ceramic ferrule that was connected to the implanted fibre cannula by a ceramic mating sleeve (ADAL1, Thorlabs) using index matching gel to improve coupling efficiency (G608N3, Thorlabs). Light collected from the sample fibre was measured at separate output ports (emission bandwidths 500–540 nm and 600–680 nm) by 600-µm-core, 0.48-NA fibres (Doric Lenses) connected to silicon photoreceivers (2151, Newport).

A time-division multiplexing strategy was used in which LEDs were controlled at a frequency of 100 Hz (1 ms on, 10 ms off), offset from each other to avoid crosstalk between channels. A Y-cable split each LED output between the filter cube and a photodetector to measure output power. LED output power was 50–80 µW. This low power combined with the 10% duty cycle used for multiplexing prevented local ChR2 excitation[56] by 473 nm eYFP excitation. Excitation-specific signals were recovered in post-processing by only keeping data from each channel when its LED output power was high. Data were downsampled to 100 Hz, and then band-pass filtered between 0.01 and 40 Hz with a second-order Butterworth filter. Although movement artefacts were negligible when mice were head-fixed in the rig (the movable basket was designed to minimize brain movement with respect to the skull[9]), according to standard procedure the least-squares fit of the eYFP movement artefact signal was subtracted from the jRCaMP1b signal. d$F/F$ was calculated by dividing the raw signal by a baseline defined as the polynomial trend (MATLAB detrend) across the entire session. This preserved local slow signal changes while correcting for photobleaching. Comparisons between mice were carried out using the z-scored d$F/F$.

Experimenters were blind to group identity during the initial stages of analysis when analysis windows were determined and custom code was established to quantify fibre photometry signals and behavioural measurements. Analysis windows were chosen to capture the extent of mean phasic activations following each kind of stimulus. For NAc–DA and VTA–DA, reward responses were quantified from 0 to 2 s after reward delivery and cue responses were quantified from 0 to 1 s after cue delivery. DS–DA exhibited much faster kinetics, and thus reward and cue responses were quantified from 0 to 0.75 s after delivery.

Somatic Chr2 excitation was carried out with a 473-nm laser (50 mW, OEM Laser Systems) coupled by a branching fibre patch cord (200 µm, Doric Lenses) to the VTA-implanted fibres using ceramic mating sleeves. Burst activations of 30 Hz (10 ms on, 23 ms off) were delivered with durations of either 150 ms for calibrated stimulation or 500 ms for large stimulations. For calibrated stimulation, laser power was set between 1 and 3 mW (steady-state output) to produce a NAc–DA reactive of similar amplitude to the largest transients observed during the first several trials of the session. This was confirmed post hoc to have roughly doubled the size of reward-related NAc–DA transients (Figs. 3a and 5b). For large stimulations, steady-state laser output was set to 10 mW.

**ACTR computational learning model**
**Behavioural plant.** An important aspect of this modelling work was to create a generative agent model that would produce core aspects of reward-seeking behaviour in mice. To this end, we focused on licking, which in the context of this task is the unique aspect of behaviour critical for reward collection. A reader may look at the function dlRNN_Pcheck_transfer.m within the software repository to appreciate the structure of the plant model. We describe the function of the plant

briefly here. It is well known that during consumptive, repetitive licking mice exhibit preparatory periods of about 7 Hz licking. We modelled a simple fixed rate plant with an active, 'lick' state that emitted observed licks at a fixed time interval of 150 ms. The onset of this lick pattern relative to entry into the lick state was started at a variable phase of the interval (average latency to lick initialization from transition into lick state about 100 ms). Stochastic transitions between 'rest' and 'lick' states were governed by forward and backward transition rates. The reverse transition rate was a constant that depended on the presence of reward ($5 \times 10^{-3}$ ms without reward, $5 \times 10^{-1}$ ms with reward). This change in the backwards rate captured the average duration of consumptive licking bouts. The forward rate was governed by the scaled policy network output and a background tendency to transition to licking as a function of trial time (analogous to an exponential rising hazard function; $\tau = 100$ ms). The output unit of the policy network was the sum of the RNN output unit (constrained {−1,1} by the tanh activation function) and a large reactive transient proportional to the sensory weight ({0,max_scale}), in which max_scale was a free parameter generally bounded from 5 to 10 during initialization. This net output was scaled by $S = 0.02$ ms$^{-1}$ to convert to a scaled transition rate in the policy output. Behaviour of the plant for a range of policies is illustrated in Extended Data Fig. 2. A large range of parameterizations were explored with qualitatively similar results. Chosen parameters were arrived at by scanning many different simulations and matching average initial and final latencies for cue–reward pairings across the population of animals. More complicated versions (high-pass filtered, nonlinear scaling) of the transition from RNN output to transition rate can be explored in the provided code. However, all transformations were found to produce qualitatively similar results, and thus the simplest (scalar) transformation was chosen for reported simulations for clarity of presentation.

**RNN.** As noted in the main text, the RNN component of the model and the learning rules used for training drew on inspiration from ref. [36], which itself drew on inspiration variants of node perturbation methods[61] and the classic policy optimization methods known as REINFORCE rules[3,21]. Briefly, ref. [36] demonstrated that a relatively simple learning rule that computed a nonlinear function of the correlation between a change in input and change in output multiplied by the change in performance on the objective was sufficiently correlated with the analytic gradient to allow efficient training of the RNN. We implemented a few changes relative to this prior work. Below we delve into the learning rule as implemented here or a reader may examine the commented open source code to get further clarification as well. First, we describe the structure of the RNN and some core aspects of its function in the context of the model. The RNN was constructed largely as described in ref. [36], and was very comparable to the structure of a re-implementation of that model in ref. [62].

Although we explored a range of parameters governing RNN construction, many examples of which are shown in Extended Data Fig. 2, the simulations shown in the main results come from a network with 50 units ($N_u = 50$; chosen for simulation efficiency; larger networks were explored extensively as well), densely connected ($P_c = 0.9$), spectral scaling to produce preparatory dynamics ($g = 1.3$), a characteristic time constant ($\tau = 25$ ms) and a standard tanh activation function for individual units. Initial internal weights of the network ($W_{ij}$) were assigned according to the equation (in RNN-dudlab-master-LearnDA.m)

$$W_{ij} = g \times \mathcal{N}(0, 1) \times (P_c \times N_u)^{-1/2} \tag{1}$$

The RNN had a single primary output unit with activity that constituted the continuous time policy (that is, $\pi(t)$) input to the behaviour plant (see above), and a 'feedback' unit that did not project back into the network as would be standard, but rather was used to produce adaptive changes in the learning rate (described in more detail in the section below entitled Learning rules).

**Objective function.** Evaluation of model performance was calculated according to an objective function that defines the cost as the performance cost (equation (2), cost$_P$) and an optional network stability cost (equation (3), cost$_N$) (for example, lines 269 and 387 in dlRNN-train_learnDA.m, for equations (4) and (5), respectively)

$$\text{cost}_P = 1 - e^{-\Delta t/500} \tag{2}$$

$$\text{cost}_N = \text{sum}(|\delta\pi(t)/\delta t|) \tag{3}$$

$$R_{\text{obj}} = (1 - \text{cost}_P) - \pi(t_{\text{reward}}) \tag{4}$$

$$\langle R(T)\rangle = \alpha_R \times R_{\text{obj}}(T) + (1 - \alpha_R) \times R_{\text{obj}}(T-1) \tag{5}$$

in which $T$ is the trial index. In all presented simulations, $W_N = 0.25$. A filtered average cost, $R$, was computed as before[36] with $\alpha_R = 0.75$ and used in the update equation for changing network weights through the learning rule described below. For all constants a range of values were tried with qualitatively similar results. The performance objective was defined by cost$_P$, for which $\Delta t$ is the latency to collected reward after it is available. The network stability cost (cost$_N$) penalizes high-frequency oscillatory dynamics that can emerge in some (but not all) simulations. Such oscillations are inconsistent with observed dynamics of neural activity so far.

**Identifying properties of RNN required for optimal performance.** To examine what properties of the RNN were required for optimal performance, we scanned through thousands of simulated network configurations (random initializations of $W_{ij}$) and ranked those networks according to their mean cost ($R_{\text{obj}}$) when run through the behaviour plant for 50 trials (an illustrative group of such simulations is shown in Extended Data Fig. 2). This analysis revealed a few key aspects of the RNN required for optimality. First, a preparatory policy that spans time from the detection of the cue through the delivery of water reward minimizes latency cost. Second, although optimal RNNs are relatively indifferent to some parameters (for example, $P_c$), they tend to require a coupling coefficient ($g$) $\geq 1.2$. This range of values for the coupling coefficient is known to determine the capacity of an RNN to develop preparatory dynamics[63]. Consistent with this interpretation, our findings showed that optimal policies were observed uniquely in RNNs with large leading eigenvalues (Extended Data Fig. 2; that is, long-time-constant dynamics[64]). These analyses define the optimal policy as one that requires preparatory dynamics of output unit activity that span the interval between the cue offset and reward delivery and further reveal that an RNN with long-timescale dynamics is required to realize such a policy. Intuitively: preparatory anticipatory behaviour, or 'conditioned responding', optimizes reward collection latency. If an agent is already licking when reward is delivered the latency to collect that reward is minimized.

**RNN initialization for simulations.** All mice tested in our experiments began training with no preparatory licking to cues and a long latency (about 1 s or more) to collect water rewards. This indicates that animal behaviour is consistent with an RNN initialization that has a policy $\pi(t) \approx 0$ for the entire trial. As noted above, there are many random initializations of the RNN that can produce clear preparatory behaviour and even optimal performance. Thus, we carried out large searches of RNN initializations (random matrices $W_{ij}$) and used only those that had approximately 0 average activity in the output unit. We used a variety of different initializations across the simulations reported (Fig. 1 and Extended Data Fig. 2) and indeed there can be substantial differences in the observed rate of convergence depending on initial conditions (as there are across mice as well). For simulations of individual differences

(Fig. 1j and Extended Data Fig. 2), distinct network initializations were chosen (as described above), and paired comparisons were made for the control initialization and an initialization in which the weights of the inputs from the reward to the internal RNN units were tripled.

**Learning rules.** Below we articulate how each aspect of the model acronym, ACTR (adaptive rate cost of performance to REINFORCE), is reflected in the learning rule that governs updates to the RNN. The connections between the variant of node perturbation used here and REINFORCE[21] has been discussed in detail previously[36]. There are two key classes of weight changes governed by distinct learning rules within the ACTR model. First, we will discuss the learning that governs changes in the 'internal' weights of the RNN ($W_{ij}$). The idea of the rule is to use perturbations (1–10 Hz rate of perturbations in each unit; simulations reported used 3 Hz) to drive fluctuations in activity and corresponding changes in the output unit that could improve or degrade performance. To solve the temporal credit assignment problem, we used eligibility traces similar to those described previously[36]. One difference here was that the eligibility trace decayed exponentially with a time constant of 500 ms and it was unclear whether decay was a feature of prior work. The eligibility trace ($e$) for a given connection $i,j$ could be changed at any time point by computing a nonlinear function ($S$) of the product of the derivative in the input from the $i$th unit ($x_i$) and the output rate of the $j$th unit ($r_j$) in the RNN according to the equation (in dlRNN_engine.m)

$$e_{i,j}(t) = e_{i,j}(t-1) + \phi[r_j(t-1) \times (x_i(t) - \langle x_i \rangle)] \tag{6}$$

As noted in ref. [36], the function $S$ need only be a signed, nonlinear function. Similarly, in our simulations we also found that a range of functions could all be used. Typically, we used either $\phi(y) = y^3$ or $\phi(y) = |y| \times y$, and simulations presented were generally the latter, which runs more rapidly.

The change in a connection weight ($W_{ij}$) in the RNN in the original formulation[36] is then computed as the product of the eligibility trace and the change in PE scaled by a learning rate parameter. Our implementation kept this core aspect of the computation, but several critical updates were made and will be described. First, as the eligibility trace is believed to be 'read out' into a plastic change in the synapse by a phasic burst of dopamine firing[58], we chose to evaluate the eligibility at the time of the computed burst of dopamine activity estimated from the activity of the parallel feedback unit (see below for further details). Again, models that do not use this convention can also converge, but in general converge worse than and less similarly to observed mice. The update equation is thus (for example, line 330 in dlRNN-train_learnDA.m)

$$W_{i,j}(T) = W_{i,j}(T-1) + \beta_{DA} \times \eta_S \times e_{i,j}(t_{DA}) \times (R_{obj}(T) - \langle R(T) \rangle) \tag{7}$$

in which $\eta_S$ is the baseline learning rate parameter and is generally used in the range $5 \times 10^{-4} \pm 1 \times 10^{-3}$ and $\beta_{DA}$ is the 'adaptive rate' parameter that is a nonlinear function (sigmoid) of the sum of the derivative of the policy at the time of reward plus the magnitude of the reactive response component plus a tonic activity component, $T$ ($T = 1$ except in Extended Data Fig. 2 where noted and $\phi$ is a sigmoid function mapping inputs from {0,10} to {0,3} with parameters: $\sigma = 1.25$, $\mu = 7$) (for example, line 259 in dlRNN-train_learnDA.m):

$$\beta_{DA} = T + \phi(\Delta\pi(t_{reward}) + S_{i,reward}) \tag{8}$$

As noted in the description of the behavioural data described in Fig. 1, it is clear that animal behaviour exhibits learning of both preparatory behavioural responses to the cue as well as reactive learning that reduces reaction times between sensory input (either cues or rewards) and motor outputs. This is particularly prominent in early training during which a marked decrease in reward collection latency occurs even in the absence of particularly large changes in the preparatory component of behaviour. We interpreted this reactive component as a 'direct' sensorimotor transformation consistent with the treatment of reaction times in the literature[65], and thus reactive learning updates weights between sensory inputs and the output unit (one specific element of the RNN indexed as 'o' below). This reactive learning was also updated according to PEs. In particular, the difference between $R_{obj}(T)$ and the activity of the output unit at the time of reward delivery. For the cue, updates were proportional to the difference between the derivative in the output unit activity at the cue and the PE at the reward delivery. These rates were also scaled by the same $\beta_{DA}$ adaptive learning rate parameter (for example, line 346 in dlRNN-train_learnDA.m):

$$W_{trans,o}(T) = W_{trans,o}(T-1) + \beta_{DA} \times \eta_S \times (R_{obj}(T) - \pi(t_{reward})) \tag{9}$$

in which $\eta_l$ is the baseline reactive learning rate and typical values were about 0.02 in presented simulations (again a range of different initializations were tested).

We compared acquisition learning in the complete ACTR model to observed mouse behaviour using a variety of approaches. We scanned about two orders of magnitude for two critical parameters $\eta_l$ and $\eta_W$. We also aimed to sample the model across a range of initializations that approximately covered the range of learning curves exhibited by control mice. To scan this space, we followed the following procedure. We initialized 500–1,000 networks with random internal weights and initial sensory input weights (as described above). As no mice that we observed initially exhibited sustained licking, we selected six network initializations with preparatory policies approximately constant and 0. For these 6 net initializations, we ran 24 simulations with 4 conditions for each initialization. Specifically, we simulated input vectors with initial weights $S = [0.1, 0.125, 0.15, 0.175]$ and baseline learning rates $\eta_l = [2, 2.25, 2.5, 2.75] \times 8 \times 10^{-3}$. Representative curves of these simulations are shown in Fig. 1j.

**Visualizing the objective surface.** To visualize the objective surface that governs learning, we scanned a range of policies (combinations of reactive and preparatory components) passed through the behaviour plant. The range of reactive components covered was [0:1.1] and preparatory was [−0.25:1]. This range corresponded to the space of all possible policy outputs realizable by the ACTR network. For each pair of values, a policy was computed and passed through the behaviour plant 50 times to get an estimate of the mean performance cost. These simulations were then fitted using a third-order, two-dimensional polynomial (analogous to the procedure used for experimental data) and visualized as a three-dimensional surface.

In the case of experimental data, the full distribution of individual trial data points across all mice ($N = 7,200$ observations) was used to fit a third-order, two-dimensional polynomial (MATLAB; fit). Observed trajectories of preparatory versus reactive were superimposed on this surface by finding the nearest corresponding point on the fitted two-dimensional surface for the parametric preparatory and reactive trajectories. These data are presented in Fig. 1j.

**Simulating closed-loop stimulation of mDA experiments.** We sought to develop an experimental test of the model that was tractable (as opposed to inferring the unobserved policy for example). The experimenter in principle has access to real-time detection of licking during the cue–reward interval. In simulations, this also can easily be observed by monitoring the output of the behavioural plant. Thus, in the model we kept track of individual trials and the number of licks produced in the cue–reward interval. For analysis experiments (Fig. 5e), we tracked these trials and separately calculated the predicted dopamine responses depending on trial type classification (lick− vs lick+). For simulations in Fig. 5e, we ran simulations from the same

initialization in nine replicates (matched to the number of control mice) and error bars reflect the standard error.

To simulate calibrated stimulation of mDA neurons, we multiplied the adaptive rate parameter, $\boldsymbol{\beta}_{DA}$, by 2 on the appropriate trials For simulations reported in Fig. 5e, we used three conditions: control, stimLick− and stimLick+. For each of these three conditions, we ran 9 simulations (3 different initializations, 3 replicates) for 27 total learning simulations (800 trials). This choice was an attempt to estimate the expected experimental variance as trial classification scheme is an imperfect estimate of underlying policy.

**Pseudocode summary of model.** Here we provide a description of how the model functions in pseudocode to complement the graphical diagrams in the main figures and the discursive descriptions of individual elements that are used below.

Initialize trial to $T = 0$
Initialize ACTR with $W(0)$, $\mathcal{S}_{rew}(T)$, $\mathcal{S}_{cue}(T)$
repeat
 Run RNN simulation engine for trial $T$
 Compute plant input $\boldsymbol{\pi}(T) = O(T) + \mathcal{S}(T)$
 Compute lick output $L(t) = \text{Plant}(\boldsymbol{\pi}(T))$
 Compute latency to collect reward $t_{collect} \leftarrow \text{find } L(t) > t_{reward}$
 Compute $\text{cost}(T) = 1 - \exp(-\Delta t/500)$
 Evaluate eligibility trace at collection $e \leftarrow e_{i,j}(t_{collect})$
 Compute $\boldsymbol{\beta}_{DA} = 1 + \boldsymbol{\phi}(\Delta\boldsymbol{\pi}(t_{reward})) + \mathcal{S}_{rew}$
 Compute $R_{obj}(T) = 1 - (1 - \exp(-\Delta t/500)) - O(T, t_{reward} - 1)$
 Estimate objective gradient $PE = R_{obj}(T) - \langle R(T) \rangle$
 Compute update $\Delta W = -\eta_J \times e \times PE \times \boldsymbol{\beta}_{DA}$
 Update $W(T+1) \leftarrow W(T) + \Delta W$
 Update $\mathcal{S}_{reward}(T+1) \leftarrow \mathcal{S}_{rew}(T) + \eta_{\mathcal{S}} \times R_{obj}(T) \times \boldsymbol{\beta}_{DA}$
 Update $\mathcal{S}_{cue}(T+1) \leftarrow \mathcal{S}_{cue}(T) + \eta_{\mathcal{S}} \times R_{obj}(T) \times \boldsymbol{\beta}_{DA}$
Until $T == 800$

in which $T$ is the current trial and $t$ is time within a trial, $W$ is the RNN connection weight matrix, $\mathcal{S}$ is the sensory input strength, $O$ is the RNN output, $\pi$ is the behavioural policy, $\Delta t = t_{collect} - t_{reward}$, $\boldsymbol{\phi}$ is the nonlinear (sigmoid) transform, $\langle R(T) \rangle$ is the running mean PE, $\eta_J$ is the baseline learning rate for $W$ and $\eta_{\mathcal{S}}$ is the baseline learning rate for input $\mathcal{S}$.

**ACTR model variants.** In Fig. 1k, we consider three model variants equivalent to dopamine signalling PEs, dopamine depletion and loss of phasic dopamine activity—all manipulations that have been published in the literature. To accomplish these simulations, we: changed $\boldsymbol{\beta}_{DA}$ to equal PE; changed $\boldsymbol{\beta}_{DA}$ offset to 0.1 from 1; and changed $\boldsymbol{\beta}_{DA}$ to equal 1 and removed the adaptive term.

In Figs. 3 and 5, calibrated stimulation was modelled as setting $\boldsymbol{\beta}_{DA}$ to double the maximal possible magnitude of $\boldsymbol{\beta}_{DA}$ under normal learning. In Figs. 3c–e and 5i, we modelled uncalibrated dopamine stimulation as setting PE = +1 in addition to the calibrated stimulation effect.

**TD learning model.** To model a standard TD value learning model we reimplemented a previously published model that spanned a range of model parameterizations from ref. [66].

**Policy learning model equivalent to the low-parameter TD learning model.** The ACTR model that we articulate seeks to provide a plausible mechanistic account of naive trace conditioning learning using: RNNs; a biologically plausible synaptic plasticity rule; conceptually accurate circuit organization of mDA neurons; a 'plant' to control realistic behaviour; and multiple components of processing of sensory cues and rewards. However, to facilitate formal comparison between value learning and direct policy learning models, we sought to develop a simplified model that captures a key aspect of ACTR (the specific gradient it uses) and allows for explicit comparison against existing

value learning models with the same number of free parameters. To model a low-parameter (as compared to ACTR) policy learning equivalent of the TD value learning model from ref. [67], we used the same core structure, basis function representation and free parameters. However, rather than using an RPE (value gradient) for updating, we follow previous work[32] and consider a direct policy learning version in which a policy gradient is used for updates as originally described in ref. [21] and equivalent in terms of the effective gradient to the ACTR implementation. First, we consider the latency to collect reward rather than the reward value per se as used in TD models. The latency to collect reward is a monotonic function of the underlying policy such that increased policy leads to increased anticipatory licking as a reduction in the collection latency (Fig. 1). Typically one uses a nonlinearity that saturates towards the limits 0,1. For simplicity, we choose a soft nonlinearity (half-Gaussian) for convenience of the simple policy gradient that results. Regardless of the scaling parameter of the Gaussian (sigma), the derivative of the log of the policy is then proportional to $1 - p_t$, in which $p_t$ is the policy on trial $t$ (subject to scaling by a constant proportional to sigma that is subsumed into a learning rate term in the update equation). According to the REINFORCE algorithm family[21], we have an update function proportional to $(r_{curr} - b) \times (1 - p_t)$, in which $r_{curr}$ is the current trial reward collection latency and $b$ is a local average of the latency calculated by $b = \boldsymbol{v} \times r_{curr} + (1 - \boldsymbol{v}) \times b$. Typical values for $\boldsymbol{v}$ were 0.25 (although a range of different calculations for $b$, including $b = 0$, yield consistent results as noted previously[21]).

**Formal model comparison.** As in previous work[32], we sought to compare the relative likelihood of the observed data under the optimal parameterization of either the value learning (TD) model or the direct policy learning model. The data we aimed to evaluate were the frequency of anticipatory licking during the delay period over the first approximately 1,000 trials of naive learning for each mouse. We used a recent model formalization proposed to describe naive learning[67] and used grid search to find optimal values of the parameters $\lambda$, $\alpha$ and $\gamma$. To compute the probability of observing a given amount of anticipatory licking as a function of the value function or policy, respectively, we used a normal probability density (sigma = 1) centred on the predicted lick frequency (7 Hz × value or policy). Initial examination revealed that sigma = 1 minimized the LL for all models, but the trends were the same across a range of sigma. The −LL of a given parameterization of the model was computed as the negative sum of log probabilities over trials for all combinations of free parameters. We also computed the Akaike information criterion[68]—sum of ln(sum(residuals$^2$))—as preferred in some previous work[69]. The results were consistent and the number of free parameters was equivalent; thus, we primarily report −LL in the manuscript. For direct comparison, we took the minimum of the −LL for each model (that is, its optimal parameterization) and compared these minima across all animals. To examine the 'brittleness' of the model fit, we compare the median −LL across the entire grid search parameter space for each model.

**Estimating PEs from behavioural data.** First, we assume that on average the number of anticipatory licks is an unbiased estimate of the underlying policy (the core assumption of the low-parameter models described above). The latency to collect reward can be converted into a performance cost using the same equation (2) described for ACTR. The PE was then computed as in equation (4). A smoothed baseline estimate was calculated by smoothing PE with a 3-order, 41-trial-wide Savitzky–Golay filter and the baseline subtracted PE calculated analogous to equations (4) and (5).

## Histology

Mice were killed by anaesthetic overdose (isoflurane, >3%) and perfused with ice-cold phosphate-buffered saline, followed by paraformaldehyde (4% wt/vol in phosphate-buffered saline). Brains were post-fixed

for 2 h at 4 °C and then rinsed in saline. Whole brains were then sectioned (100 μm thickness) using a vibrating microtome (VT-1200, Leica Microsystems). Fibre tip positions were estimated by referencing standard mouse brain coordinates[70].

## Statistical analysis

Two-sample, unpaired comparisons were made using Wilcoxon's rank sum test (MATLAB rank sum); paired comparisons using Wilcoxon signed rank test (MATLAB signrank). Multiple comparisons with repeated measures were made using Friedman's test (MATLAB friedman). Comparisons between groups across training were made using two-way ANOVA (MATLAB anova2). Correlations were quantified using Pearson's correlation coefficient (MATLAB corr). Linear regression to estimate contribution of fibre position to variance in mDA reward signals was fitted using MATLAB fitlm. Polynomial regression used to fit objective surfaces were third order and (MATLAB fit). Errors are reported as s.e.m. Sample sizes ($n$) refer to biological, not technical, replicates. No statistical methods were used to predetermine sample size. Data visualizations were created in MATLAB or GraphPad Prism.

## Reporting summary

Further information on research design is available in the Nature Portfolio Reporting Summary linked to this article.

## Data availability

The data used to generate results supporting the findings of this study are available at https://janelia.figshare.com/account/collections/6369111 or https://doi.org/10.25378/janelia.c.6369111; the primary dataset is 10.25378/janelia.21816054.

## Code availability

All code relating to simulating the ACTR model and for a reader to explore both described parameterizations and explore a number of implemented, but unused in this manuscript, features is available at https://github.com/dudmanj/RNN_learnDA. Specific line numbers are provided within the code for a subset of critical computations in the model.

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

**Acknowledgements** We thank L. Grima and J. Keller and all the members of the Dudman laboratory, B. Mensh, A. Hermundstad, S. Romani and J. Day for project feedback; J. Arnold, Janelia Experimental Technologies and J. Keller for technical assistance; the Janelia GENIE team for designing and supplying viral vectors; and M. Copeland and B. Foster for histology. This work was supported by the Howard Hughes Medical Institute, where J.T.D. is a Senior Group Leader at the Janelia Research Campus.

**Author contributions** L.T.C. and J.T.D. conceived of the project and experiments, analysed data, and wrote the manuscript with input from S.E.L. J.T.D. built models with input from L.T.C. and ran simulations. S.E.L. carried out animal surgeries. L.T.C. collected all experimental data.

**Competing interests** The authors declare no competing interests.

**Additional information**
**Correspondence and requests for materials** should be addressed to Luke T. Coddington or Joshua T. Dudman.

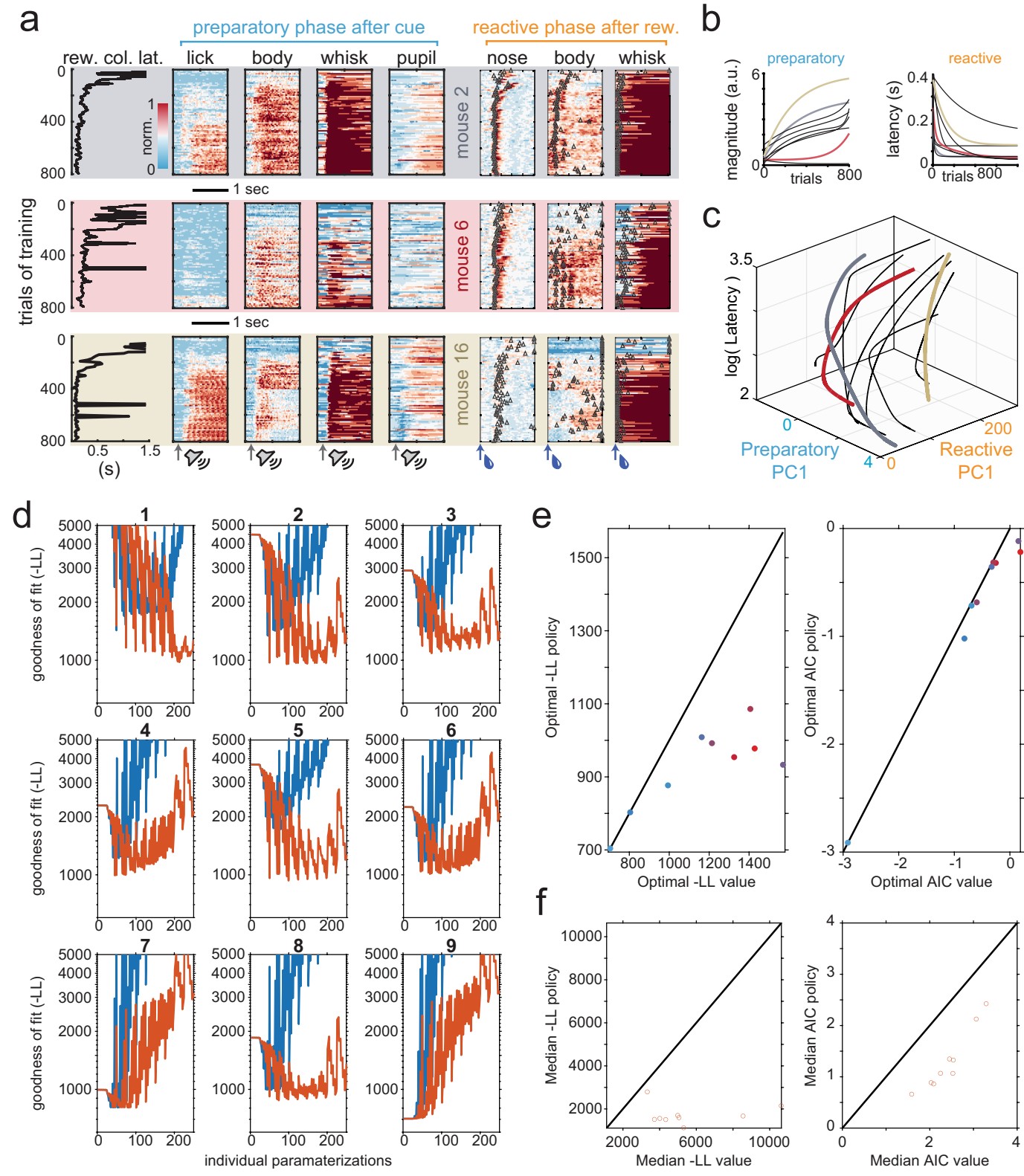

**Extended Data Fig. 1** | See next page for caption.

**Extended Data Fig. 1 | Comparison of low dimensional policy learning and value learning model fits to behavioral learning. a**) Reward collection latency (leftmost column) compared to normalized heat maps of preparatory measures of licking, body movement, whisking probability and pupil diameter (middle 4 columns), and reactive measures of nose motion, body movements, and whisking probability (right 3 columns, with mean first response following reward delivery indicated by black triangles) for standard trials in which a 0.5 s auditory cue (grey arrows at cue start) predicted 3 µL sweetened water reward (blue arrows), averaged in 10-trial bins across training. Each color-coded row corresponds to a separate example mouse, with the background color identifying the same mice in the low dimensional learning trajectories in Extended Data Fig. 1b, c. **b**) Abstract learning trajectories were described as exponential fits to the first principal. component of "preparatory" (left) and "reactive" (right) behavioral measurements. (example mice from panel A: red, yellow, grey; all other mice: thin black, total n = 9 mice). **c**) The relationship of inferred policy updates to reward collection performance visualized for each mouse as exponential fits to the first principal component of reactive and preparatory measurement variables (as shown in panel B), then plotted against the latency to collect reward (example mice from panel a: red, yellow, grey; all other mice: thin black, total n = 9 mice). **d**) Fits of value (blue) and policy (orange) learning models for each mouse across the space of possible parameterizations, measured as -log likelihood (smaller number is better fit) (n = 9 mice). **e**) Comparison of optimally parameterized policy and value models for each mouse, quantified by -log likelihood (left) or Aikake information criterion per trial (right). **f**) Comparison of median parameterized policy and value models for each mouse, quantified by -log likelihood (left) or Aikake information criterion per trial (right).

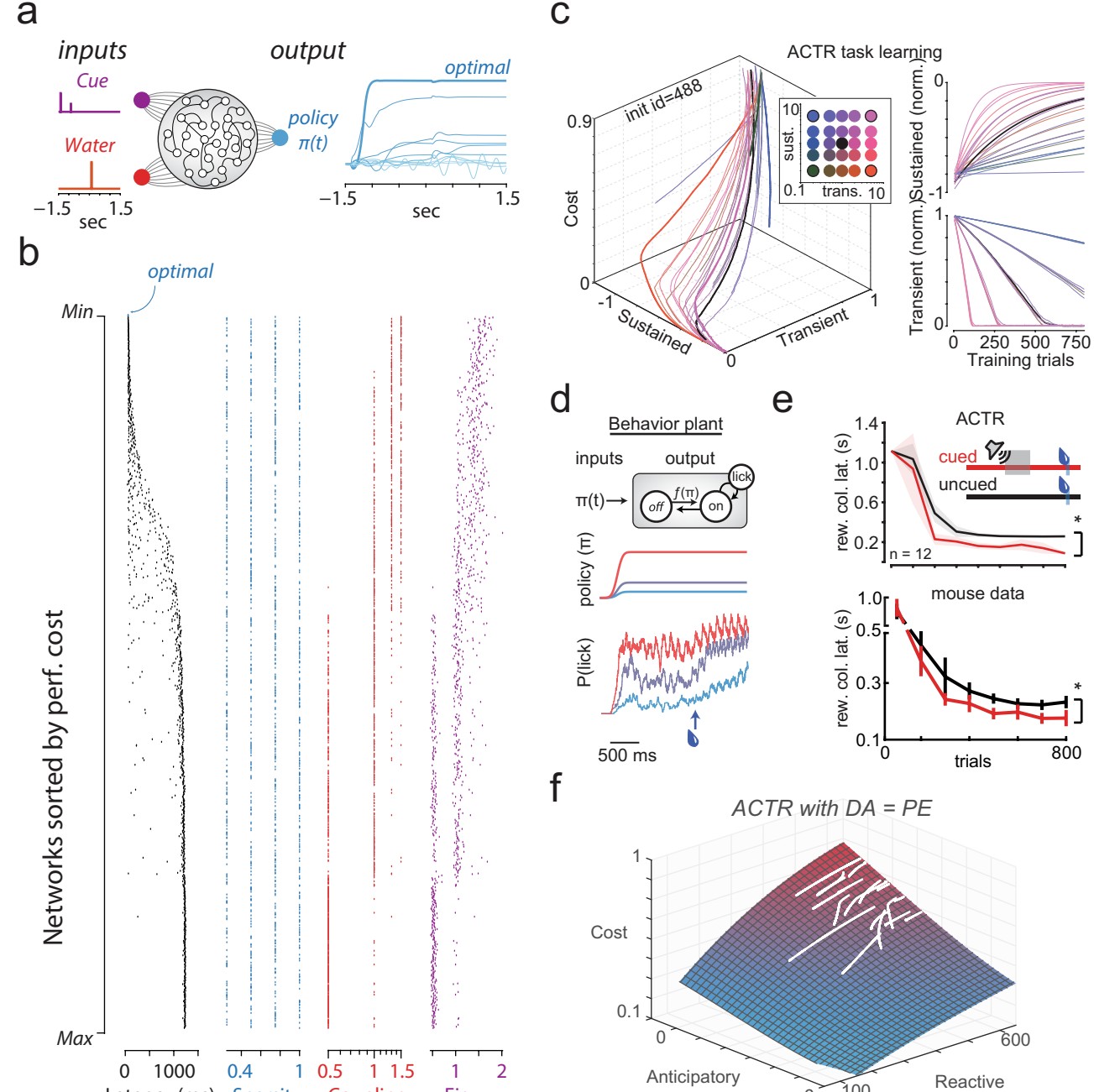

**Extended Data Fig. 2 | ACTR model details. a) top)** Schematic of ACTR policy recurrent neural network (RNN) and licking output from example different network initialization (right). **b)** Thousands of randomized initial network configurations ranked according to their performance cost (Fig. 1h; cost is a combination of latency and a network variance cost, see methods for details). Displayed are the latency to collect reward (black), network sparsity (blue), coupling coefficient (red), leading eigenvalue (purple). This analysis reveals a few key aspects. First, a sustained policy that spans time from the detection of the cue through the delivery of water reward is necessary to minimize latency cost. Second, while optimal RNNs are relatively indifferent to some parameters (sparsity of connectivity) they tend to require a strong coupling coefficient which is known to determine the capacity of a RNN to develop sustained dynamics[86], and thus optimal policies were observed uniquely in RNNs with large leading eigenvalues (i.e. long time constant dynamics[87]). These analyses indicate that there are realizable RNN configurations sufficient to produce an optimal policy, given an effective learning rule. **c)** Different ratios of sustained vs transient learning rates (inset color code) produced a range of trajectories similar to observed trajectories in individual mice (Fig. 1e). **d)** (top row) Licking

behavior was modeled as a two state ({off,on}) plant that emitted 7 Hz lick bouts from the 'on' state. Forward transition rate (off→on) was determined by a policy $\pi(t)$. Reverse transition rate (on→off) was a constant modulated by the presence of water. Bottom three rows illustrate example licking behavior produced by the plant for three different constant policies (red, purple, blue) before and after water reward delivery (vertical black line) for 100 repetitions of each policy. **e)** Learning quantified by a decrease in reward collection latency over training. As training progressed, a predictive cue led to faster reward collection (red) as compared to uncued probe trials (black) in both the ACTR model (top, trials 600–800, cued: 146 ± 21 ms, uncued: 205 ± 7 ms, two-tailed signed rank p = 0.01) and mouse data (bottom, cued: 176 ± 26 ms, uncued: 231 ± 23 ms, two-tailed signed rank p = 0.03). **f,** Cost surface (red = high, blue = low) overlaid with trajectories of individual initializations (white) as in Fig 1g, except having a constant learning rate and using the dopamine-like feedback unit to set the performance error (PE) instead of an adaptive learning rate (β), showing that the dopamine-like signal does not perform well as the PE in the ACTR learning rule. All error bars and bands represent +/− SEM around the mean.

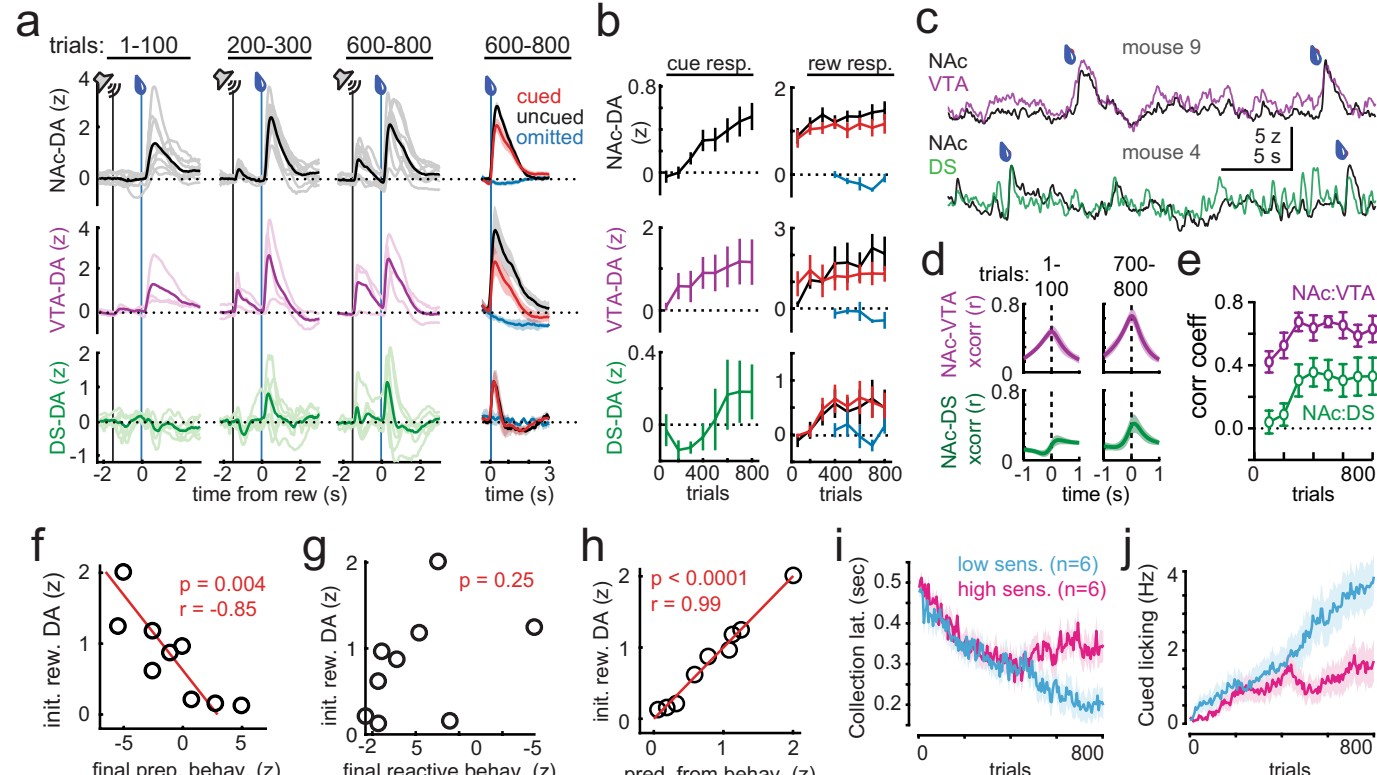

**Extended Data Fig. 3 | Dopamine signals across learning. a)** (left 3 columns) jRCaMP1b dopamine signals in the nucleus accumbens core (NAc, black, n = 9) and simultaneous recordings in the ventral tegmental area (VTA, purple, n = 3) and dorsal striatum (DS, green, n = 6), for cued reward trials in the trial bins indicated across training. (right) Reward or omission signals in NAc (top), VTA (middle), and DS (bottom) in trials 600–800, for cued (red), uncued (black), and cued but omitted (blue) trials. **b)** Mean signals from the data in panel (a) during the 1 s following cue delivery (left) and 2 s following reward delivery (right) across training for NAc (black, n = 9), VTA (purple, n = 3), and DS (green, n = 6). **c)** Example simultaneous recordings from NAc + VTA (top) and NAc + DS (bottom). **d)** Mean cross correlations for simultaneously measured NAc + VTA signals (top row, n = 3) and NAc + DS signals (bottom row, n = 6) in trials 1–100 (left) and trials 700–800 (right) within trial periods (1 s before cue to 3 s after reward). **e)** Peak cross correlation coefficients between NAc + VTA (purple, n = 3) and NAc + DS (green, n = 6) signal pairs across training, within "trial periods" defined as the time between cue start and 3 s after reward delivery. **f)** Correlation of initial NAc–DA reward signals with final combined preparatory behaviors (see Methods) (Pearson's r = −0.85, p = 0.004). **g)** No correlation of initial NAc–DA reward signals with final combined reactive behaviors (Pearson's p = 0.25). **h)** Correlation of initial NAc–DA reward signals predicted from behavior measures in trials 700–800 (see Methods) to observed initial NAc–DA reward signals (Pearson's r = 0.99, p < 0.0001). **i)** Reward collection latency for simulations with low (cyan) and high (magenta) initial reward-related sensory input. **j)** Preparatory cued licking for simulations with low (cyan) and high (magenta) initial reward-related sensory input. All error bars and bands represent +/− SEM around the mean.

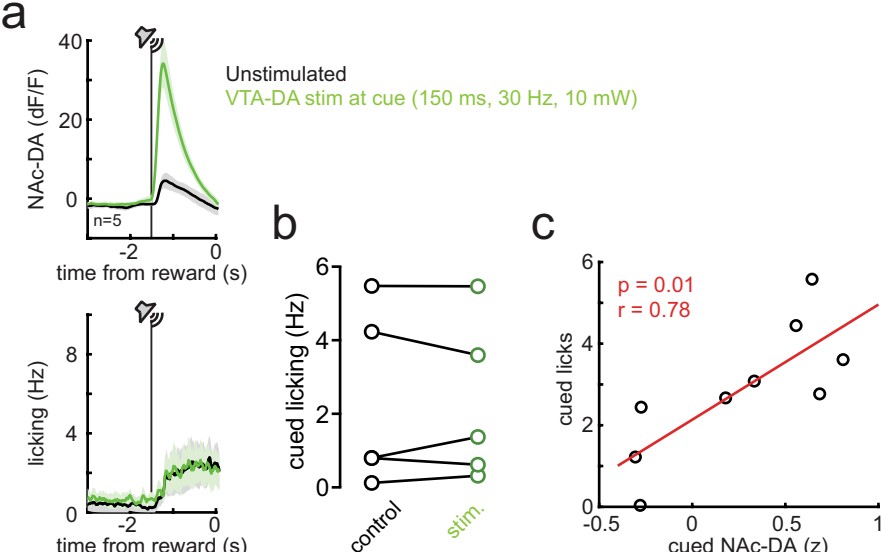

**Extended Data Fig. 4 | Augmenting mesolimbic cue signals does not affect cued behavior. a**) To test for a causal connection between the size of mesolimbic dopamine cue responses and cued behavior, in a new session after regular training was complete, we delivered large, uncalibrated VTA–DA stimulation on a random subset of cued reward trials (light green). Shown are NAc–DA responses (top) and licking (bottom) for this session. **b**) Quantification of cued preparatory licking during the delay period for unstimulated (black) vs stimulated (green) trials. **c**) Cued licking was correlated with the size of NAc–DA cue responses across animals (Pearson's r = 0.78, p = 0.01), even though manipulations did not support a causal relationship. All error bands represent +/− SEM around the mean.

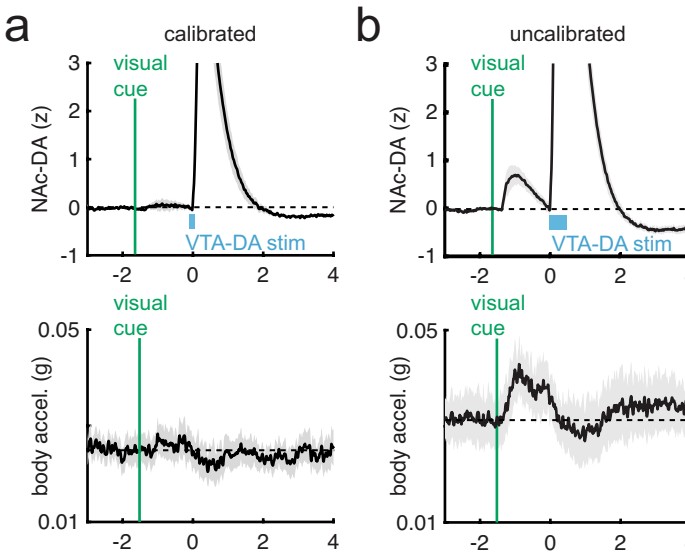

**Extended Data Fig. 5 | Movement correlates after learning induced by exogenous VTA-DA stimulation. a**) NAc–DA responses (top) and body movement measured as acceleration of the basket holding the mice (bottom) at the end of training in the paradigm in Fig 3a–e in which a visual cue predicted VTA–DA stimulation calibrated to measured uncued reward responses. **b**) Same as (a) except for larger, uncalibrated stimulation. All error bands represent +/– SEM around the mean.

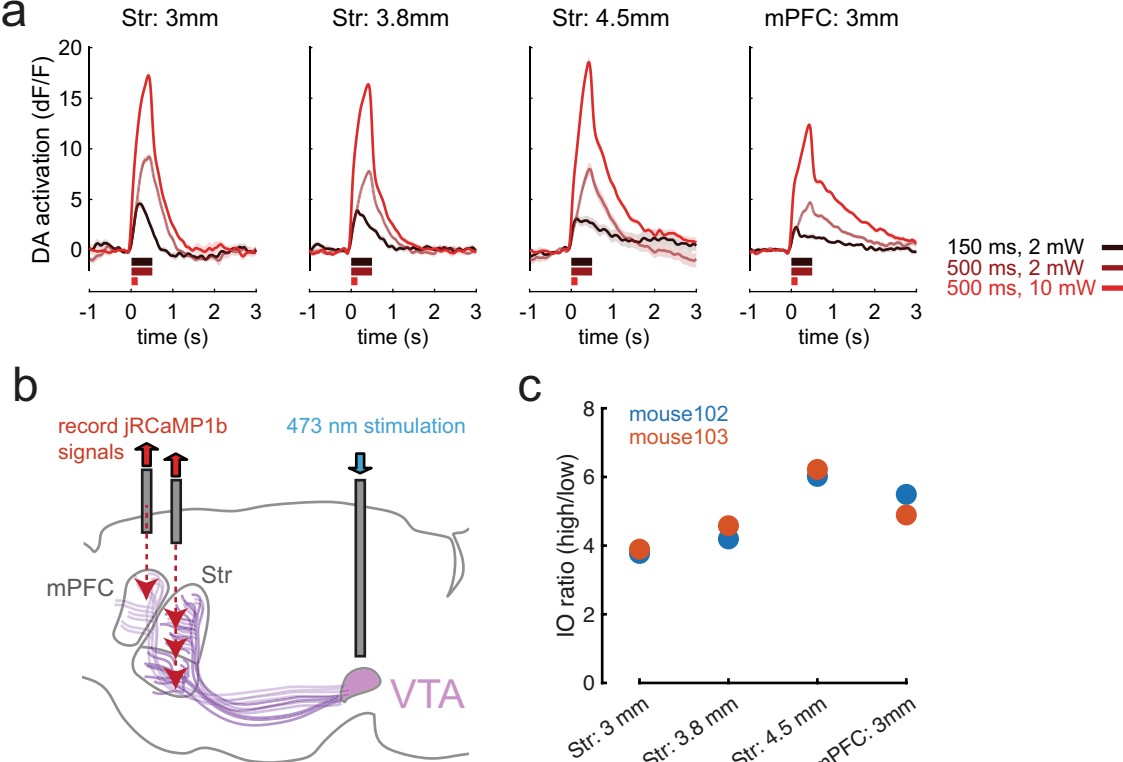

**Extended Data Fig. 6 | Exogenous stimulation produces similar input-output ratios across sampled mDA projection targets. a**) Dopamine responses measured through the same fiber for the low (black), medium (dark red), and high (bright red) stim parameters indicated at right, inserted either in the mPFC or at the indicated depth in the striatum. **b**) Schematic of experiment. **c**) Input-output ratio (response to high stim parameters divided by response to low stim parameters at each recording site in two separate mice (blue and orange).

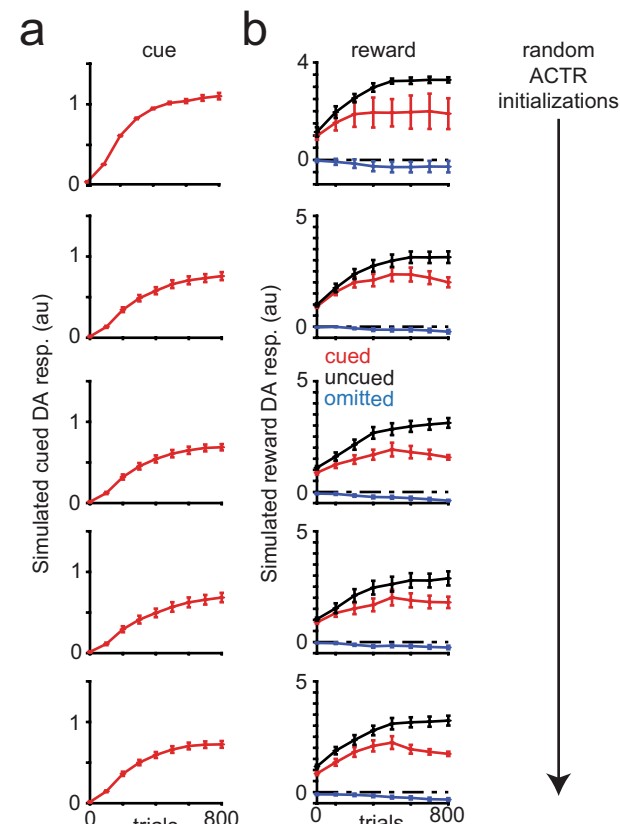

**Extended Data Fig. 7 | ACTR-modeled dopamine dynamics are robust to initialization conditions. a**) Predicted dopamine responses to cues for 5 random initializations of the ACTR model. **b**) Predicted dopamine responses to rewards over the same initializations as (a), comparing cued (red), uncued (black) and omitted (blue) reward trials (n = 5 simulations per initialization). RPE correlates emerged robustly across initializations. All error bars represent +/− SEM around the mean.

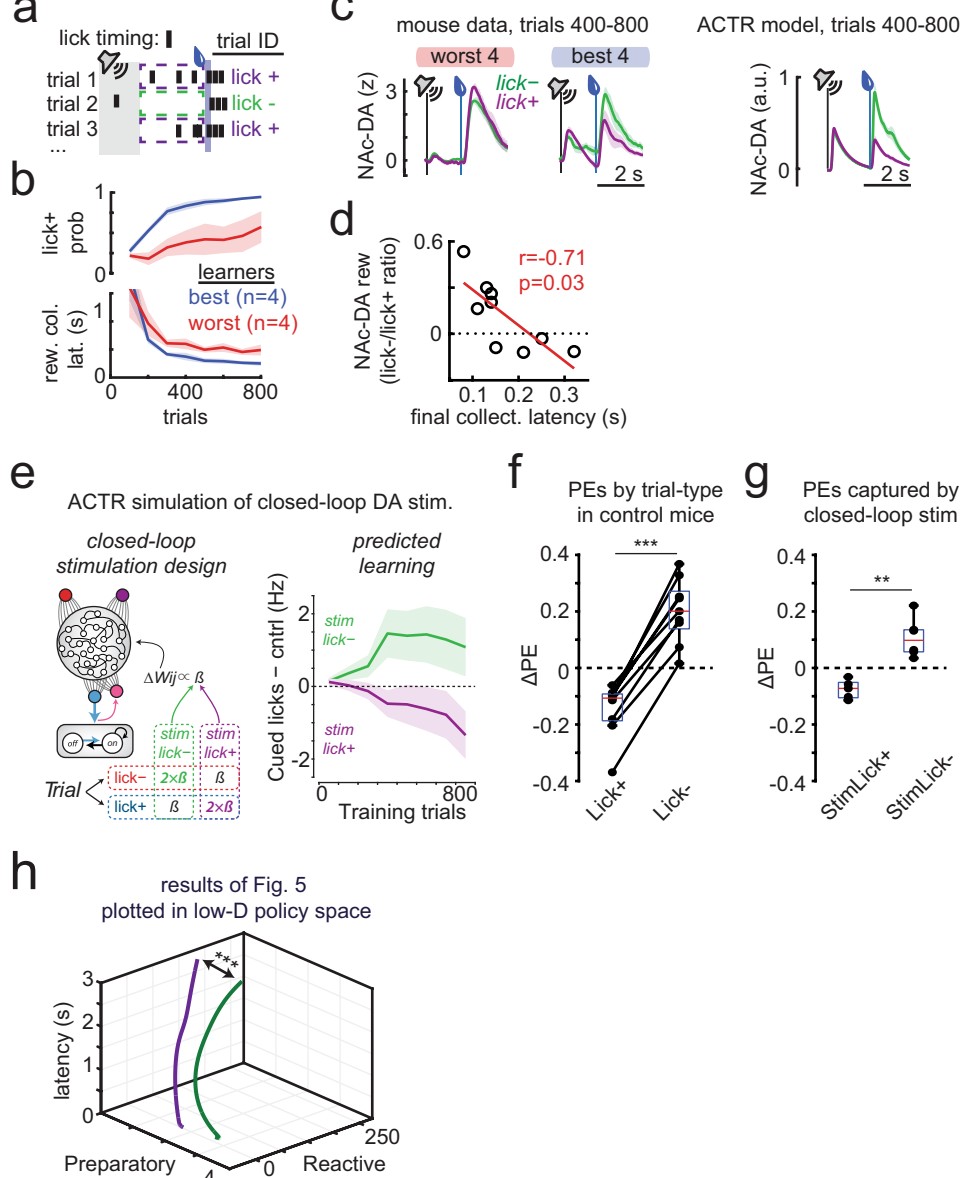

**Extended Data Fig. 8 | Relationship of NAc−DA signals and performance errors to the presence of preparatory licking. a**) Trial types defined as "lick+" trials (purple) with at least one lick during the delay between cue and reward and "lick-" trials (green) with no delay period licking. **b**) Percent of total trials that are "lick+" (top) and reward collection latency (bottom), for best 4 (green) and worst 4 (purple) performing mice, as determined by their reward collection latency in trials 700–800. **c**) NAc–DA signals in the second half of training (trials 400–800) lick- (green) and lick+ (purple) trials, for ACTR simulations (top) and worst (left) and best (right) top 4 performing mice in terms of reward collection latency in trials 700–800. **d**) The ratio of NAc–DA reward signals on lick- vs lick+ trials was correlated with the final reward collection latency (Pearson's r = −0.71, p = 0.03). **e**) Enhancing the mDA-like adaptive learning rate signal at reward on either lick- (green) or lick+ (purple) trials (schematic at left) in the ACTR model biases future licking behavior in opposite directions from the stimulation

contingency across training (right) for n = 9 initializations. In other words, enhancing mDA-like reward signals on trials with cued licking decreases cued licking in the future. **f**) The average change in performance error (ΔPE, the learning signal for preparatory learning in the ACTR model (Fig. 1i)) for each control mouse (n = 9) switched sign on lick+ or lick- trial types (two-tailed signed rank test p < 0.001). **g**) Average ΔPE on all stimulated trials in each mouse that received VTA–DA stimulation depending on whether they licked during the delay period preceding reward ("StimLick+", n = 5), or whether they did not lick during the delay period ("StimLick-", n = 6) (two-tailed signed rank p < 0.004 StimLick+ vs StimLick-). **h**) 3d learning trajectories as in Fig. 1e, comparing preparatory and reactive components of behavior to the latency to collect reward across training. All error bands represent +/− SEM around the mean. Box plots represent the median at their center bounded by the 25th and 75th percentile of the data, with whiskers to each extreme.

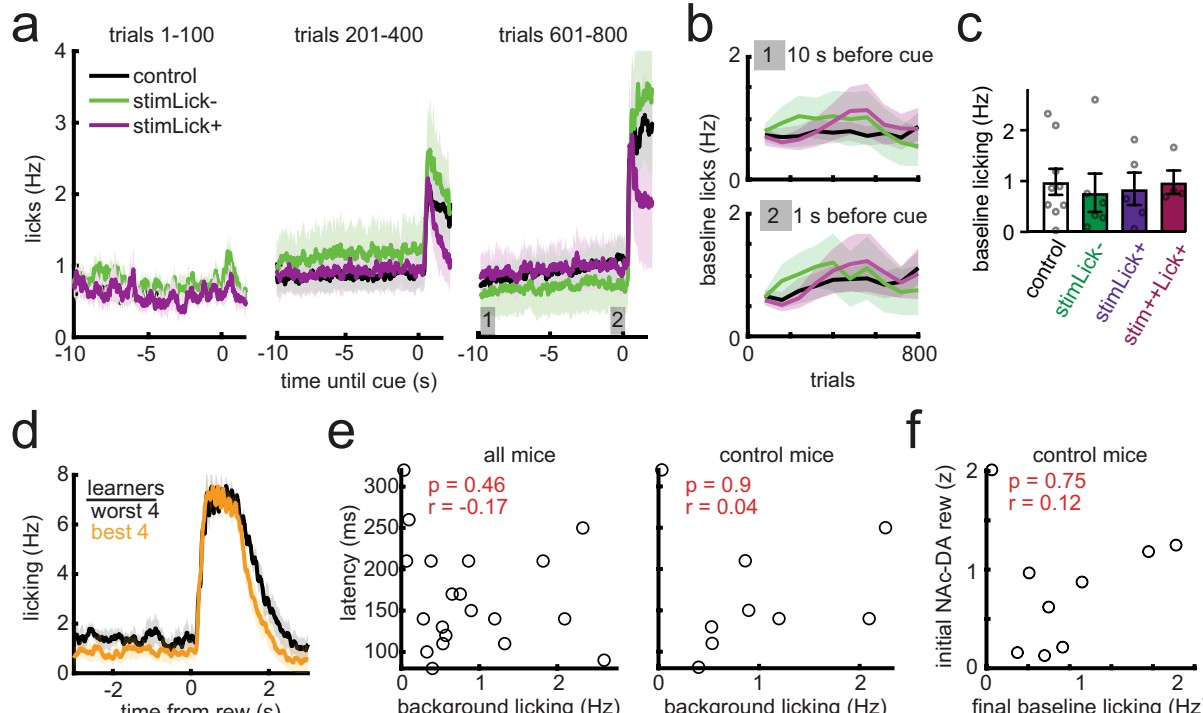

**Extended Data Fig. 9 | Baseline licking across training in all animals.**
**a)** Licking behavior showed in extended time before and after reward delivery to illustrate baseline intertrial licking behavior across the indicated training trials, for control (black), stimLick- (green), and stimLick+ (purple) mice. Grey numbered blocks indicate analysis epochs for panel (b). **b)** (top) Quantification of baseline licking at analysis epoch 1 (indicated at right of panel (a)) across training. (bottom) Quantification of baseline licking at analysis epoch 2. **c)** Mean baseline licking rate (final 300 trials of training) for all the experimental groups shown in Fig 1–6. No stimulation controls (white, n = 9), stimLick- (green, n = 6), stimLick+ (dark purple, n = 5), stim+Lick+ (light purple, n = 4). **d)** Licking behavior over the 3 s preceding uncued trials at the end of training

(trials 600–800) for the best 4 and worst 4 performing mice displayed an insignificant trend towards more baseline licking in bad learners. **e)** (left) No correlation between baseline licking and final latency to collect reward (a measure of learned performance) for all mice (Pearson's p = 0.46, n = 20). (right) No correlation between baseline licking and final latency to collect reward (a measure of learned performance) for only control mice that received no exogenous dopamine manipulations during training (corresponding to data from Fig 1–2, Pearson's p = 0.9, n = 9). **f)** No correlation between baseline licking and initial NAc–DA reward responses for control mice (Pearson's p = 0.75). All error bands represent +/- SEM around the mean.

# a

## outline of key points

reward collection optimization occurs on distinct timescales:
- reactive learning→large early gains
- preparatory learning→smaller later gains

Fig. 1a-f<br>Ext. Fig. 2

**naive trace conditioning explained as direct policy learning**

equivalent simple model comparison: policy learning provides better and more robust fits of behavior than value learning

Fig. 1g<br>Ext. Fig. 1

ACTR: an RNN that controls licking, learns to improve from the performance error of reward collection latency
- reactive learning modeled at sensory inputs to RNN
- preparatory learning modeled at internal weights of RNN

Fig 1h-j<br>Ext. Fig. 2

DA-like signal (see Fig 4) sets an adaptive learning rate in ACTR

Fig 1h-j

Individual differences in VTA→NAc-DA reward signals:
- negatively correlate with behavioral learning
- don't correlate with learned DA cue signals (strong correlation expected if DA reward signal teaches value)

Fig 2

**DA signals rate, not error, for policy updates**

supraphysiological, but not physiologically-calibrated, VTA-DA stimulation signals an error that updates cue encoding

Fig 3<br>Fig 6

mesolimbic DA (including RPE correlates) is well modeled as derivatives of sensory- and action-components of policy

Fig 4

calibrated VTA-DA manipulation in closed-loop with preparatory behavior distinguishes rate from error signaling

Fig 5<br>Fig 6

# b

## components of reinforcement learning

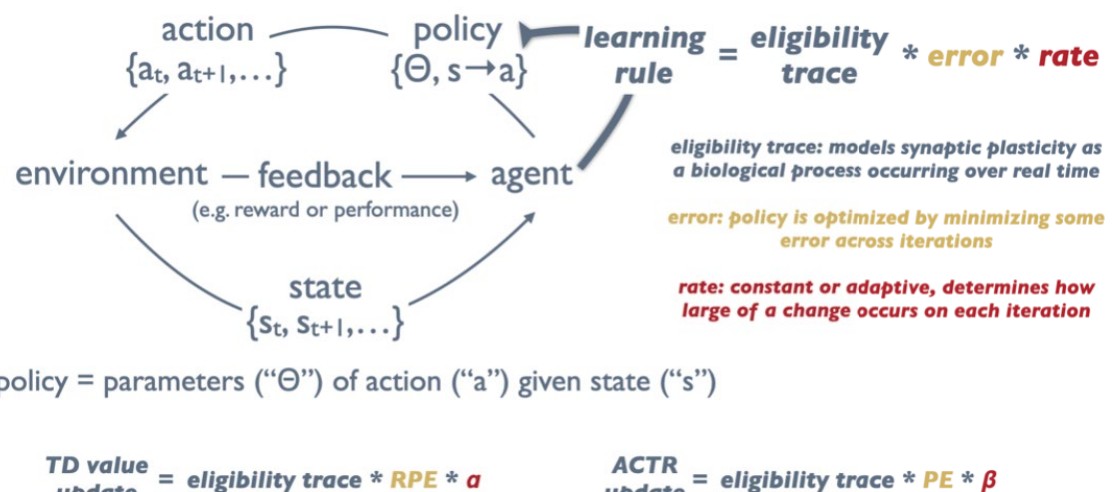

action $\{a_t, a_{t+1}, \ldots\}$ — policy $\{\Theta, s{\rightarrow}a\}$

learning rule = eligibility trace * error * rate

environment — feedback (e.g. reward or performance) → agent

state $\{s_t, s_{t+1}, \ldots\}$

eligibility trace: models synaptic plasticity as a biological process occurring over real time

error: policy is optimized by minimizing some error across iterations

rate: constant or adaptive, determines how large of a change occurs on each iteration

*policy = parameters ("Θ") of action ("a") given state ("s")

TD value update = eligibility trace * RPE * $\alpha$

ACTR update = eligibility trace * PE * $\beta$

**Extended Data Fig. 10 | Logic Outline. A**) Key points of the paper grouped by theme (left), with location in figures for primary supporting data (blue). **B**) In Reinforcement Learning, an agent learns iteratively from environmental feedback to improve a policy, which is a set of parameters (Θ) describing an action (a) that is performed given a state (s). In policy learning, the agent applies a learning rule.

# Reporting Summary

## Statistics

For all statistical analyses, confirm that the following items are present in the figure legend, table legend, main text, or Methods section.

| n/a | Confirmed | |
|---|---|---|
| ☐ | ☒ | The exact sample size (*n*) for each experimental group/condition, given as a discrete number and unit of measurement |
| ☐ | ☒ | A statement on whether measurements were taken from distinct samples or whether the same sample was measured repeatedly |
| ☐ | ☒ | The statistical test(s) used AND whether they are one- or two-sided *Only common tests should be described solely by name; describe more complex techniques in the Methods section.* |
| ☐ | ☒ | A description of all covariates tested |
| ☐ | ☒ | A description of any assumptions or corrections, such as tests of normality and adjustment for multiple comparisons |
| ☐ | ☒ | A full description of the statistical parameters including central tendency (e.g. means) or other basic estimates (e.g. regression coefficient) AND variation (e.g. standard deviation) or associated estimates of uncertainty (e.g. confidence intervals) |
| ☐ | ☒ | For null hypothesis testing, the test statistic (e.g. *F*, *t*, *r*) with confidence intervals, effect sizes, degrees of freedom and *P* value noted *Give P values as exact values whenever suitable.* |
| ☒ | ☐ | For Bayesian analysis, information on the choice of priors and Markov chain Monte Carlo settings |
| ☒ | ☐ | For hierarchical and complex designs, identification of the appropriate level for tests and full reporting of outcomes |
| ☐ | ☒ | Estimates of effect sizes (e.g. Cohen's *d*, Pearson's *r*), indicating how they were calculated |

*Our web collection on statistics for biologists contains articles on many of the points above.*

## Software and code

Policy information about availability of computer code

| Data collection | Task-related and fiber photometry data were collected using Cerebus Central Suite (v 7.0.6.0) software from Blackrock Microsystems. Video data was collected using Fly Capture 2 software from FLIR as well as custom written python code found at www.github.com/neurojak/pySpinCapture |
|---|---|
| Data analysis | Data were analyzed and visualized using custom code in MATLAB_R2019a (requiring the Signal Processing, Image Processing, Statistics and Machine Learning toolboxes) as well as GraphPad Prism 6. The data and custom code used to generate results supporting the findings of this study are within https://github.com/DudLab - including both modeling code (https://github.com/DudLab/RNN_learnDA) and analysis code (https://github.com/DudLab/TONIC). |

For manuscripts utilizing custom algorithms or software that are central to the research but not yet described in published literature, software must be made available to editors and reviewers. We strongly encourage code deposition in a community repository (e.g. GitHub). See the Nature Portfolio guidelines for submitting code & software for further information.

# Data

Policy information about availability of data

All manuscripts must include a data availability statement. This statement should provide the following information, where applicable:

- Accession codes, unique identifiers, or web links for publicly available datasets
- A description of any restrictions on data availability
- For clinical datasets or third party data, please ensure that the statement adheres to our policy

The data and custom code used to generate results supporting the findings of this study are within  https://github.com/DudLab - including both modeling code (https://github.com/DudLab/RNN_learnDA) and analysis code (https://github.com/DudLab/TONIC).

# Human research participants

Policy information about studies involving human research participants and Sex and Gender in Research.

| | |
|---|---|
| Reporting on sex and gender | *Use the terms sex (biological attribute) and gender (shaped by social and cultural circumstances) carefully in order to avoid confusing both terms. Indicate if findings apply to only one sex or gender; describe whether sex and gender were considered in study design whether sex and/or gender was determined based on self-reporting or assigned and methods used. Provide in the source data disaggregated sex and gender data where this information has been collected, and consent has been obtained for sharing of individual-level data; provide overall numbers in this Reporting Summary.  Please state if this information has not been collected. Report sex- and gender-based analyses where performed, justify reasons for lack of sex- and gender-based analysis.* |
| Population characteristics | *Describe the covariate-relevant population characteristics of the human research participants (e.g. age, genotypic information, past and current diagnosis and treatment categories). If you filled out the behavioural & social sciences study design questions and have nothing to add here, write "See above."* |
| Recruitment | *Describe how participants were recruited. Outline any potential self-selection bias or other biases that may be present and how these are likely to impact results.* |
| Ethics oversight | *Identify the organization(s) that approved the study protocol.* |

Note that full information on the approval of the study protocol must also be provided in the manuscript.

# Field-specific reporting

Please select the one below that is the best fit for your research. If you are not sure, read the appropriate sections before making your selection.

☒ Life sciences          ☐ Behavioural & social sciences          ☐ Ecological, evolutionary & environmental sciences

For a reference copy of the document with all sections, see nature.com/documents/nr-reporting-summary-flat.pdf

# Life sciences study design

All studies must disclose on these points even when the disclosure is negative.

| | |
|---|---|
| Sample size | No statistical methods were used to predetermine sample size. Sample sizes were determined in accordance with existing studies measuring and manipulating dopamine activity in awake behaving animals (e.g. Eshel et al and Uchida, Nature 2015; Saunders et al and Janak, Nat. Neuro 2018, Lee et al and Masminidis Nat. Neuro 2020). |
| Data exclusions | A small number of mice (n=4) were removed from the study after initial data was collected, due to poor signals. Histology determined that fibers were mistargeted and/or virus expression was insufficient in each of these mice. |
| Replication | We did not replicate the findings here in a new group of animals, however the dataset was compiled through serial small cohorts of animals randomly assigned to experimental groups, and cohorts displayed qualitatively similar behavioral and neural learning trajectories. Behavioral and neural learning trajectories also compare well to previously published similar experiments (https://doi.org/10.1038/s41593-018-0245-7). |
| Randomization | Experiments were done in repeated small cohorts (n=2-4) of mice across several months. Within each cohort, mice were randomly assigned to a group (control, stimLick-, stimLick+). |
| Blinding | Blinding during data collection was not possible as experimenter had to rely on group identity to determine the protocol for optogenetic manipulation. Experimenters were blind to group identity during the initial stages of analysis when analysis windows were determined and custom code was established to quantify fiber photometry signals and behavioral measurements. |

# Reporting for specific materials, systems and methods

We require information from authors about some types of materials, experimental systems and methods used in many studies. Here, indicate whether each material, system or method listed is relevant to your study. If you are not sure if a list item applies to your research, read the appropriate section before selecting a response.

## Materials & experimental systems

| n/a | Involved in the study |
|-----|------------------------|
| ☒ | ☐ Antibodies |
| ☒ | ☐ Eukaryotic cell lines |
| ☒ | ☐ Palaeontology and archaeology |
| ☐ | ☒ Animals and other organisms |
| ☒ | ☐ Clinical data |
| ☒ | ☐ Dual use research of concern |

## Methods

| n/a | Involved in the study |
|-----|------------------------|
| ☒ | ☐ ChIP-seq |
| ☒ | ☐ Flow cytometry |
| ☒ | ☐ MRI-based neuroimaging |

## Animals and other research organisms

Policy information about studies involving animals; ARRIVE guidelines recommended for reporting animal research, and Sex and Gender in Research

| | |
|---|---|
| Laboratory animals | We used male, adult DAT-cre x ai32 mice (12-40 weeks old) resulting from the cross of DAT-IREScre (The Jackson Laboratory stock 006660) and Ai32 (The Jackson Laboratory stock 012569) lines of mice. Mice were maintained under specific-pathogen-free conditions. Mice were housed on a free-standing, individually ventilated (~60 air changes hourly) rack (Allentown Inc, Allentown, NJ). The holding room was ventilated with 100% outside filtered air with >15 air changes hourly. Each ventilated cage (Allentown) was provided with corncob bedding (Shepard Specialty Papers, Milford, NJ), at least 8g of nesting material (Bed-r'Nest, The Andersons, Maumee, OH), and red Mouse Tunnel (Bio-Serv, Flemington, NJ). Mice were maintained on a 12:12-h (8am-8pm) light:dark cycle and recordings were done between 9am-3pm. The holding room temperature was maintained at 70±2°F with a relative humidity of 30% to 70%. Irradiated rodent laboratory chow (LabDiet 5053) was provided ad libitum. Following at least 4 days recovery from headcap implantation surgery, animals' water consumption was restricted to 1.2 mL per day for at least 3 days before training. Mice underwent daily health checks, and water restriction was eased if mice fell below 75% of their original body weight. |
| Wild animals | This study did not involve wild animals. |
| Reporting on sex | Only male animals were used for this study. |
| Field-collected samples | This study did not involve field-collected samples. |
| Ethics oversight | All procedures and animal handling were performed in strict accordance with a protocol (#19-190) approved by the Janelia Institutional Animal Care and Use Committee (IACUC, protocol 19-190) and consistent with the standards set forth by the Association for Assessment and Accreditation of Laboratory Animal Care (AALAC). |

Note that full information on the approval of the study protocol must also be provided in the manuscript.

