## [Peer Review File · Nature]

Manuscript Title: Mesolimbic dopamine adapts the rate of learning from action

Reviewer Comments & Author Rebuttals

Reviewer Reports on the Initial Version:

Referees' comments:

Referee #1 (Remarks to the Author):

This study provides a new perspective on dopamine that could in theory begin to drag the field away from some of the rather dogmatic viewpoints. One of the problems with most studies on the role of dopamine and RPEs in learning is that the 'expectations' are never actually measured, but are only inferred post-hoc based on the size of the dopamine response, which is circular. For this reason, I was excited to see proxy measures of expectation based on various behavioral markers. More importantly, while a host of models can capture overall learning rates, there has not been a principled way to capture the fluctuations in performance that individual animals exhibit. This study proposes a comprehensive theory to explain both overall learning rates as well as these fluctuations by arguing that the phasic dopamine signal adaptively changes the learning rate. The closed loop contingent stimulation results are quite compelling, as they are counter-intuitive yet fit nicely with model predictions. The theory also provides an explanation about how large stimulation during the reward epoch of certain trials can produce larger dopamine responses to a cue, consistent with value learning interpretations.

Obviously, any theory that challenges dogma will be greeted with apprehension. For this reason, I feel strongly that the authors need to do a better job of describing their theory and the evidence that potentially supports it. The following critiques are primarily made with this goal in mind. I found it difficult to grasp what the authors were trying to convey until near the end of the Results section. The theory remained murky until Fig 4 when I realized the main point was that dopamine could adaptively determine the learning rate. I felt this concept was not clearly laid out in the abstract, introduction or first paragraph of the results. I would strongly advise a major rewrite to emphasize this critical point. A brief synopsis of the basic theory as described on lines 187-201 followed by a simplified and clear description about how this differs from current models of dopamine function would be helpful. I also think that the paragraph beginning on line 404 in the discussion is a better summary of the results than the first paragraph of the current discussion. It is easy to model overall learning rates on such a simple task but what is much more difficult to model are the trial-by-trial fluctuations of individual animals. While I can see how this is possible within the framework proposed, the results are not really presented in the best possible light. The reader needs to see how dopamine transients track with the trial-by-trial changes in performance. In this regard, it is important to focus on the counter-intuitive cases, where for example the sustained lick policy was high but a negative performance error occurred, thereby pushing the system away from the sustained policy and causing the animal to subsequently exhibit less sustained licking. I understand that the idea is that dopamine changes the overall policy rather than evoking an immediate change in sustained behavior, but some type of trial-lag dependent approach might be possible.

Fig 3 is critical to the theory, but it was not well set up and I wasn't sure what part of it was relevant to the overall story. Also, I don't think I completely understand it. Fig 3A shows that all behavioral measures increase late, but in the right panels the blue and red lines overlap. Shouldn't the blue lines be much higher for Fig 3C to work? This is especially true in Fig 3a_{ii} where the blue lines are always slightly lower in the right panels. What am I missing?

The data presented in Fig 5c_i are also critical to the theory, but my concern here is that they could be interpreted in the traditional RPE framework. If the animal is doing well, it has a good idea of what the cue means and that it should start licking when it hears it. However, there would be trials when the cue does not register, due to perhaps a lapse of attention. As a result, the animal would not expect reward on these trials. A skeptic might therefore argue that the results in Fig 5C are simply due to the unexpected reward generating a larger dopamine 'RPE' on the lick- trials.

It is important to point out that the sound cue predicted the reward on 90% of trials but the solenoid was the most reliable predictor of reward on 100% of trials. Therefore, the purpose of the cue was different than in most RPE studies where it usually indicates the presence of the reward. In this study, the cue evoked sustained licking that in turn helped the animal collect the reward, but again, it was not the most reliable predictor of reward. It could therefore be argued that the unexpected differences in cue responses were due in part to this unique aspect of the experiment.

The data presented in Figure 5f-I are compelling and provide a nice way to square the results of this study with past studies. I do however think it would be beneficial to walk the reader through these results by contrasting the differences between the predictions of the two theories each step of the way.

I can see how the optogenetic stimulation could create a state of confusion in the animal. Based on what I garnered from the Methods, in some animals the stimulation was first added to the reward, then it was added to the cue, then it was added in place of the reward when a new cue (light) was introduced.

A minor issue relates to the implementation of the eligibility traces in the model. First, Shindou et al (2019) cannot be used as a basis for eligibility trace modulation as this was an in vitro study and had nothing to do with eligibility traces. Second and more importantly, the task is so simple and performance improves monotonically such that all past behaviors directly relate to all future states. Adding modulated eligibility traces to the model incorporates another free parameter but should not be necessary for the model to work.

Referee #2 (Remarks to the Author):

The study by Coddington, Lindo and Dudman uses behavioral, multi-site photometry, and optogenetic experiments, together with modeling, to investigate the role of midbrain dopamine neurons in policy learning through performance errors. The authors conclude using a trace conditioning pavlovian task the initial phasic response of dopamine to reward in the NAcc correlates with the probability of having "predictive" licking responses later in learning. Through a variety of analyses, models, and manipulations the authors take this as evidence that the phasic response of dopamine to reward is related to learning control policies rather than the value of cues (and

actions). There is much to like about this study, especially the ACTR learning rule and model for learning control policies. Certainly, the topic is important, and the role of midbrain dopamine neurons in signaling performance errors or has been understudied. However, there are a variety of experimental considerations that merit careful reflection before reaching such conclusions.

The choice of task in order to study if phasic dopamine activity regulates policy learning from performance errors is unexpected. One would expect that a task where the policy would determine the probability of getting reward would be more appropriate. A pavlovian task, in which licking represents a consummatory response, may not be ideal for studying the learning of control policies and performance prediction errors. Furthermore, the particular design of the task is peculiar. For example, there are unsignaled rewards delivered in 10% of the trials and there seems to be no CS-. This design may have biased what animals learned. Typically, in a pavlovian task, animals predict that a neutral cue predicts reward and anticipate the consummatory response to the neutral cue but not to other cues. Furthermore, they learn to not present consummatory responses in the absence of the cue. So, learning can be measured by increased licking after CS+, but not CS- or the intertrial interval. And, also by a decrease in the latency to collect reward to the CS+ as shown here, but not to CS-. Therefore, a CS- would really help the interpretation of the results. Apart from that, in the version of the task presented here it would be important to show the licking dynamics/rates right after cue presentation versus during the intertrial interval and right before cue presentation across learning. The fact that the latency to collect reward also decreases dramatically for the uncued trials signifies that the licking rate in the intertrial periods also increased dramatically. This could mean that in the current version of the task the cue is not the only or even main predictor of reward (although there is a small difference in the latency of cued versus uncued trials), and maybe the animals are also learning that they should lick when they are in this task, and that with the passage of time there is an increased probability of getting a reward. This, of course, could mean that animals with a strong dopamine response to reward early in training learn different strategies than animals with a low response to reward, namely how predictive of reward the cue is. Hence, one potential explanation for the results is that animals that do not have a strong phasic dopamine response to reward show more licking in general late in training both after the cue but also before the cue (before the cue late compared to before the cue early), and are just increasing expectation for reward as the intertrial interval goes by.

Another issue that deserves attention is the method used for measuring dopamine activity. The fact that the responses in NAcc and DS are different suggests that these are either different dopamine populations of dopamine neurons, or less likely, that the projections to these targets have different local modulation of axonal excitability. At any rate, at a time when the field is debating the heterogeneity of dopamine populations and how some could signal reward-prediction errors and others not, it is unclear what the authors are claiming here. Is the claim that different populations in VTA encode RPEs versus performance errors, or that most dopamine neurons in VTA do both, or that most VTA DA neurons are mainly involved in policy learning? It seems that fiber photometry is not the best method to dissect this.

The different optogenetic stimulation protocols yield interestingly results, and this is predicted by the ACTR model. The different results that the “calibrated” versus “uncalibrated” stimulation produce are important. However, there are also some important considerations here. In both

protocols, a 30Hz frequency of stimulation is used, which is way higher than the physiological firing of dopamine neurons. The authors should make sure that the dopamine neurons can follow the stimulation frequency of 30Hz. Additionally, besides different stimulation lengths being used, different optical powers are used. With higher optical powers many more dopamine neurons will be activated, and importantly in regions further away from the fiber, and hence the results could be because different subregions/subpopulations are being activated. Furthermore, in the current version of the task, the calibrated stimulations could lead to less predictive value of the cue and hence more sustained licking overall (not only after the cue), while strong activation of dopamine after the cue could increase the predictive value of the cue and hence result in less licking overall.

Finally, a small note. Although the topic of dopamine's role in policy learning through performance errors is much less explored than in RPE, there are a few important studies demonstrating the role of dopamine in signaling performance prediction errors, and hence the introduction and abstract seem to set up too much of a strawman. One of the most important studies in the topic is only reference number 38, and is first referenced in bundle with a lot of other studies (not mentioning that it claims a role for midbrain dopamine neurons in signaling performance prediction errors).

Referee #3 (Remarks to the Author):

I thought the goal of the paper was admirable -- quantitatively linking recorded dopamine signals to individual differences in learning and reinforcement learning models. However I found the specifics of the paper very confusing, both in terms of what scientific points were being made, and what evidence they had to support their arguments. For example, are they claiming that DA contributes to policy learning (which is a mainstream view from the actor-critic framework), or are they claiming DA does not contribute to value learning (which is hard to disprove, and I certainly do not think they show convincingly it does not)? More details below.

--

Regarding what points were being made: the paper puts great emphasis on the distinction between policy-only models, vs actor-critic models (the latter are commonly used to model the basal ganglia, and posit a role for dopamine not only in value learning but also in policy learning). The difference between policy-only vs actor-critic is subtle. After all, an RPE-like DA signal is still being learned in this version of REINFORCE, and that signal is still being used to modulate plasticity to guide reinforcement learning. I'm not clear if their goal is to provide more support for that view, and show it also applies in ventral striatum, or if they are trying to argue that dopamine does not also contribute to value learning. Whatever point they are trying to make, I think it is very subtle, and in addition, the latter would be a hard point to prove given that it's hard to prove a negative.

Making the big picture even more confusing is that the choice of behavior is pavlovian conditioning. Learning a “policy” is exactly equivalent to learning a value function in the context of pavlovian conditioning, since mice are expected to display pavlovian behaviors in proportion to the value function. Relatedly, the objective function that was used to train the network was based on time to collect reward once available, which can be considered a discounted value function. Therefore, the network seemed to be trained to calculate value, and then the derivative of that signal was used to produce the dopamine signal (RPE) - which all in all seem exceedingly similar to traditional actor-critic models.

I did think it was interesting that dopamine early on was so predictive of individual differences in behavior, and that their model recapitulated it. But I couldn't find an explanation of why their model produced that effect. Also, could they show the learning trajectory of the two groups of mice?

It seems they claim a key piece of evidence in support of their model is that dopamine causes activity away from the policy when they are in showing little licking behavior. Their explanation on line 267 of why DA enhancement on lick+ trials lead to less licking is as follow : “Such trials with a negative performance error are generally lick+ trials that occur relatively late in learning and have sustained licking that happens to terminate prior to reward delivery. Selectively enhancing learning rates on these trials with a negative performance error can have the effect of pushing the model away from a policy with sustained licking especially by reducing transient response components.” My understanding of this explanation is that despite calling these trials lick+, they are actually licking less immediately before the reward. Can they show this directly in both model and mice? If this explanation is correct, a value-based model would make the same prediction - if the behavior immediately before the reward is “not licking” that will strengthen the policy in a policy model and reinforce the action in a value learning model (since most recent behavior is most eligible for modification given eligibility trace). Ultimately, it is not at all convincing that they have found a regime that dissociates a value model from a policy model.

I found the mouse's learned behavior itself a bit confusing - The mice are getting much more efficient at licking w/ low latency over time, irrespective of the presence of the cue. Any learning about the cue is much more subtle and comes later. Is that because they are mostly learning about the solenoid click? Perhaps that's why the DA correlates in general are rather subtle for the cue, the behavior learning seems very much about the solenoid opening, and much less learning about the cue.

I found the Introduction to be very confusing. For example, in Line 26 about ‘performance errors’ versus RPEs - When I first read it I thought perhaps they were referring to RPEs with respect to actions vs stimuli. But eventually I thought they meant the discounted value function (since they used time to reward collection as their performance error). It would be helpful to provide a clear

definition of performance prediction errors. In addition, evidence for claim in Introduction that policy-only learning algorithms vs actor-critic models (that also include a policy) are better suited to understand individual differences is lacking

It was extremely difficult to read & follow the Methods section on their model. I suggest the authors should separate an explanation of the equations and concepts of the model versus details of the implementation, simulations etc. A particularly important issue is lack of clarity on how the DA signal is calculated, as the dopamine "beta" signal doesn't seem to have an equation in methods, and is explained in words very differently in Line 770 vs line 202 since one spot mentions a derivative and the other a sum. Not knowing how the DA signal was calculated really hampered my understanding of the model.

Is 4k & 4l referenced in Results? I searched for a reference to those figure panels and it did not come up. This seems like another critical omission, since those panels are the key predictions of the model that seem to really be the crux of the paper.

Why did lick- group lick more than lick+ in Fig 4H?

For figure 3.a.2, they should include the uncued reward response as well, to demonstrate if this an ROE or reward response.

Calibration experiment in Figure 6 is interesting, but I don't think it makes the effect of DA on producing a cue response at all uninteresting or unconvincing. They are calibrating to the DA signal immediately adjacent to the optical fiber in NAc. This should be conservative since not all dopamine neurons are in the immediate vicinity of the NAc fiber. Also there may be other neuromodulators etc that help enhance the effect of DA. They are still showing that DA signals are sufficient to enhance a cue response, consistent w/ learning about the value of the cue. Presumably inhibition would be sufficient to decrease that cue response. Also, does strong stimulation of DA in their model lead to a greater cue response? If not, why not?

Author Rebuttals to Initial Comments:

Referee #1 (Remarks to the Author; Replies):

This study provides a new perspective on dopamine that could in theory begin to drag the field away from some of the rather dogmatic viewpoints. One of the problems with most studies on the role of dopamine and RPEs in learning is that the 'expectations' are never actually measured, but are only inferred post-hoc based on the size of the dopamine response, which is circular. For this reason, I was excited to see proxy measures of expectation based on various behavioral markers. More importantly, while a host of models can capture overall learning rates, there has not been a principled way to capture the fluctuations in performance that individual animals exhibit. This study proposes a comprehensive theory to explain both overall learning rates as well as these fluctuations by arguing that the phasic dopamine signal adaptively changes the learning rate. The closed loop contingent stimulation results are quite compelling, as they are counter-intuitive yet fit nicely with model predictions. The theory also provides an explanation about how large stimulation during the reward epoch of certain trials can produce larger dopamine responses to a cue, consistent with value learning interpretations.

We thoroughly appreciate these comments and the summary offered by the reviewer. It is a very accurate and to our opinion insightful summary of our work placed nicely in the context of the field. Thank you. We have tried to ensure the revised (overhauled) manuscript helps convey these points more clearly than our initial submission.

Obviously, any theory that challenges dogma will be greeted with apprehension. For this reason, I feel strongly that the authors need to do a better job of describing their theory and the evidence that potentially supports it. The following critiques are primarily made with this goal in mind.

We thank the reviewer for these critiques and found them very helpful in clarifying the main themes of this work. Briefly, we have rearranged all figures to bring model and data closer together allowing detailed comparisons. We have expanded our modeling to compare both multiple variants of ACTR models and in a few key places comparison to a widely used alternative model in the field. Finally, we have done some new analyses inspired by reviewer comments and some additional experiments to further solidify our conclusions. Your guidance, and the time spent writing it out so clearly, have allowed us to strengthen the manuscript and it is greatly appreciated.

I found it difficult to grasp what the authors were trying to convey until near the end of the Results section. The theory remained murky until Fig 4 when I realized the main point was that dopamine could adaptively determine the learning rate. I felt this concept was not clearly laid out in the abstract, introduction or first paragraph of the results. I would strongly advise a major rewrite to emphasize this critical point.

On reflection we agreed very much with this assessment, and indeed performed a major rewrite. We have re-focused on the central importance of dopamine as a controller of adaptive learning rate and more clearly contrast our discovery with alternative functions that DA signals could have in signaling errors (which the data does not support). We now introduce the ACTR model earlier in the paper to focus on its predictive power for behavioral learning data and then detail how we consider multiple different ways in which DA might map onto functions in the model through close comparisons with data in all figures (Fig 1h-j; Fig 2f-g; Fig 3 c, e-f; Fig 4 g,i). This allows us to establish the unique and excellent fit between the adaptive rate term of the model and our observations of DA activity and function (via optogenetic manipulations).

A brief synopsis of the basic theory as described on lines 187-201 followed by a simplified and clear description about how this differs from current models of dopamine function would be helpful. I also think that the paragraph beginning on line 404 in the discussion is a better summary of the results than the first paragraph of the current discussion.

Thank you, we have taken this advice and used those passages to clarify the introduction and model description.

It is easy to model overall learning rates on such a simple task but what is much more difficult to model are the trial-by-trial fluctuations of individual animals. While I can see how this is possible within the framework proposed, the results are not really presented in the best possible light. The reader needs to see how dopamine transients track with the trial-by-trial changes in performance. In this regard, it is important to focus on the counter-intuitive cases, where for example the sustained lick policy was high but a negative performance error occurred, thereby pushing the

system away from the sustained policy and causing the animal to subsequently exhibit less sustained licking. ***I understand that the idea is that dopamine changes the overall policy rather than evoking an immediate change in sustained behavior, but some type of trial-lag dependent approach might be possible.***

We appreciate the thinking here and the reviewer is quite right about both the importance of this analysis and also its subtlety. We now include in Figure 1f-g analysis inspired by this point. Briefly, we show that (1) there are two distinguishable components of learning and (2) performance errors on the prior trial are predictive of the change in these components of the behavioral policy on the subsequent trial.

The reviewer is also correct that our model suggests that this effect should be modulated by dopamine transients on the prior trial for anticipatory components. We would note from the outset that this point is complicated because DA transients are also predicted to correlate, albeit in a bit of a complex way, with performance errors because of shared components in the computations. This is just a restatement of the well known limitations of using correlations to examine feedback in any closed-loop model. That being said, at right we show the requested analysis and note that it shows the expected effect consistent with the model. However, we also ran this analysis in the model and found it to be very noisy (even though we know the underlying computations) and indeed in the data the variance is substantial. Thus, we chose not to incorporate this in the main manuscript. It is unclear whether this variance is due to measurement noise inherent in estimating single trial dopamine responses or whether there are additional subtleties that will need to be discovered in future work (e.g. the effect of postsynaptic dopamine could integrate over trials in a way we do not yet fully understand due to long time constant biochemical processes; or the single trial heterogeneity in DA magnitude may reflect subtle variance in release location / cell type / etc that is not yet captured in our modeling predictions).

Fig 3 is critical to the theory, but it was not well set up and I wasn't sure what part of it was relevant to the overall story. Also, I don't think I completely understand it. Fig 3A shows that all behavioral measures increase late, but in the right panels the blue and red lines overlap. Shouldn't the blue lines be much higher for Fig 3C to work? This is especially true in Fig 3aii where the blue lines are always slightly lower in the right panels. What am I missing?

We regret that this figure was confusing as originally constructed. The current version (now Fig 2c-e) was simplified by (1) removing the individual examples, (2) zooming in on the remaining traces and highlighting the delay period in between cue and reward where we measured preparatory behavior (gray dashed boxes in middle column), and (3) plotting the quantified preparatory behaviors to the right of the traces. We also now show in the same figure the corresponding simulations from the ACTR model that explains the relationship between individual differences in DA signals and learned behavior (Fig 3 f-g). We hope this has clarified the presentation and we believe everything is internally consistent.

The data presented in Fig 5ci are also critical to the theory, but my concern here is that they could be interpreted in the traditional RPE framework. If the animal is doing well, it has a good idea of what the cue means and that it should start licking when it hears it. However, there would be trials when the cue does not register, due to perhaps a lapse of attention. As a result, the animal would not expect reward on these trials. A skeptic might therefore argue that the results in Fig 5C are simply due to the unexpected reward generating a larger dopamine 'RPE' on the lick- trials.

Both DA and licking responses to the cue indicate that the cue was perceived and affected behavior even on Lick- trials (shown below; in the submitted manuscript we only compare these DA signals with the equivalent model prediction). This is consistent with our view that behavior can be controlled differently across successive trials even without dramatic differences in the perception or predictive power of the cue.

However, we agree that while this particular data point is predicted by the ACTR model and thus confirmatory, it is also to some extent explainable within an RPE account of DA function. Certainly this will be the case for many individual pieces of evidence, and it emphasizes the importance of the manipulation experiments in Fig 3 and 4, which create situations in which error signaling and rate signaling accounts fundamentally differ. We have de-emphasized this piece of data, moving it into Ext Data Fig 6, as while it is an important analysis to highlight as it should be true under our policy learning account, it is not a piece of evidence that clearly distinguishes our interpretation from possible versions inspired by value learning.

It is important to point out that the sound cue predicted the reward on 90% of trials but the solenoid was the most reliable predictor of reward on 100% of trials. Therefore, the purpose of the cue was different than in most RPE studies where it usually indicates the presence of the reward. In this study, the cue evoked sustained licking that in turn helped the animal collect the reward, but again, it was not the most reliable predictor of reward. It could therefore be argued that the unexpected differences in cue responses were due in part to this unique aspect of the experiment.

The data presented in Figure 5f-I are compelling and provide a nice way to square the results of this study with past studies. I do however think it would be beneficial to walk the reader through these results by contrasting the differences between the predictions of the two theories each step of the way.

We thank the reviewer for this suggestion and we agree there is substantial benefit to comparing our model ACTR with existing value learning models. We have now clearly contrasted the major predictions in question (Fig 4g) with the actual closed-loop stimulation results (Fig 4c-e). We note that one might be concerned that the ACTR model is more complex in its construction than standard value learning models used in the field. On the one hand this is because our goal is much broader - we have attempted to explain in detail the learning of reward-related behavior as well as the time course of DA signaling. Thus, we have also now included comparisons where we compare DA activity functioning like the error term in ACTR with simulations in which DA activity functions as the adaptive rate term. While the performance error term in ACTR is meaningfully distinct from RPEs for value learning, they share the analogous function of directing learning towards or away from the agent's state on the current trial. We find that the evidence and data overwhelmingly support a role as an adaptive rate term as opposed to either type of error term. This modification, inspired by the reviewers comments, has importantly clarified the distinct predictions of the different models and better demonstrates the uniquely good fit of the ACTR model in which DA acts as the adaptive rate component.

I can see how the optogenetic stimulation could create a state of confusion in the animal.

Based on what I garnered from the Methods, in some animals the stimulation was first added to the reward, then it was added to the cue, then it was added in place of the reward when a new cue (light) was introduced.

In other work we have found the calibrated stimulation regime is imperceptible to the animals and does not interfere with reward seeking when combined with cues (W.-X. Pan, Coddington, and Dudman 2021). Given that and also the control in this paper that showed that stimulation at the cue did not immediately affect ongoing behavior (Ext Fig 3), we believe we have good controls strongly suggesting animals were not confused or directly disrupted in their behavior by the stimulation.

We do appreciate the concern over whether the order of experiments could confound their interpretation, and were cognizant of that danger during experiment design. The experiments were thus ordered to avoid confounds from previous stimulation paradigms—augmentation of DA cue responses (Ext. Data Fig 3) was only done for a single brief session and only interpreted with respect to immediate effects on behavior, and then the cued DA experiment used a novel visual cue to minimize confounds from any previous learning about an auditory cue (including any disruptions that might have occurred as a result of augmenting cued DA signals for that one brief session). In addition, as illustrated in Sup. Table 1, membership in the specific stim groups were counterbalanced across all these experiments so that if there were lingering effects from one paradigm they would not bias the results of the next experiment.

A minor issue relates to the implementation of the eligibility traces in the model. First, Shindou et al (2019) cannot be used as a basis for eligibility trace modulation as this was an in vitro study and had nothing to do

with eligibility traces. Second and more importantly, the task is so simple and performance improves monotonically such that all past behaviors directly relate to all future states. Adding modulated eligibility traces to the model incorporates another free parameter but should not be necessary for the model to work.

This comment was very interesting. To be honest we thought about it in the converse. Initially we ran the simulations without a decaying eligibility trace (which works as well), but worried that was biologically implausible and thus would elicit concerns. So we introduced a decaying eligibility trace to ensure that it could work even with a decaying trace. Essentially the two views can be reconciled - a non-decaying trace is a choice of a parameter with a huge time constant whereas here we chose a reasonable time constant (half a second). In this sense there is no additional parameter. We certainly could report results either way, but there is long standing evidence that associated learning depends at least to a degree upon the interval between cue and reward and we presume that is reasonably well captured by a decaying eligibility trace. We can remove the inappropriate reference, however, and will keep the reference to (Izhikevich 2007). Below we show some example simulations with a relatively broad permutation of initializations +/- eligibility trace decay. There are possibly subtle differences beyond stochastic gradient effects, but it is not the case that the decay is necessary, we simply find it to more plausibly connect to biological signals for plasticity.

Referee #2 (Remarks to the Author):

The study by Coddington, Lindo and Dudman uses behavioral, multi-site photometry, and optogenetic experiments, together with modeling, to investigate the role of midbrain dopamine neurons in policy learning through performance errors. The authors conclude using a trace conditioning pavlovian task the initial phasic response of dopamine to reward in the NAcc correlates with the probability of having “predictive” licking responses later in learning. Through a variety of analyses, models, and manipulations the authors take this as evidence that the phasic response of dopamine to reward is related to learning control policies rather than the value of cues (and actions).

There is much to like about this study, especially the ACTR learning rule and model for learning control policies. Certainly, the topic is important, and the role of midbrain dopamine neurons in signaling performance errors has been understudied. However, there are a variety of experimental considerations that merit careful reflection before reaching such conclusions.

The choice of task in order to study if phasic dopamine activity regulates policy learning from performance errors is unexpected. One would expect that a task where the policy would determine the probability of getting reward would be more appropriate. A pavlovian task, in which licking represents a consummatory response, may not be ideal for studying the learning of control policies and performance prediction errors.

Indeed, the pavlovian task was chosen in part for comparison to the wealth of previous data due to its historical association with DA-dependent learning, and not completely because it is specifically designed for the study of performance error. However, one contribution of this work is the novel demonstration that learning in this well-trodden context is surprisingly well-explained as optimization of policy guided by performance error, including some new analyses in the current revised version (Fig 1g, j; Ext Fig 6e). It was not clear from previous accounts that a task with such simple states and contingencies should produce the rich variation in individual learning trajectory that we describe, and we find this to be an important illustration of how control policies and optimization of performance need to be modeled explicitly in order to account for individual differences in behavior.

Additionally, when and how to lick is, though subtle, nonetheless a complicated and actively learned and controlled process (Bollu et al. 2021; Gutierrez et al. 2006; Gong et al. 2020) and not a static consummatory response. Given this, we do find it ideal to study the control policy for licking in isolation from the complications of other purposive behaviors (e.g. navigation, interactions with manipulandum) and with a simple pavlovian contingency in which naive learning is practical to capture.

Furthermore, the particular design of the task is peculiar. For example, there are unsignaled rewards delivered in 10% of the trials and there seems to be no CS-. This design may have biased what animals learned. Typically, in a pavlovian task, animals predict that a neutral cue predicts reward and anticipate the consummatory response to the neutral cue but not to other cues. Furthermore, they learn to not present consummatory responses in the absence of the cue. So, learning can be measured by increased licking after CS+, but not CS- or the intertrial interval. And, also by a decrease in the latency to collect reward to the CS+ as shown here, but not to CS-. Therefore, a CS- would really help the interpretation of the results.

We thank the reviewer for suggesting analyses of baseline licking here and below, and we have followed up on this and it provides additional evidence for cue-specific learning. We see little to no modulation of licking during this period over the course of training and the extent of such ITI licking is uncorrelated with performance across mice (see final figure in our response to you) - unlike the strong correlation between cue-responsive licking and performance/learning.

In our opinion there are some important reasons not to utilize a CS- for this study. One important reason is that some very compelling recent work has provided strong evidence that a CS- induces a kind of suppressive learning with a separate time course from the appetitive learning to the CS+. Perhaps the nicest example of this comes from recent collaborative work from Rob Froemke and Peter Holland (Kuchibhotla et al. 2019). Although unpublished currently, we have similar results from our lab observing that learning to suppress false alarms happens with a distinct time course from learning on CS+ trials. Indeed, in some prior work mice fail to learn to suppress responses to CS- despite

monotonic learning of the CS+ (Lee et al. 2018). Subsequent work from that same group has focused on CS+ only paradigms as a relevant example (Lee et al. 2020).

These two components of learning are an additional and fascinating aspect to model going forward and we think our policy learning perspective has great potential to explain the different time courses of suppressive and appetitive learning; however, we felt that that was beyond the scope of the already substantial study herein and our focus on the specifically on the learning of the appetitive component.

Apart from that, in the version of the task presented here it would be important to show the licking dynamics/rates right after cue presentation versus during the intertrial interval and right before cue presentation across learning.

We thank the reviewer for the specific suggestions about licking analyses and we have compiled the data requested here and below in a new supplemental figure (Ext Data Fig 8). At right we show licking for control and stimulation groups at an extended time scale, across learning (10 seconds before and after reward is the largest scale that can be visualized without starting to include data from reward delivery on adjacent trials). As learning progresses, there is a trend towards a small ramp up in licking as the next trial approaches. This seems consistent with animals having a flat hazard function for predicting the next trial, but also clear knowledge about the fact that considerable time has elapsed since the previous trial. We now include this data in the manuscript as requested.

The fact that the latency to collect reward also decreases dramatically for the uncued trials signifies that the licking rate in the intertrial periods also increased dramatically.

Above, we do not find a “dramatic” increase in lick rate in the intertrial period that would explain the fact that animals on average learn to collect uncued rewards in ~0.3 seconds. Instead the entirety of the data presented in the manuscript is consistent with improvements in animals’ reactions to sensory evidence of reward delivery. We describe this in our current paper as a “reactive” component - namely, mice learn to use sensory evidence of reward delivery (whether that be a solenoid click or the smell or chemosensation of water in front of the face (Coddington and Dudman 2018; Galiñanes, Bonardi, and Huber 2018)) to rapidly initiate licking and collect reward. A naive mouse (as we show in figure 1) has relatively slow reaction time to the delivery of water reward, but this reaction time improves in a way we model as strengthening a sensorimotor pathway from sensory evidence of water reward -> initiation of licking. This is sufficient to explain the improvement in reward collection latency for uncued trials since the same evidence of reward availability is present in both trial types. Consistent with this we observe quite low levels of baseline or background licking. In contrast, in published work we have observed >2Hz, hazard-modulated baseline licking for fully uncued reward delivery after learning a cue (W.-X. Pan, Coddington, and Dudman 2021).

This could mean that in the current version of the task the cue is not the only or even main predictor of reward (although there is a small difference in the latency of cued versus uncued trials), and maybe the animals are also learning that they should lick when they are in this task, and that with the passage of time there is an increased probability of getting a reward. This, of course, could mean that animals with a strong dopamine response to reward early in training learn different strategies than animals with a low response to reward, namely how predictive of reward the cue is. **Hence, one potential explanation for the results is that animals that do not have a strong**

phasic dopamine response to reward show more licking in general late in training both after the cue but also before the cue (before the cue late compared to before the cue early), and are just increasing expectation for reward as the intertrial interval goes by.

We thank the reviewer for their guidance on evaluating the alternative interpretation they propose. We do not find a correlation between initial DA reward responses and baseline licking just before trial start. We are certainly in agreement that “animals with a strong dopamine response to reward early in training learn different strategies than animals with a low response to reward” - that is our conclusion as well, but the question is how to describe the difference in terms of learning mechanisms. Our argument from modeling and analysis is that this observation is accounted for as a different behavioral policy for preparatory licking. Specifically, high dopamine responses are associated with excessively strong reactive policies early that impair the slower development of the predictive, preparatory policy (due to a vanishing gradient as explained below and in the text on Fig 2 of the current manuscript). Our modeling nicely captures this and importantly captures this phenomenon in the context of a full model that has substantial additional explanatory power (the same model is used to capture all the other key observations in the manuscript).

We should clarify a few points here. First, we agree that the cue is not the only predictor of reward, only that it is the earliest predictor of reward. There is a rich tradition of study in the learning field that emphasizes that the time interval between trials is also a reward predictor that can be used to anticipate reward delivery (albeit with substantial timing uncertainty) via a hazard function (Gallistel and Gibbon 2000). We go to some lengths to degrade the information present in the intertrial interval by using an exponential interval distribution that produces a flat hazard function (Gallistel and Gibbon 2000; Kepecs et al. 2008; Coddington and Dudman 2018) limiting the amount of information the passage of time conveys about reward. This exponential distribution is a relatively standard approach, but we also highlight that the intertrial intervals we use are considerably longer than other comparable work (~25 s on average, compared to <10 s for work from the Uchida lab for instance), magnifying the usefulness of the cue in preparing for reward delivery. It is also of interest to note that results from the Schultz group (and others) did not use the now standard exponentially-distributed ITI in many classic papers (a short uniform distribution of 5-7 seconds was often used (Schultz, Apicella, and Ljungberg 1993)) and thus it is an important caveat to interpretation of those prior results for the reasons the reviewer nicely illustrates.

Next, while we agree there is a large improvement in overall reward collection latency at the beginning of training for all mice, the difference between uncued and cued is not small in a relative sense that it is demonstrably significant. Cued trials allow animals to halve their latency to collect reward. If, as is standard in many other studies, we only reported data from mice late in training or after extensive pre-shaping, then the cued vs uncued latency effect would dominate. But by showing how initially slow mice can be to collect water reward on the same plot we believe it gives the most holistic representation of the full learning process. Very importantly, we also highlight that our ACTR model formally demonstrates that the relatively small gain in performance achieved by preparatory licking (Ext Data Fig 1e) is sufficient to drive the learning that we observe in mice.

The reviewer wonders about an additional possibility of how these individual differences arise; proposing that mice with a large reward response fail to learn how predictive the reward cue is (and one that we also wondered about initially). One line of evidence against this is that the dopamine responses to cues are not different in the high DA response cohort relative to the low DA cohort. Typically, the DA response to cues is taken as a measure of the learned predictive properties of the cue (Schultz 2015) and thus not

consistent with mice failing to learn the predictive cue. Second, we note that our analysis in Figure 2c (shown at right) is absolute lick frequency (not normalized). One can see that the high DA cohort still does elicit a clear uptick in lick frequency relative to baseline after the cue. This increase in preparatory licking is learned, but the learning is smaller and saturates at a low level associated with less good performance. The ACTR model captures this effect quite well. While there is a statistically insignificant trend towards an offset in baseline licking in the higher DA response subset apparent in the graph, when we examined the mice there was no correlation between background licking rate and reward collection latency ($p=0.9$, $r=0.04$) or initial DA responses ($p=0.75$, $r=0.12$) in contrast to the very clear and robustly significant correlation between DA transients and final reward collection latency ($p=0.008$, $r=0.81$; Fig. 2e). Thus, we do not believe these data are consistent with the proposal that high DA reward response is associated with a different strategy with respect to background licking rate.

Finally, we do think it is valuable to note that the reviewer's proposed interpretation is on some levels not so different from the explanation we argue for in the paper. We do indeed propose that mice fail to effectively use the cue to elicit sustained, preparatory behavioral policies. As we note this can happen because of the somewhat complicated interplay between reactive and preparatory components of behavioral policies and key properties of policy learning and not because mice used a totally distinct strategy (i.e. baseline licking). In policy learning agents are attempting to follow the noisy gradient of performance as they adjust their policies. A key problem that arises for complex policies (ours isn't extremely complex, just a reactive and sustained component, but that is enough complexity) is something referred to as "vanishing gradients". If an agent learns to react very fast to the presence of reward and learns that too soon before they have started to develop a predictive, preparatory policy they have a bit of a problem. The marginal improvement in performance becomes very small for incremental increases in preparatory licking. As a result the learning rule cannot pick up on the improvement of an incremental change in preparatory policy over the noisiness of the licking plant and the good reactive component. Thus, the preparatory policy doesn't 'see' much of a gradient (hence the term 'vanishing gradient') and this component of learning stalls out (although the fast, reactive components to cues and reward are unaffected - explaining the normal DA cue response in the context of our model). While this computational language is different from the language chosen by the reviewer, we find it advantageous because it describes the mechanisms of the ACTR model which fits a broad set of data and is consistent with the dual components of learning observed clearly from behavioral data (Fig. 1a-g). Nonetheless, we believe the descriptions are not so far apart conceptually.

Another issue that deserves attention is the method used for measuring dopamine activity. The fact that the responses in NAcc and DS are different suggests that these are either different dopamine populations of dopamine neurons, or less likely, that the projections to these targets have different local modulation of axonal excitability. At any rate, at a time when the field is debating the heterogeneity of dopamine populations and how some could signal reward-prediction errors and others not, it is unclear what the authors are claiming here. Is the claim that different populations in VTA encode RPEs versus performance errors, or that most dopamine neurons in VTA do both, or that most VTA DA neurons are mainly involved in policy learning? It seems that fiber photometry is not the best method to dissect this.

Recent work has, like us, used fiber photometry measurements to argue for distinct signaling in dopamine release / axonal activity relative to the mean somatic spiking activity in VTA (Mohebi et al. 2019). So we don't think photometry per se is a problem unique to our study in this respect. It is clear from anatomy in mice that dorsal striatum and nucleus accumbens are innervated by largely non-overlapping and distinct sets of dopamine neurons primarily located in the substantia nigra and ventral tegmental area, respectively (W. X. Pan, Mao, and Dudman 2010). Unlike prior work which only compares across different cohorts of animals for somatic and axonal physiology (Mohebi et al. 2019), we used simultaneous imaging at somata (VTA) and axon terminals (NAcc) and found highly correlated and consistent activity with clearly lower correlations as compared to DS (image at right). In addition we found similar evidence for RPE correlates in both NAc and

VTA signals, so we are certainly not claiming different encoding of information at soma and axon terminals. We are also distinctly not claiming that DA signals performance errors, as readers might have regrettably inferred from our focus in the previous manuscript on the idea that policy learning is driven by performance error. We have overhauled the current manuscript to clearly reflect what we are claiming: that the mesolimbic DA pathway (VTA→NAc) transmits a signal that modulates the learning rate on policy updates.

Fiber photometry offers the current most accurate and tractable path to measuring DA activity in projection regions where DA exerts its function over learning. We have produced the largest dataset to date of single cell recordings of optogenetically identified dopamine neurons (Coddington and Dudman 2018). In that prior work we found that there were not categorically distinct subsets of dopamine neurons within VTA or SNc, but there were differences in the mean distribution of properties across those areas. As a result we thought it was a good approach to use photometry to address our specific questions here and allow for the potential to see distinct axonal and somatic signaling (however, we found high correlation). In addition to the reasons above, there are also clear merits to knowing the average dopamine response in a downstream region since it is often thought to be a paracrine signaling mechanism and postsynaptic neurons within an area the size of a fiber might well be sensitive to the average dopamine output and not be able to detect subtle differences across individual DA neurons. This again is a fascinating question that will require future work in the field.

We develop a computational model, ACTR, that is a biologically plausible model of a class of RL models known as direct policy learning. In Fig 1 we show that this model is the first model that can capture core behavioral details of learning as well as capturing the diversity of individual differences in learning (a point nicely summarized by Rev 1). We then use recording of DA neuron activity in VTA/NAc and find that it is highly consistent with a specific component of the ACTR model - namely, 'Beta' a feedback unit that is responsible for adaptively modulating the learning rate. The activity of this unit Beta encodes something that can be correlated to inferred reward prediction errors, but there is no reward prediction error computed in ACTR. We then show using both further model-based analysis and a key optogenetic manipulation experiment that VTA-NAc dopamine activity is most consistent with modulating learning rate (i.e. Beta) and is not consistent with the predictions of models in which dopamine neurons function like signed prediction errors (either the performance errors computed in ACTR nor reward prediction errors computed in published value learning models that we implemented; Fig. 4 in particular). Finally, we show that when we use very strong, uncalibrated stimulation of dopamine neurons we get some effects that are more similar to prediction errors, but still not quite the same (clarified better in revised Fig 3d-e).

Thus, our study arrives at the conclusion that the role of VTA dopamine neurons is more consistent with models in which dopamine adaptively modulates learning rate, and not fully consistent with predictions of a signed reward prediction error.

We finally note that this is not to say that there are no predictions of RPE-based models that appear consistent with prior experiments - indeed many do and we replicate some of those here. But, the reason for that is because there are experiments that fail to distinguish between an effect on learning rate and an effect on signed errors. When we design a key, novel experiment that can distinguish rate from error we find evidence strongly in favor of the rate interpretation (Fig. 4). Moreover, we show that a model (ACTR) that does not compute reward prediction errors, can nonetheless fully and with unprecedented accuracy capture the correlates of dopamine neuron activity over initial learning (Fig. 3f; including the observations that are known to be inconsistent with RPE predictions (Coddington and Dudman 2019)).

The different optogenetic stimulation protocols yield interesting results, and this is predicted by the ACTR model. The different results that the “calibrated” versus “uncalibrated” stimulation produce are important. However, there are also some important considerations here. In both protocols, a 30Hz frequency of stimulation is used, which is way higher than the physiological firing of dopamine neurons. The authors should make sure that the dopamine neurons can follow the stimulation frequency of 30Hz.

We use a careful method we have developed to calibrate dopamine activity evoked by optogenetic stimulation and the observed dopamine responses to rewards (Coddington and Dudman 2021). We do not find 30Hz stim to be “way higher” than physiological firing. Many papers in mice (not just from our group) report peak dopamine activity at or above 30 Hz (W.-X. Pan, Coddington, and Dudman 2021; W.-X. Pan, Brown, and Dudman 2013; Coddington and Dudman 2018; Cohen et al. 2012; Eshel et al. 2015). At right we show an example figure panel from our previous work showing 30 Hz endogenous response peak in physiology, a population of DA neurons following 30 Hz light stim, and the quality of calibration with respect to dopamine release in NAcc. We note that the first dopamine optotagging paper from Cohen and Uchida showed individual DA neurons following optogenetic activation up to 50 Hz (Cohen et al. 2012). This is also not only a feature of mouse dopamine neuron recordings, at right we show an example from primates (Pasquereau and Turner 2013) with ≥ 30 Hz peak responses to reward.

Additionally, besides different stimulation lengths being used, different optical powers are used. With higher optical powers many more dopamine neurons will be activated, and importantly in regions further away from the fiber, and hence the results could be because different subregions/subpopulations are being activated.

We appreciate this point, and we have now added a new control experiment to address this (Ext Data Fig 5). When we stimulate over the VTA at different strengths, the input-output relationship is similar across PFC, dorsal and ventral striatum. However, as we acknowledged in the original discussion we cannot rule out that increased release in regions other than the NAc contributes to the categorically different effects of large DA stimulation. This remains a very interesting area for further research that is motivated by one of our main findings—that there are dissociable effects between calibrated and large DA stimulations. As a final note, in previous work we showed a dissociation between calibrated and uncalibrated DA stimulation in SNc (Coddington and Dudman 2018). The uncalibrated stimulation in SNc could produce some direct movement effects. However, uncalibrated stimulation in VTA did not produce these movement effects (and again did not produce them in the current work) suggesting that uncalibrated stim maintains some anatomical/functional specificity. This provides some additional evidence that a change in spatial spread may not be the most parsimonious explanation of the dissociation between calibrated and uncalibrated stimulation in VTA. Nonetheless, as stated we acknowledge the impossibility of fully ruling out this alternative or some more complex interaction between spatial spread and duration and amplitude.

Furthermore, in the current version of the task, the calibrated stimulations could lead to less predictive value of the cue and hence more sustained licking overall (not only after the cue), while strong activation of dopamine after the cue could increase the predictive value of the cue and hence result in less licking overall.

It is not entirely clear to us how or why the reviewer hypothesizes that calibrated stimulation leads to less predictive value of the cue since a cue always preceded the calibrated stimulation. For example, we used a published value learning model (representative of the model class) to simulate dopamine stimulation as per standard interpretation in the literature. These models predict an increased predictive value of the cue (Fig. 4g). Although not shown we also implemented several other value learning models from the literature all of which make the same prediction. Moreover, in our work, and that of others, increasing the value of the reward (e.g. larger volume) increases the magnitude of dopamine reward responses and increases the predictive value of cues (Eshel et al. 2015).

We can also consider what the predictions would be *if* calibrated stimulation reduced the predictive value of the cue (independent of whether that is expected from existing models or what the mechanism underlying such an effect could). In that case when we couple calibrated stimulation to the lick+/- contingency we should see reduced preparatory licking and reduced DA responses to the cue in either case. However, as we describe in the paper (Fig

4c-d), calibrated stimulation triggered on trials that lack anticipatory licking (“lick-”) enhances DA encoding of cues and enhances preparatory licking in response to the cue (i.e. increases predictive value). Thus, calibrated stimulation does not *necessarily* reduce the predictive value of the cue.

In addition, we have done the requested analysis and analyzed baseline lick rate outside of the trial (Ext Data Fig 8, and shown below). Again, we find no increase in baseline licking as a consequence of calibrated stimulation and no difference between uncalibrated (stim++Lick+) and calibrated (stimLick+/-) on the baseline lick rate. We also find no relationship between the extent of learning across all mice and the amount of learned baseline licking.

Finally, a small note. Although the topic of dopamine’s role in policy learning through performance errors is much less explored than in RPE, there are a few important studies demonstrating the role of dopamine in signaling performance prediction errors, and hence the introduction and abstract seem to set up too much of a strawman. One of the most important studies in the topic is only reference number 38, and is first referenced in bundle with a lot of other studies (not mentioning that it claims a role for midbrain dopamine neurons in signaling performance prediction errors).

Thank you, this is a really important point and we put substantial effort into better describing our results in the context of this prior work as well as highlighting this important reference much earlier in the manuscript. In the revised manuscript we distinguish 3 types of computations that have been associated with dopamine function: (1) reward prediction error from value learning models (Eshel et al. 2015; Schultz 2015), (2) performance errors invoked in the context of an actor-critic model (Gadagkar et al. 2016; Chen et al. 2019), (3) adaptive rate modulation from a policy learning model (our proposal). Both 1 & 2 are signed error terms that are critical to guide the direction of change (towards or away) at each learning step. Our proposal, #3, which has close parallels in the machine learning / optimization / RNN literature (Bottou, Curtis, and Nocedal 2018; Kingma and Ba 2014; Sussillo and Abbott 2009), has been little if at all explored in neuroscience (although it has some similarities to Pearce Hall attentional modulation models as well as with other qualitative theories).

The key difference is that rates are unsigned quantities that determine what fraction of the error is captured on a given iteration’s update. Deciding when and how to adapt the learning rate turns out to use some related, but clearly distinct, computations to the calculation of errors. For example, some purely computational models make the learning rate directly proportional to the average size of errors (Sussillo and Abbott 2009) whereas other optimization methods have quite complex calculation of learning rate based upon a number of computed quantities (Kingma and Ba 2014). Our computation (described in Fig. 1) is somewhere in between these prior examples and draws upon previous insights that dopamine activity is a combination of 2 components (Coddington and Dudman 2018). This is the reason that the adaptive error rate term can have similar *correlates* to prediction/performance errors. However, the really key difference is the effect on learning.

Learning rates are sensitive to the sign of the error and thus exaggerating learning rates when errors are negative can, for example, impair learning by pushing the model away from that policy. On the other hand, selectively exaggerating learning rates when errors are positive can speed learning towards the improved or optimal policy. This

is the reason for adapting learning rates in the first place and can be very critical when errors are stochastic (as they are in direct policy learning). We thus show that closed-loop perturbation experiments can selectively distinguish a rate from an error effect (either prediction error or performance error) in Fig. 4. This is the key distinction between our current work and prior work and a critical component of why we argue for our conclusion (#3 above).

Referee #3 (Remarks to the Author):

I thought the goal of the paper was admirable -- quantitatively linking recorded dopamine signals to individual differences in learning and reinforcement learning models. However I found the specifics of the paper very confusing, both in terms of what scientific points were being made, and what evidence they had to support their arguments. **For example, are they claiming that DA contributes to policy learning (which is a mainstream view from the actor-critic framework), or are they claiming DA does not contribute to value learning** (which is hard to disprove, and I certainly do not think they show convincingly it does not)? More details below.

Regarding what points were being made: the paper puts great emphasis on the distinction between policy-only models, vs actor-critic models (the latter are commonly used to model the basal ganglia, and posit a role for dopamine not only in value learning but also in policy learning). **The difference between policy-only vs actor-critic is subtle.** After all, an RPE-like DA signal is still being learned in this version of REINFORCE, and that signal is still being used to modulate plasticity to guide reinforcement learning. I'm not clear if their goal is to provide more support for that view, and show it also applies in ventral striatum, or if they are trying to argue that dopamine does not also contribute to value learning. Whatever point they are trying to make, I think it is very subtle, and in addition, the latter would be a hard point to prove given that it's hard to prove a negative.

Thank you for this excellent critique, it has very much provided useful guidance as to how to clarify the novelty and distinctions in our current conclusions while also allowing us to continue to demonstrate how our model can nonetheless account for many phenomena of prior models. We understand two main criticisms from the above which are tricky to address until they are disentangled:

1) There is no substantial difference between policy learning and actor-critic.

We find the passage included at right (from a widely cited review of RL for robotics (Kober, Bagnell, and Peters 2013)) to be a clear and intuitive delineation of actor (policy learning), critic (value learning), and actor-critic methods. It clarifies that in actor-critic, learning is restricted to a critic that learns a value function. The RPEs that are computed from this value function are then used to evaluate policies that are explicitly maintained. Policy learning uses performance of the policy (e.g. in ACTR we use latency to collect reward) to evaluate policies. Thus we agree with the reviewer that actor-critic and actor-only share some important architecture (explicit maintenance of a policy with respect to action), but we (and the field of computational reinforcement learning (e.g. (Nachum et al. 2017; Konda and Tsitsiklis 2000))) do not find the distinctions too subtle to be meaningful, and we find the distinctions to be mechanistic, not semantic.

Policy-search methods are sometimes called *actor*-only methods; value-function methods are sometimes called *critic*-only methods. The idea of a critic is to first observe and estimate the performance of choosing controls on the system (i.e. the value function), then derive a policy based on the gained knowledge. In contrast, the actor directly tries to deduce the optimal policy. A set of algorithms called *actor-critic* methods attempt to incorporate the advantages of each: a policy is explicitly maintained, as is a value-function for the current policy. The value function (i.e. the critic) is not employed for action selection. Instead, it observes the performance of the actor and decides when the policy needs to be updated and which action should be preferred. The resulting update step features the local

When DA is proposed to participate in actor-critic learning in the basal ganglia, what is specifically being proposed is the same as for critic-only learning: that DA signals an RPE that updates a value function. Thus it is not "mainstream" to argue that dopamine participates in direct evaluation of policies rather than indirect evaluation through construction of value functions. Recent work from Yael Niv and colleagues (Bennett, Niv, and Langdon 2021) for example has clearly summarized that prior work had considered only relatively limited roles for dopamine in actor learning and had focused extensively on a primary role of dopamine in determining learning of the critic (i.e. value learning) (see also a prescient work from Loewenstein and colleagues (Mongillo, Shteingart, and Loewenstein 2014)). These well-written reviews from experts in learning theory make it abundantly clear that direct policy learning (which can be referred to as "learning in the actor", policy-gradient learning, policy search) represents a clear alternative to the mainstream view in which the goal of learning is to create a value function that predicts reward from states.

We find it helpful to separate the above discussion from the associated criticism below.

2) It is unclear which component of learning DA is signaling in our account (and if it is clear, it is not novel)

Our paper and its conclusions constitute a substantial revision to the “reward prediction error hypothesis” of dopamine function. That hypothesis clearly exists and has been declared a success for a good and important reason: there is a well replicated set of observations that are consistent with the RPE hypothesis. We recognize the significance of revising this important insight and we also appreciate that it requires a possibly tricky combination: (a) account for the key results that support the RPE hypothesis, but also (b) distinguish our claims from this hypothesis.

In the revised manuscript we distinguish 3 types of computations that have been associated with dopamine function: (1) reward prediction error from value learning models e.g. (Eshel et al. 2015; Schultz 2015), (2) performance errors invoked in the context of an actor-critic model e.g. (Gadagkar et al. 2016; Chen et al. 2019), (3) adaptive rate modulation from a policy learning model (our proposal). Both 1 & 2 are signed error terms that are critical to guide the direction of change (towards or away) at each learning step. Our proposal, #3, has close parallels in the machine learning / optimization / RNN literature (Bottou, Curtis, and Nocedal 2018; Kingma and Ba 2014; Sussillo and Abbott 2009) but has been little if at all explored in neuroscience (although it has some similarities to Pearce Hall attentional modulation models as well as with other qualitative theories).

The key difference is that rates are unsigned quantities that determine what fraction of the error is captured on a given iteration’s update. Deciding when and how to adapt the learning rate turns out to use some related, but clearly distinct, computations to the calculation of errors. For example, some purely computational models make the learning rate directly proportional to the average size of errors (Sussillo and Abbott 2009) whereas other optimization methods have quite complex calculation of learning rate based upon a number of computed quantities (Kingma and Ba 2014). Our computation (described in Fig. 1) is somewhere in between these prior examples and draws upon previous insights that dopamine activity is a combination of 2 components (Coddington and Dudman 2018). This is the reason that the adaptive error rate term can have similar *correlates* to prediction/performance errors, even though something very distinct is being signaled to dopamine-recipient areas.

Inspired by the Reviewer’s insight we have tried to clarify the above issues in the current manuscript. It is true that our REINFORCE-based model has an error computation that is used to estimate the policy gradient. This error computation is, as the reviewer notes, “RPE-like” (e.g. it is a signed error). However, as we now show more clearly dopamine signaling recorded during our task and the effects of optogenetic dopamine stimulation are not consistent with dopamine playing this “RPE-like” error computation role. Rather dopamine activity is most consistent with the term beta in our model which is an adaptive learning rate term that is separate from the signed error term.

Finally, while it is important to report results that are inconsistent with current dogma, we would indeed like to avoid resting the success of this work on “proving a negative.” We have made a concerted effort instead to present an alternative formulation to value learning by RPE signaling, one that can explain dopaminergic correlates to inferred RPE signals while still explaining a collection of observations that are inconsistent with value learning.

Making the big picture even more confusing is that the choice of behavior is pavlovian conditioning. Learning a “policy” is exactly equivalent to learning a value function in the context of pavlovian conditioning, since mice are expected to display pavlovian behaviors in proportion to the value function.

This is an exceptionally important point made by the reviewer and we thank them for the clarity of this statement. This was in fact one inspiration for our choice of paradigm: ***in the context of value learning***, there should be no meaningful dissociations between the time course of the emergence of “pavlovian behaviors” and RPE correlates in dopamine activity - since they are both a function of the same underlying value function. To reference the previous discussion about “*actor-only*” vs. “*actor-critic*”, in this task, as we think the reviewer has pointed, one would not expect to be able to dissociate “*critic-only*” learning from “*actor-critic*” learning. However, policy learning (“*actor-only*”) instead implies substantial individual differences as the stochastic search process dominates the learning curves, and does not share the same constraint with respect to matching the emergence of DA signals. Indeed, we see a clear dissociation between the learning of putative value encoding by DA cue signals and the emergence of individual differences learned behavior (Fig 2). We show that there is excellent quantitative agreement between those sources of variance in policy learning and the variance we observe across individuals (Fig 1 and 2).

We can also give a more qualitative description that we believe is insightful. When Bellman first derived a solution to what we now call value learning his goal was to model the task with an auxiliary function (attach a value to environment states) rather than model the agent’s control policy itself (the agent’s state-action mapping) (Kober,

Bagnell, and Peters 2013). This was because of the understandable concern that the parameter space of an agent's control policies seemed too large to search efficiently. As it turns out some ~40 years later Williams and Sutton realized that there are efficient approaches for direct policy learning (learning the gradient wrt policy parameters) even in the absence of learning value (Williams 1992; Sutton et al. 2000). The strength of value learning (attempting to represent the environment states) that allows it to nicely converge on an optimal approximation is also what is limiting in its ability to account for the diverse solutions that individual animals may find. The environment we use here is deterministic and thus all value models converge similarly. The individual animal's solutions, as we show in Fig 1, are not deterministic or related to each other by a simple scaling term, but rather quite idiosyncratic befitting the multiple solutions to the problem. In this sense ACTR focuses on the agent rather than the environment - and we believe this is a very intriguing conceptual shift in thinking about RL and associative learning, both of which have focused almost exclusively on the information present in the environment.

Relatedly, the objective function that was used to train the network was based on time to collect reward once available, which can be considered a discounted value function. Therefore, the network seemed to be trained to calculate value, and then the derivative of that signal was used to produce the dopamine signal (RPE) - which all in all seem exceedingly similar to traditional actor-critic models.

We appreciate the reviewers point and drawing out connections between our formulation and existing theory. Importantly, both direct policy learning and value learning are methods that seek to, in the terminology of reinforcement learning, maximize the returned reward from the environment - thus, both methods can be considered to maximize value. They are just distinct algorithms for achieving such maximization. We do not mean to claim that policy learning is not trying to maximize reward return - the difference is just what gradient is being estimated and descended. In policy learning it is a gradient of reward with respect to change in policy. In value learning it is a gradient of the predicted value (value function).

Or to state this another way - we are generally sympathetic to the point that there must be approximate similarities that can be drawn between reinforcement learning models and they certainly all share some common features. However, there are a number of differences between the methods (as noted above these differences are described well in the RL literature, e.g. (Kober, Bagnell, and Peters 2013)) despite the fact that the goal of maximizing returned reward is shared. Moreover, there must be additional similarities. The field has found approximate correlates to RPEs for a long time across many groups (Schultz 2015), our group's data included. But, the exceptions that one model explains but another does not is critical to arriving at models that provide the best description of the underlying learning process. We provide one example of why this is valuable. To date there is no description of a biologically plausible policy learning formulation that captures animal behavior in the detail that we succeed to, nor is there a description of a biological implementation of adaptive modulation of learning rate. These are both interesting insights that will spur future research. It is not to say that continuing work on value learning is not interesting as well, but all modern machine learning approaches combine these methods in sophisticated ways that nonetheless remain less efficient than animal brains. We hope to provide insight that pushes such thinking forward and inspires new experimental inquiry.

Finally, we do not just draw subtle distinctions in correlates, but we also use causal manipulations to demonstrate how this distinction matters for learning. We hope that a similar filter will be applied to observations of similarities between models: are these models similar enough that they don't make meaningfully distinct predictions? We believe that we have shown that they are not.

I did think it was interesting that dopamine early on was so predictive of individual differences in behavior, and that their model recapitulated it. But I couldn't find an explanation of why their model produced that effect. Also, could they show the learning trajectory of the two groups of mice?

We now show the analysis requested by the reviewer and appreciate the suggestion greatly. It is a very helpful visualization (Fig 2c, right). Second, we have endeavored to be more explicit about how this aspect of the model works in the text.

Here is a complementary, additional attempt to explain this result that is a bit longer than we have space for in the text.

We propose that a subset of mice fail to effectively use the cue to elicit sustained, preparatory behavioral policies as a consequence of individual differences in initialization. This can happen because of the somewhat complicated interplay between reactive and preparatory components of behavioral policies and key properties of policy learning and not because mice used a totally distinct strategy (i.e. baseline licking). In policy learning agents are attempting to follow the noisy gradient of performance as they adjust their policies. A key problem that arises for complex policies (ours isn't extremely complex, just a reactive and sustained component, but that is enough complexity) is something referred to as "vanishing gradients". If an agent learns to react very fast to the presence of reward and learns that too soon before they have started to develop a predictive, preparatory policy they have a bit of a problem. The marginal improvement in performance becomes very small for incremental increases in preparatory licking. As a result the learning rule cannot pick up on the improvement of an incremental change in preparatory policy over the noisiness of the licking plant and the good reactive component. Thus, the preparatory policy doesn't 'see' much of a gradient (hence the term 'vanishing gradient') and this component of learning stalls out (although the fast, reactive components to cues and reward are unaffected - explaining the normal DA cue response in the context of our model). Since DA transients (beta) reflects the balance of reactive and sustained components it provides some insight into the initial conditions of the model and we propose latent states in animal's that are not revealed well in overt behavior.

It seems they claim a key piece of evidence in support of their model is that dopamine causes activity away from the policy when they are showing little licking behavior. Their explanation on line 267 of why DA enhancement on lick+ trials lead to less licking is as follows : "Such trials with a negative performance error are generally lick+ trials that occur relatively late in learning and have sustained licking that happens to terminate prior to reward delivery. Selectively enhancing learning rates on these trials with a negative performance error can have the effect of pushing the model away from a policy with sustained licking especially by reducing transient response components." ***My understanding of this explanation is that despite calling these trials lick+, they are actually licking less immediately before the reward. Can they show this directly in both model and mice? If this explanation is correct, a value-based model would make the same prediction - if the behavior immediately before the reward is "not licking" that will strengthen the policy in a policy model and reinforce the action in a value learning model (since most recent behavior is most eligible for modification given eligibility trace). Ultimately, it is not at all convincing that they have found a regime that dissociates a value model from a policy model.***

Just in case this is a simple misunderstanding, we want to clarify at the outset that the above statement referred to a select subgroup of lick+ trials in which preparatory licking did not produce good performance, while in many lick+ trials licking does not end early in a suboptimal way. Even though these select lick+ trials with negative performance errors do not always happen, the vanishing gradient for lick+ trials with good performance means that the subgroup of "bad" lick+ trials can come to dominate learning under the close-loop stimulation contingency in our experiment. We now include a graph in Ext Data Fig 6e (shown at right) that illustrates this point. For performance errors across trials in the second half of training (when stimLick+ effects are most clear), there is a bias in the distribution towards negative performance errors in lick+ trials. The only way for these trials to exhibit a negative performance error is to have a good or pretty good policy and then have an individual trial (for a variety of reasons) exhibit less good performance. Since lick+ correlates with having a good/prett good policy, negative performance errors are more likely on lick+ trials. On the converse, the probability of having a good policy (necessary for negative PE) but exhibiting a lick- trial type is low.

That possible misunderstanding aside, the explanation offered by the reviewer for a mechanism revealed by StimLick+ trials cannot be reconciled with the other half of the experimental data. It would imply StimLick- could not work - it would reinforce "not licking", but we observe stimLick- produces extra licking beyond that in control animals. Moreover, their explanation conflicts with other results in the previous manuscript that larger, uncalibrated

stimulations reinforce licking (and thus do not reinforce “not licking” as proposed by the reviewer) when conditioned in an identical way on lick+ trials (previously Fig 6e-g, now Fig 4h-i). It is also not clear that the reviewer’s explanation would account for the changes in DA cue signals in stimLick- vs stimLick+ animals—regardless of the actions being reinforced, value learning predicts that increased DA signals at reward should result in larger DA signals at a predictive cue. The only route that we have found to reconciling all of these results is to invoke a learning rate signaled by phasic DA activity that is independent from (though can be correlated with) the error signal that directs learning.

I found the mouse’s learned behavior itself a bit confusing - The mice are getting much more efficient at licking w/ low latency over time, irrespective of the presence of the cue. Any learning about the cue is much more subtle and comes later. Is that because they are mostly learning about the solenoid click? Perhaps that’s why the DA correlates in general are rather subtle for the cue, the behavior learning seems very much about the solenoid opening, and much less learning about the cue.

These are very important points and we have now done 2 new experiments to address the reviewer’s concern, a new set of model-based analyses, and simulations to address these multiple questions. Here we describe the two new experiments.

Learning about the sensory evidence of reward availability—be it a solenoid click or chemosensation of a water drop under the nose—does not necessarily preclude learning about the cue. We have found that if anything mice take longer to learn cued behavior when the solenoid is silenced. Shown for comparison are 4 mice trained under identical conditions and trial statistics as the mice in this study, but with a silent solenoid whose opening is undetectable to the animals (unfortunately DA signals and face video were not recorded for these animals). Intuitively, this is likely the same principle that pet trainers exploit when using a clicker to confirm that desired behavior has been performed. Formally, neither RPE models nor our ACTR policy model predict that sensory evidence of reward availability would disrupt cue learning. One touted strength of temporal difference learning is its ability to transfer to the earliest predictor of reward regardless of whether subsequent predictors are present, a quality that famously allowed it to reproduce psychological phenomena like blocking. Nonetheless, we take the reviewer’s question to be about behavioral learning in mice and we do not find evidence that an audible solenoid impairs learning. We do have a 1-sec long trace interval (time from tone offset to reward) and this is well known to slow learning relative to delay conditioning or shorter traces (0.5 sec is often used instead). In this particular case we see that as an advantage as the somewhat slower learning allows us to discover the multiple separable components that can be otherwise obscured by pre-shaping, rapid delay conditioning, and other paradigms.

We also have collected some new data that we believe further address this issue. The current pavlovian design was modified such that after the 1 sec delay between cue and reward, animals had to lick in order to trigger reward delivery. Thus, the cue provides much more unique predictive information about when reward will be available, as the solenoid click will only occur after mice have already licked once for reward. Here again we find that mice consistently learn this task by driving down the reward collection latency in part by establishing cue-driven preparatory responses. If anything the learning of this cue driven response is a bit slower compared to the standard paradigm used throughout the paper with an audible (but quiet relative to background noise (Coddington and Dudman 2018)) solenoid. Thus, these two new experiments suggest that the audible solenoid does not impair cue learning, rather if anything it facilitates it. The most parsimonious account is that the somewhat long trace interval (1.5 sec if animals primarily respond to cue onset), possibly lower salience auditory cue compared with high

concentration odors used in many studies are more likely explanations for “slow” learning. But again, we believe that the time constant of learning here is very useful to reveal the two components of learning that could otherwise have been missed.

I found the Introduction to be very confusing. For example, in Line 26 about ‘performance errors’ versus RPEs - When I first read it I thought perhaps they were referring to RPEs with respect to actions vs stimuli. But eventually ***I thought they meant the discounted value function (since they used time to reward collection as their performance error). It would be helpful to provide a clear definition of performance prediction errors. It was extremely difficult to read & follow the Methods section on their model. I suggest the authors should separate an explanation of the equations and concepts of the model versus details of the implementation, simulations etc.***

We have now updated Fig 1 to be much more clear about the calculation of variables . Furthermore, we have clarified these calculations substantially in the Methods and now offer a ‘pseudocode’ version of the ACTR algorithm to facilitate understanding of the key computations which is reproduced below for convenience. We believe this addresses the major concerns of the reviewer and appreciate the detailed guidance on what was confusing. Moreover, the code will be available and we cross reference individual model components to the line of code where they are computed in the model code.

ACTR Simulation - pseudocode

Initialize trial to T=0
Initialize ACTR with $W(0)$, $\mathcal{S}_{rew}(T)$, $\mathcal{S}_{cue}(T)$

repeat

Run RNN simulation engine for trial T

Compute plant input $\pi(T) = O(T) + \mathcal{S}(T)$

Compute lick output $L(t) = \text{Plant}(\pi(T))$

Compute latency to collect reward $t_{collect} \leftarrow \text{find } L(t) > t_{rew}$

Compute Cost(T) = $1 - \exp(-\Delta t/500)$

Evaluate eligibility trace at collection $e \leftarrow e_{ij}(t_{collect})$

Compute $\beta_{DA} = 1 + \phi(\Delta\pi(t_{rew}) + \mathcal{S}_{rew})$

Compute $R_{obj}(T) = 1 - (1 - \exp(-\Delta t/500)) - O(T, t_{rew} - 1)$

Estimate objective gradient PE = $R_{obj}(T) - \langle R(T) \rangle$

Compute update $\Delta W = -\eta_l \times e \times \text{PE} \times \beta_{DA}$

Update $W(T+1) \leftarrow W(T) + \Delta W$

Update $\mathcal{S}_{reward}(T+1) \leftarrow \mathcal{S}_{rew}(T) + \eta_s \times R_{obj}(T) \times \beta_{DA}$

Update $\mathcal{S}_{cue}(T+1) \leftarrow \mathcal{S}_{cue}(T) + \eta_s \times R_{obj}(T) \times \beta_{DA}$

Until T==800

Definitions

T = current trial
W = RNN connection weight matrix
 \mathcal{S} = sensory input strength

O = RNN output; π = behavioral policy

$\Delta t = t_{collect} - t_{rew}$

ϕ = nonlinear (sigmoid) transform

$\langle R(T) \rangle$ = running mean PE

η_l = baseline learning rate for RNN

η_s = baseline learning rate for input \mathcal{S}

In addition, evidence for claim in Introduction that policy-only learning algorithms vs actor-critic models (that also include a policy) are better suited to understand individual differences is lacking.

In the introduction we note that a primary issue with direct policy learning is that it is much noisier and more variable than value learning. This is due to the fact that the gradient of the objective with respect to the parameters of the policy generally cannot be calculated directly. Specifically, in our biologically plausible implementation there is stochasticity from both the approximate gradient estimation (Miconi 2017) and the due to the noise in the motor plant. Finally, we now provide a few references that all state this point about the variability of policy learning explicitly including (Konda and Tsitsiklis 2000; Kober, Bagnell, and Peters 2013; Schulman et al. 2017; Bottou, Curtis, and Nocedal 2018). An example at right.

A particularly important issue is lack of clarity on how the DA signal is calculated, as the dopamine "beta" signal doesn't seem to have an equation in methods, and is explained in words very differently in Line 770 vs line 202 since one spot mentions a derivative and the other a sum. Not knowing how the DA signal was calculated really hampered my understanding of the model.

We apologize for any confusion that may have resulted. We now show the key equation for beta in the main figure, define the equation in the methods ($\beta_{DA} = T + \phi(\Delta\pi(t_{rew}) + \delta_{rew})$) where T is 'tonic activity' and equal to 1 for majority of the paper except where noted in Fig. 1j, and also in the newly added pseudocode shown above.

Is 4k & 4l referenced in Results? I searched for a reference to those figure panels and it did not come up. This seems like another critical omission, since those panels are the key predictions of the model that seem to really be the crux of the paper.

Fig. 4 like all other figures have been substantially revised. This has allowed us to put data and model predictions close together in the same figure that facilitates careful comparison. Moreover, we add new simulations of alternative model formulations in multiple figures, but of particular relevance to this point in Fig 1 and Fig 4.

Why did lick- group lick more than lick+ in Fig 4H?

Assuming this is an issue relating to confusing presentation in original figures, we hope that it has been resolved in the updated manuscript. Otherwise, since there was nothing about "lick-" and "lick+" in the original Fig 4, we infer that this question might have referred to licking on lick- vs lick+ trials in Fig 4L? That panel was modeling the closed-loop stimulation experiment, and so the explanation is that DA actually signals a learning rate rather than an error in the ACTR model, and so closed loop stimulation produces these seemingly paradoxical effects of biasing against the stimulation contingency (mouse and ACTR data is now the subject of Fig 4 in the new text).

For figure 3.a.2, they should include the uncued reward response as well, to demonstrate if this an RPE or reward response.

The cued vs uncued analysis for this data is shown in the new Fig 3f, and was originally shown in the original manuscript in Fig 2c.

Calibration experiment in Figure 6 is interesting, but I don't think it makes the effect of DA on producing a cue response at all uninteresting or unconvincing. They are calibrating to the DA signal immediately adjacent to the optical fiber in NAc. This should be conservative since not all dopamine neurons are in the immediate vicinity of the NAc fiber.

We do agree that not all dopamine neurons are in the immediate vicinity of the fiber, but we note that the relevant fraction for our optogenetics experiment is all dopamine neurons stimulated by the optical fibers over VTA. We firmly argue that in this case the fiber location in NAcc should be considered by far the most relevant representative

Actor-Critic Algorithms

Vijay R. Konda John N. Tsitsiklis
 Laboratory for Information and Decision Systems,
 Massachusetts Institute of Technology,
 Cambridge, MA, 02139.
 konda@mit.edu, jnt@mit.edu

The vast majority of Reinforcement Learning (RL) [9] and Neuro-Dynamic Programming (NDP) [1] methods fall into one of the following two categories:

- (a) Actor-only methods work with a parameterized family of policies. The gradient of the performance, with respect to the actor parameters, is directly estimated by simulation, and the parameters are updated in a direction of improvement [4, 5, 8, 13]. A possible drawback of such methods is that the gradient estimators may have a large variance. Furthermore, as the policy changes, a new gradient is estimated independently of past estimates. Hence, there is no "learning," in the sense of accumulation and consolidation of older information.

tion of older information.

Hence, there is no "learning," in the sense of accumulation and consolidation of older information. A new gradient is estimated independently of past estimates. Hence, there is no "learning," in the sense of accumulation and consolidation of older information.

fraction, since it is the largest single projection target of VTA neurons and it is the focus of standing arguments about the causality of DA signaling for value learning (Steinberg et al. 2013). Consistent with this we find very clear correlations when simultaneously imaging in VTA and NAcc (see figure at right) - suggesting that we sample a largely overlapping population in VTA and NAcc. It is common practice in the field to interpret differences of only a few Hz in peak firing rates to be significant confirmations of predicted roles of DA in learning. Further, it is often the case that uncalibrated stimulation is delivered to only one hemisphere in order to produce reinforcement effects. Here we are stimulating a majority of VTA-DA neurons bilaterally in a way that reproduces the entirety of measured reward responses in the main projection target of the VTA.

We do agree with the reviewer that even calibration in the manner we performed has some limitations and caveats to its interpretation. We have also now performed a new control experiment to assess whether uncalibrated stimulation produces a different spatial pattern of dopamine release across frontal cortex, dorsal and ventral striatum (Ext Data Fig 5). This new experiment strongly suggests that uncalibrated stimulation produces longer and larger changes in dopamine release, but does not produce a substantial change in the spatial pattern of release across major DA target areas. Moreover, it suggests that the measurement in NAcc is accurate and proportional to stimulation, indicating it is a reasonable target for calibration. Finally, it also demonstrates that as reason would suggest, even the relatively small (although still large in that it reproduces the largest measured physiological signals in our data) calibrated stimulation over the VTA recruits significant DA signals across many projection targets. It is unclear what specifically the reviewer had in mind, but if they were concerned that our calibrated stimulation did not produce release in areas such as the prefrontal cortex or the dorsal striatum, their concerns should be addressed here.

Also there may be other neuromodulators etc that help enhance the effect of DA. **They are still showing that DA signals are sufficient to enhance a cue response, consistent w/ learning about the value of the cue.** Presumably inhibition would be sufficient to decrease that cue response. **Also, does strong stimulation of DA in their model lead to a greater cue response? If not, why not?**

We now have reorganized the relevant figures to make these points much more clear. We agree that our 'uncalibrated' stimulation experiments show that DA signals can be sufficient to enhance a cue response. But, we also note a few important points that further distinguish our results from those in prior studies.

(1) Our data are consistent with either a pure effect on learning rate without a value-like effect (calibrated stimulation; other correlates in the paper); and

(2) We also now clarify in Figure 3 that there are still differences between the effect we observe and predictions in value learning models, and thus we refer to the effect as 'value-like'.

(3) The data is most consistent with our models in which uncalibrated DA stimulation produces *both* a learning rate effect and a value-like (error) effect. Thus, we always see data consistent with DA modulating learning rate even when value-like effects start to emerge in addition.

(4) We show that calibrated stimulation when delivered contingent on a specific trial type can be sufficient to decrease cue responses - an effect that is not consistent with value learning predictions (multiple different model predictions now shown in Fig 4). This leads to the insight that there is a surprising dissociation between the learning effects of calibrated and uncalibrated stimulation.

The strong stimulation of DA does produce a greater cue response in the ACTR model and the effect is very similar to that observed in the data when we consider DA stimulation to produce a combined rate and value-like effect. The panel from Fig 4 is shown at right, right column is model, left column is data.

- Bennett, Daniel, Yael Niv, and Angela J. Langdon. 2021. "Value-Free Reinforcement Learning: Policy Optimization as a Minimal Model of Operant Behavior." *Current Opinion in Behavioral Sciences* 41 (October): 114–21.
- Bollu, Tejasratan, Brendan S. Ito, Samuel C. Whitehead, Brian Kardon, James Redd, Mei Hong Liu, and Jesse H. Goldberg. 2021. "Cortex-Dependent Corrections as the Tongue Reaches for and Misses Targets." *Nature* 594 (7861): 82–87.
- Bottou, Léon, Frank E. Curtis, and Jorge Nocedal. 2018. "Optimization Methods for Large-Scale Machine Learning." *SIAM Review*. <https://doi.org/10.1137/16m1080173>.
- Chen, Ruidong, Pavel A. Puzerey, Andrea C. Roeser, Tori E. Riccelli, Archana Podury, Kamal Maher, Alexander R. Farhang, and Jesse H. Goldberg. 2019. "Songbird Ventral Pallidum Sends Diverse Performance Error Signals to Dopaminergic Midbrain." *Neuron*. <https://doi.org/10.1016/j.neuron.2019.04.038>.
- Coddington, Luke T., and Joshua T. Dudman. 2018. "The Timing of Action Determines Reward Prediction Signals in Identified Midbrain Dopamine Neurons." *Nature Neuroscience* 21 (11): 1563–73.
- . 2019. "Learning from Action: Reconsidering Movement Signaling in Midbrain Dopamine Neuron Activity." *Neuron* 104 (1): 63–77.
- . 2021. "In Vivo Optogenetics with Stimulus Calibration." *Methods in Molecular Biology* 2188: 273–83.
- Cohen, Jeremiah Y., Sebastian Haesler, Linh Vong, Bradford B. Lowell, and Naoshige Uchida. 2012. "Neuron-Type-Specific Signals for Reward and Punishment in the Ventral Tegmental Area." *Nature* 482 (7383): 85–88.
- Eshel, Neir, Michael Bukwich, Vinod Rao, Vivian Hemmelder, Ju Tian, and Naoshige Uchida. 2015. "Arithmetic and Local Circuitry Underlying Dopamine Prediction Errors." *Nature*. <https://doi.org/10.1038/nature14855>.
- Gadagkar, Vikram, Pavel A. Puzerey, Ruidong Chen, Eliza Baird-Daniel, Alexander R. Farhang, and Jesse H. Goldberg. 2016. "Dopamine Neurons Encode Performance Error in Singing Birds." *Science* 354 (6317): 1278–82.
- Galifianes, Gregorio Luis, Claudia Bonardi, and Daniel Huber. 2018. "Directional Reaching for Water as a Cortex-Dependent Behavioral Framework for Mice." *Cell Reports*. <https://doi.org/10.1016/j.celrep.2018.02.042>.
- Gallistel, C. R., and J. Gibbon. 2000. "Time, Rate, and Conditioning." *Psychological Review* 107 (2): 289–344.
- Gong, Rong, Shengjin Xu, Ann Hermundstad, Yang Yu, and Scott M. Sternson. 2020. "Hindbrain Double-Negative Feedback Mediates Palatability-Guided Food and Water Consumption." *Cell* 182 (6): 1589–1605.e22.
- Gutierrez, Ranier, Jose M. Carmona, Miguel A. L. Nicolelis, and S. A. Simon. 2006. "Orbitofrontal Ensemble Activity Monitors Licking and Distinguishes among Natural Rewards." *Journal of Neurophysiology* 95 (1): 119–33.
- Izhikevich, Eugene M. 2007. "Solving the Distal Reward Problem through Linkage of STDP and Dopamine Signaling." *Cerebral Cortex* 17 (10): 2443–52.
- Kepecs, Adam, Naoshige Uchida, Hatim A. Zariwala, and Zachary F. Mainen. 2008. "Neural Correlates, Computation and Behavioural Impact of Decision Confidence." *Nature* 455 (7210): 227–31.
- Kingma, Diederik P., and Jimmy Ba. 2014. "Adam: A Method for Stochastic Optimization." *arXiv [cs.LG]*. arXiv. <http://arxiv.org/abs/1412.6980>.
- Kober, J., J. A. Bagnell, and J. Peters. 2013. "Reinforcement Learning in Robotics: A Survey." *The International Journal of*. <https://journals.sagepub.com/doi/abs/10.1177/0278364913495721>.
- Konda, Vijay R., and John N. Tsitsiklis. 2000. "Actor-Critic Algorithms." In *Advances in Neural Information Processing Systems*, 1008–14.
- Kuchibhotla, Kishore V., Tom Hindmarsh Sten, Eleni S. Papadopyannis, Sarah Elnozahy, Kelly A. Fogelson, Rupesh Kumar, Yves Boubenec, Peter C. Holland, Srdjan Ostojic, and Robert C. Froemke. 2019. "Dissociating Task Acquisition from Expression during Learning Reveals Latent Knowledge." *Nature Communications* 10 (1): 2151.
- Lee, Kwang, Leslie D. Claar, Ayaka Hachisuka, Konstantin I. Bakhurin, Jacquelyn Nguyen, Jeremy M. Trott, Jay L. Gill, and Sotiris C. Masmanidis. 2020. "Temporally Restricted Dopaminergic Control of Reward-Conditioned Movements." *Nature Neuroscience*. <https://doi.org/10.1038/s41593-019-0567-0>.
- Lee, Kwang, Sandra M. Holley, Justin L. Shobe, Natalie C. Chong, Carlos Cepeda, Michael S. Levine, and Sotiris C. Masmanidis. 2018. "Parvalbumin Interneurons Modulate Striatal Output and Enhance Performance during Associative Learning." *Neuron* 99 (1): 239.
- Miconi, Thomas. 2017. "Biologically Plausible Learning in Recurrent Neural Networks Reproduces Neural Dynamics Observed during Cognitive Tasks." *eLife* 6 (February). <https://doi.org/10.7554/eLife.20899>.
- Mohebi, Ali, Jeffrey R. Pettibone, Arif A. Hamid, Jenny-Marie T. Wong, Leah T. Vinson, Tommaso Patriarchi, Lin Tian, Robert T. Kennedy, and Joshua D. Berke. 2019. "Dissociable Dopamine Dynamics for Learning and Motivation." *Nature* 570 (7759): 65–70.
- Mongillo, Gianluigi, Hanan Shteingart, and Yonatan Loewenstein. 2014. "The Misbehavior of Reinforcement Learning." *Proceedings of the IEEE* 102 (4): 528–41.
- Nachum, Ofir, Mohammad Norouzi, Kelvin Xu, and Dale Schuurmans. 2017. "Bridging the Gap between Value and Policy Based Reinforcement Learning." *Advances in Neural Information Processing Systems* 30. <https://proceedings.neurips.cc/paper/2017/hash/fac9f743b083008a894eee7baa16469-Abstract.html>.
- Pan, Wei-Xing, Jennifer Brown, and Joshua T. Dudman. 2013. "Neural Signals of Extinction in the Inhibitory Microcircuit of the Ventral Midbrain." *Nature Neuroscience* 16 (1): 71–78.
- Pan, Wei-Xing, Luke T. Coddington, and Joshua T. Dudman. 2021. "Dissociable Contributions of Phasic Dopamine

- Activity to Reward and Prediction." *Cell Reports* 36 (10): 109684.
- Pan, Weixing X., Tianyi Mao, and Joshua T. Dudman. 2010. "Inputs to the Dorsal Striatum of the Mouse Reflect the Parallel Circuit Architecture of the Forebrain." *Frontiers in Neuroanatomy* 4 (December): 147.
- Pasquereau, B., and R. S. Turner. 2013. "Limited Encoding of Effort by Dopamine Neurons in a Cost-Benefit Trade-off Task." *The Journal of Neuroscience: The Official Journal of the Society for Neuroscience* 33 (19): 8288–8300.
- Schulman, John, Filip Wolski, Prafulla Dhariwal, Alec Radford, and Oleg Klimov. 2017. "Proximal Policy Optimization Algorithms." *arXiv [cs.LG]*. arXiv. <http://arxiv.org/abs/1707.06347>.
- Schultz, W. 2015. "Neuronal Reward and Decision Signals: From Theories to Data." *Physiological Reviews* 95 (3): 853–951.
- Schultz, W., P. Apicella, and T. Ljungberg. 1993. "Responses of Monkey Dopamine Neurons to Reward and Conditioned Stimuli during Successive Steps of Learning a Delayed Response Task." *The Journal of Neuroscience: The Official Journal of the Society for Neuroscience* 13 (3): 900–913.
- Steinberg, Elizabeth E., Ronald Keiflin, Josiah R. Boivin, Ilana B. Witten, Karl Deisseroth, and Patricia H. Janak. 2013. "A Causal Link between Prediction Errors, Dopamine Neurons and Learning." *Nature Neuroscience* 16 (7): 966–73.
- Sussillo, David, and L. F. Abbott. 2009. "Generating Coherent Patterns of Activity from Chaotic Neural Networks." *Neuron* 63 (4): 544–57.
- Sutton, Richard S., David McAllester, Satinder Singh, and Yishay Mansour. 2000. "Policy Gradient Methods for Reinforcement Learning with Function Approximation." In *Advances in Neural Information Processing Systems*, edited by S.olla, T. Leen, and K. Müller, 12:1057–63. MIT Press.
- Williams, Ronald J. 1992. "Simple Statistical Gradient-Following Algorithms for Connectionist Reinforcement Learning." *Machine Learning* 8 (3): 229–56.
- Bollu, Tejasratan, Brendan S. Ito, Samuel C. Whitehead, Brian Kardon, James Redd, Mei Hong Liu, and Jesse H. Goldberg. 2021. "Cortex-Dependent Corrections as the Tongue Reaches for and Misses Targets." *Nature* 594 (7861): 82–87.
- Bottou, Léon, Frank E. Curtis, and Jorge Nocedal. 2018. "Optimization Methods for Large-Scale Machine Learning." *SIAM Review*. <https://doi.org/10.1137/16m1080173>.
- Chen, Ruidong, Pavel A. Puzerey, Andrea C. Roeser, Tori E. Riccelli, Archana Podury, Kamal Maher, Alexander R. Farhang, and Jesse H. Goldberg. 2019. "Songbird Ventral Pallidum Sends Diverse Performance Error Signals to Dopaminergic Midbrain." *Neuron*. <https://doi.org/10.1016/j.neuron.2019.04.038>.
- Coddingon, Luke T., and Joshua T. Dudman. 2018. "The Timing of Action Determines Reward Prediction Signals in Identified Midbrain Dopamine Neurons." *Nature Neuroscience* 21 (11): 1563–73.
- . 2019. "Learning from Action: Reconsidering Movement Signaling in Midbrain Dopamine Neuron Activity." *Neuron* 104 (1): 63–77.
- . 2021. "In Vivo Optogenetics with Stimulus Calibration." *Methods in Molecular Biology* 2188: 273–83.
- Cohen, Jeremiah Y., Sebastian Haesler, Linh Vong, Bradford B. Lowell, and Naoshige Uchida. 2012. "Neuron-Type-Specific Signals for Reward and Punishment in the Ventral Tegmental Area." *Nature* 482 (7383): 85–88.
- Eshel, Neir, Michael Bukwich, Vinod Rao, Vivian Hemmelder, Ju Tian, and Naoshige Uchida. 2015. "Arithmetic and Local Circuitry Underlying Dopamine Prediction Errors." *Nature*. <https://doi.org/10.1038/nature14855>.
- Gadagkar, Vikram, Pavel A. Puzerey, Ruidong Chen, Eliza Baird-Daniel, Alexander R. Farhang, and Jesse H. Goldberg. 2016. "Dopamine Neurons Encode Performance Error in Singing Birds." *Science* 354 (6317): 1278–82.
- Gallistel, C. R., and J. Gibbon. 2000. "Time, Rate, and Conditioning." *Psychological Review* 107 (2): 289–344.
- Gong, Rong, Shengjin Xu, Ann Hermundstad, Yang Yu, and Scott M. Sternson. 2020. "Hindbrain Double-Negative Feedback Mediates Palatability-Guided Food and Water Consumption." *Cell* 182 (6): 1589–1605.e22.
- Gutierrez, Ranier, Jose M. Carmona, Miguel A. L. Nicolelis, and S. A. Simon. 2006. "Orbitofrontal Ensemble Activity Monitors Licking and Distinguishes among Natural Rewards." *Journal of Neurophysiology* 95 (1): 119–33.
- Izhikevich, Eugene M. 2007. "Solving the Distal Reward Problem through Linkage of STDP and Dopamine Signaling." *Cerebral Cortex* 17 (10): 2443–52.
- Kepecs, Adam, Naoshige Uchida, Hatim A. Zariwala, and Zachary F. Mainen. 2008. "Neural Correlates, Computation and Behavioural Impact of Decision Confidence." *Nature* 455 (7210): 227–31.
- Kingma, Diederik P., and Jimmy Ba. 2014. "Adam: A Method for Stochastic Optimization." *arXiv [cs.LG]*. arXiv. <http://arxiv.org/abs/1412.6980>.
- Kuchibhotla, Kishore V., Tom Hindmarsh Sten, Eleni S. Papadoyannis, Sarah Elnozahy, Kelly A. Fogelson, Rupesh Kumar, Yves Boubenec, Peter C. Holland, Srdjan Ostojic, and Robert C. Froemke. 2019. "Dissociating Task Acquisition from Expression during Learning Reveals Latent Knowledge." *Nature Communications* 10 (1): 2151.
- Lee, Kwang, Leslie D. Claar, Ayaka Hachisuka, Konstantin I. Bakhurin, Jacquelyn Nguyen, Jeremy M. Trott, Jay L. Gill, and Sotiris C. Masmanidis. 2020. "Temporally Restricted Dopaminergic Control of Reward-Conditioned Movements." *Nature Neuroscience*. <https://doi.org/10.1038/s41593-019-0567-0>.

- Lee, Kwang, Sandra M. Holley, Justin L. Shobe, Natalie C. Chong, Carlos Cepeda, Michael S. Levine, and Sotiris C. Masmanidis. 2018. "Parvalbumin Interneurons Modulate Striatal Output and Enhance Performance during Associative Learning." *Neuron* 99 (1): 239.
- Mohebi, Ali, Jeffrey R. Pettibone, Arif A. Hamid, Jenny-Marie T. Wong, Leah T. Vinson, Tommaso Patriarchi, Lin Tian, Robert T. Kennedy, and Joshua D. Berke. 2019. "Dissociable Dopamine Dynamics for Learning and Motivation." *Nature* 570 (7759): 65–70.
- Pan, Wei-Xing, Jennifer Brown, and Joshua Tate Dudman. 2013. "Neural Signals of Extinction in the Inhibitory Microcircuit of the Ventral Midbrain." *Nature Neuroscience* 16 (1): 71–78.
- Pan, Wei-Xing, Luke T. Coddington, and Joshua T. Dudman. 2021. "Dissociable Contributions of Phasic Dopamine Activity to Reward and Prediction." *Cell Reports* 36 (10): 109684.
- Pan, Weixing X., Tianyi Mao, and Joshua T. Dudman. 2010. "Inputs to the Dorsal Striatum of the Mouse Reflect the Parallel Circuit Architecture of the Forebrain." *Frontiers in Neuroanatomy* 4 (December): 147.
- Pasquereau, B., and R. S. Turner. 2013. "Limited Encoding of Effort by Dopamine Neurons in a Cost-Benefit Trade-off Task." *The Journal of Neuroscience: The Official Journal of the Society for Neuroscience* 33 (19): 8288–8300.
- Schultz, W. 2015. "Neuronal Reward and Decision Signals: From Theories to Data." *Physiological Reviews* 95 (3): 853–951.
- Schultz, W., P. Apicella, and T. Ljungberg. 1993. "Responses of Monkey Dopamine Neurons to Reward and Conditioned Stimuli during Successive Steps of Learning a Delayed Response Task." *The Journal of Neuroscience: The Official Journal of the Society for Neuroscience* 13 (3): 900–913.
- Sussillo, David, and L. F. Abbott. 2009. "Generating Coherent Patterns of Activity from Chaotic Neural Networks." *Neuron* 63 (4): 544–57.

Reviewer Reports on the First Revision:

Referee expertise:

Referee #1:

Referee #2:

Referee #3:

Referees' comments:

Referee #1 (Remarks to the Author):

I was generally satisfied with the responses to my comments but I have a few more suggestions. The cost term in the model should be explained in a more intuitive way in the main body of the text. Licking should only occur after the cue is delivered. If the authors wish to show that the cue served as a discriminative lick/no lick signal, performance should be defined based on the number of licks on cued versus cue-omission trials using a ROC-type approach.

I was intrigued by this sentence and would urge the authors to unpack it.: "activity of the feedback unit was the sum of the state change in the behavioral plant (akin to an efference copy of reward-related action initiation commands 47) and the change in behavioral policy at the time of reward delivery(akin to a reward-predictive sensory evidence8). This feedback scheme has a direct and intentional parallel to the phasic activity of midbrain DA neurons in this task"

Many readers will not be familiar with REINFORCE and the authors need to spell out how REINFORCE is different from a reinforcer in psychology and an RPE in neuroscience.

When you say: "(3) a basal learning rate was intact but there was no adaptive component (akin to disruption of phasic mDA activity..)" I think you mean disruption of a phasic DA RPE signal.

Referee #3 (Remarks to the Author):

While I appreciate their efforts to revise the paper, overall I found that their responses did not help clarify.

I agree with the authors that policy models may in fact better explain behavioral and neural data than value models (while hybrid actor-critic models are probably the best to encompass all data, which has long been thought in the field). However, my enthusiasm for this paper is limited because they do not seem to be providing much in the way of rigorous support for that assertion. When I wrote the difference between various types of RL models was subtle, I didn't mean it was nonexistent, I meant they would need to provide strong and clear evidence to dissociate the models. They are not providing a systematic comparison of value versus policy RL models to see which class of models best fits behavior and/ or neural activity on a trial-by-trial basis (or moment-by-moment

basis), to rigorously support the assertion that a policy model best explains the data. Note that the model comparisons of collection latency does not seem to be informative (1j). One would expect an RNN should converge to zero latency if trained correctly so not clear what is the issue exactly. To rigorously compare the ability of different models (such as policy vs value) to explain data, the best fit parameters of each model to the data are normally identified, and the likelihood of the data under the best fit parameters is normally compared. For example see Li & Daw 2011, where they did exactly that- they compared the ability of policy versus value models to fit trial-by-trial choices, and concluded the policy model was indeed superior.

I think ultimately their central claim may be that DA provides a learning rate rather than a signed error signal. If so, I'm not sure that point has anything to do with policy versus value models, since learning rates may be "adaptive" in either class of models, again making the big picture point of their paper confusing.

I recognize that the stimlick+ and stimlick- expt appears to be a critical test in their mind of learning rate versus error signal. I have read the text and figure legend multiple times and I remain unclear about their arguments. They are not stimulating on trials with negative vs positive performance errors, they are stimulating on all lick+ vs lick- trials (if I understand correctly), so I do not follow their arguments - it's not clear which trials have which performance errors. It seems like it would be much preferable to actually design an experiment that is intended to differentiate between a contribution of DA to learning rate versus to a signed prediction error much directly, if that is their goal. They could design a task with some trials with positive and others with negative prediction errors, and showing that increasing DA increases learning rate in both sets of trials.

Moreover, their comment in the rebuttal that DA isn't traditionally thought to contribute to the policy learning in traditional actor-critic models isn't true. DA projections to dorsal striatum are thought to help train the "actor" in actor-critic models, where the actor is classically placed in dorsal striatum and the critic in ventral striatum. One idea is DA in dorsal striatum could train the actor by serving as a "baseline" term in a REINFORCE-like actor.

In addition, in re-reading the new manuscript, another big concern arose for me. They clarify "Performance errors used to train the model were proportional to the difference between the policy output at the time of reward delivery and the latency to collect water reward (see Methods)." This error signal doesn't make much sense as a learning rule for a naive agent. For the agent to calculate the difference in time between reward delivery and reward collection latency, and use that as an error signal, they would need to know when the reward has been delivered based. Isn't knowing when the reward is delivered the entire point of learning a trace conditioning task? Is the idea that they have already learned the sound of the solenoid click means that reward is available, but haven't learned how to lick in the presence of the click? This really doesn't make much sense- seems more like a supervised learning rule than a reinforcement learning rule. In fact the paper they reference (49) is a supervised learning paper.

--

Minor:

In general, the paper is hard to read. E.g. what does it mean for stimulation to have an “input-output property”? Many of the sentences are not stand-alone and they require going between the text and figures many times to understand the intended meaning.

--

Similarly, many of the figure legends seem incomplete, making it hard to understand the figures. For example Fig 4h legend doesn't seem to describe the right most panels, only the “data” panels

--

The goal of the GLM in F1 to predict behavior from other behavior measures was not clear.

--

I think they may mean “temporal difference” model, not “temporal discount” model.

--

Figure 4b- please include time axis units and clarify how many mice/sessions etc are being averaged

Author Rebuttals to First Revision:

Referee #1

-The cost term in the model should be explained in a more intuitive way in the main body of the text.

We have attempted this now in the Results section describing the model. “The performance error (PE) used to train the model was proportional to the difference between performance, as measured by the latency to collect the water reward, and expected performance, estimated by the policy output at the time of reward delivery (see Methods). Both reactive and preparatory learning occurred in proportion to this PE, but they were implemented at different positions within the network.”

-Licking should only occur after the cue is delivered. If the authors wish to show that the cue served as a discriminative lick/no lick signal, performance should be defined based on the number of licks on cued versus cue-omission trials using a ROC-type approach.

We have done this analysis, with results at right showing a high degree of discriminability between licking behavior during the delay period before reward on cued vs uncued trials.

-I was intrigued by this sentence and would urge the authors to unpack it.: “activity of the feedback unit was the sum of the state change in the behavioral plant (akin to an efference copy of reward-related action initiation commands 47) and the change in behavioral policy at the time of reward delivery(akin to a reward-predictive sensory evidence8).

This feedback scheme has a direct and intentional parallel to the phasic activity of midbrain DA neurons in this task” We now more directly explain how this relates to the findings about DA encoding the sum of action and cues from our previous work ¹, as well as why this faithfully reproduces measured DA responses.

-Many readers will not be familiar with REINFORCE and the authors need to spell out how REINFORCE is different from a reinforcer in psychology and an RPE in neuroscience.

-When you say: “(3) a basal learning rate was intact but there was no adaptive component (akin to disruption of phasic mDA activity..)” I think you mean disruption of a phasic DA RPE signal.

Thank you for the helpful guidance, we have now clarified in the revised text.

Referee #3 (Remarks to the Author):

-While I appreciate their efforts to revise the paper, overall I found that their responses did not help clarify. I agree with the authors that policy models may in fact better explain behavioral and neural data than value models... However, my enthusiasm for this paper is limited because they do not seem to be providing much in the way of rigorous support for that assertion. When I wrote the difference between various types of RL models was subtle, I didn't mean it was nonexistent, I meant they would need to provide strong and clear evidence to dissociate the models. They are not providing a systematic comparison of value versus policy RL models to see which class of models best fits behavior and/ or neural activity on a trial-by-trial basis (or moment-by-moment basis), to rigorously support the assertion that a policy model best explains the data. To rigorously compare the ability of different models (such as policy vs value) to explain data, the best fit parameters of each model to the data are normally identified, and the likelihood of the data under the best fit parameters is normally compared. For example see Li & Daw 2011, where they did exactly that- they compared the ability of policy versus value models to fit trial-by-trial choices, and concluded the policy model was indeed superior.

We thank the reviewer for providing particularly clear guidance about how additional model comparison would make them enthusiastic about our study. We have now implemented an explicit model comparison as the reviewer proposes and we also find that it provides additional, compelling motivation for the main work of the paper - articulating a plausible, neural network based implementation model of direct policy learning. We note, the reviewer has no questions whether our ACTR model explains all of our experimental data, but rather a question whether policy learning models are also better at explaining trial by trial behavioral learning data than a value learning model - which we now show they are.

In order to make a fair comparison to the low-parameter value learning models that are standard in the field²⁻⁴, we articulated a policy-gradient model with an equal number of parameters that captures a key feature of ACTR (it uses the same policy gradient). **This direct model comparison revealed both minimum negative log likelihood (-LL) and minimum AIC were obtained with a policy learning model rather than a value learning one** (revised Fig. 1h; revised Extended Data Figure 1). Moreover, policy learning models were dramatically less brittle⁵ than value learning (Ext Data Fig 1c).

These low-parameter treatments are informative as to which type of model seems best to pursue, but lack important implementational and mechanistic detail. For example, there is no clear way to account for the two dissociable learning components directly inferred from behavioral learning data (which we name 'reactive' and 'preparatory'; Fig 1), nor how to relate phasic DA signals to

Extended Data Figure 1. Comparison of low dimensional policy learning and value learning model fits to behavioral learning

- a) Fits of value (blue) and policy (orange) learning models for each mouse across the space of possible parameterizations, measured as $-\log$ likelihood (smaller number is better fit)
- b) Comparison of optimally parameterized policy and value models for each mouse, quantified by $-\log$ likelihood (left) or Aikake information criterion (right)
- c) Comparison of median parameterized policy and value models for each mouse, quantified by $-\log$ likelihood (left) or Aikake information criterion (right)

key variables in a REINFORCE policy learning formulation. Thus, we find that the principled approach the reviewer suggested above greatly strengthens the motivation in the manuscript to pursue a fuller, more plausible network-based implementation of a policy learning model, and we now incorporate their suggested model comparison into the main text, figures and add an additional extended data figure.

Note that the model comparisons of collection latency does not seem to be informative (1j). One would expect an RNN should converge to zero latency if trained correctly so not clear what is the issue exactly.

While this is already addressed, we quickly note that a comparison of performance or examination of differential predictions is a standard model comparison approach in machine learning (a recent example is Figure 2 in ⁶) due to the fact that neural network models with large numbers of parameters are not guaranteed to be practically optimizable (unlike low parameter models ⁷). This performance comparison isn't a singular piece of evidence that perfectly arbitrates between competing views, but it is a legitimate piece of evidence in favor of the function we are proposing for mesolimbic DA of setting an adaptive learning rate and not other terms in the ACTR model. With everything else equal in ACTR, using DA signals as the error term rather than the adaptive rate term caused the RNN to perform terribly because using the feedback unit activity as a signed error interferes with convergence. Given the close match of our predicted DA signals to those in real mice over learning (Fig 3), this addresses a reasonable question readers might ask: couldn't DA-like signals be used in ACTR to signal an error signal analogously to its function in value learning?

A final implication of the reviewer's statement above is that they are concerned that the model doesn't converge to zero latency, but rather to a stable plateau around ~150 ms. It is thus crucial to note that the RNN output does converge to its optimum with training (as shown in Extended Data Figure 2a); however, the latency of collection is determined by the behavioral plant which has biologically realistic reaction times and stochasticity that lead to non-zero latencies for collection. This matches mice's stable learned performance, which suffers from similar real physical constraints. We should finally note that while REINFORCE is a fascinating class of models, convergence is not proven for all implementations (as Williams noted in his original paper ⁸) and our model is a particular form similar to, but not exactly articulated by Williams, and thus convergence can only be demonstrated by simulation at this time.

I think ultimately their central claim may be that DA provides a learning rate rather than a signed error signal. If so, I'm not sure that point has anything to do with policy versus value models, since learning rates may be "adaptive" in either class of models, again making the big picture point of their paper confusing.

Our primary claim is indeed that DA functions as a learning rate that modulates teaching signals rather than serving directly as a teaching signal - the title of the paper "*Mesolimbic dopamine adapts the rate of learning ...*" is intended to reflect that primary assertion. But the motivation for AND validation of this account depends crucially on differences between direct policy learning and value learning.

Motivation: the fact that signed RPEs are not required for policy learning (i.e. the gradient is relative to policy not relative to a value function) opens up the possibility that midbrain dopamine activity could be mapped onto a different component of the learning model and motivates our theoretical exploration. Adaptive rates in particular are especially critical to policy learning methods ⁹.

Validation: The learning rate function that we propose is in many cases hard to distinguish from a signed error teaching signal. For instance, stimulating DA on every trial of learning could speed or even unblock learning whether it was functioning as a teaching signal or as a learning rate. The closed-loop stimulation paradigm that we chose (Fig. 4) offers the possibility to distinguish these functions specifically within a framework like our policy learning model ACTR where behavioral performance is being related to policy updates at a trial wise level.

Thus adopting a novel policy learning model was instrumental in elaborating and validating the novel function we are proposing for mesolimbic dopamine in signaling an adaptive learning rate.

We are enthusiastic about the reviewer's point that value learning models can also make use of adaptive learning rates – it is intriguing to consider whether even in hybrid actor-critic type models dopamine's function might also be best described as an adaptive learning rate. While policy learning models offer superior fits to behavioral and neural

data in the context studied here, there may be tasks or phases of learning when a value learning / hybrid model is the best fit compared to a policy model. We hope the reviewer agrees that if a future dataset was better captured by a value learning model than a policy learning model and dopamine was best modeled as an adaptive learning rate in that context - that would be consistent with a primary conclusion of our work (we do not claim that nothing can ever be value learning) and an exciting discovery building upon our work in two ways:

1. Dopamine has for ~25 years been associated with the signed error term in value learning¹⁰; so a subsequent paper discovering it was an adaptive rate term in value learning would add to the impact of our study.

2. We show a novel biological implementation that is useful for computing an adaptive learning rate. In machine learning, adaptive rate computations are ubiquitous (¹¹ has been cited >100,000 times in 8 years according to Google Scholar) but a plausible biological implementation of an adaptive rate term has not been proposed prior to our work.

I recognize that the stimlick+ and stimlick- expt appears to be a critical test in their mind of learning rate versus error signal. I have read the text and figure legend multiple times and I remain unclear about their arguments. They are not stimulating on trials with negative vs positive performance errors, they are stimulating on all lick+ vs lick- trials (if I understand correctly), so I do not follow their arguments - it's not clear which trials have which performance errors. It seems like it would be much preferable to actually design an experiment that is intended to differentiate between a contribution of DA to learning rate versus to a signed prediction error much directly, if that is their goal. They could design a task with some trials with positive and others with negative prediction errors, and showing that increasing DA increases learning rate in both sets of trials.

We apologize that this point has remained unclear. The reviewer states "They are not stimulating on trials with negative vs positive performance errors" and "it's not clear which trials have which performance errors". The data very simply and directly clarify this point, but we have failed to highlight this. We have now updated the main text and added new analyses (shown here as well) to directly focus on this point: ***in mice (not just the ACTR model), lick- trials on average have positive PEs, and lick+ trials on average have negative PEs (Ext. Data Fig. 8f, shown at right).*** As would then be expected, when we performed experiments with the stimLick+/stimLick- contingencies (carefully tuning stimulation to be a matched subset of each trial type; Fig. 5b), we indeed successfully captured the appropriately signed performance errors (Ext. Data Fig 8g, shown at right). ***StimLick+ and StimLick- contingencies thus are a technique for stimulating at the time of reward delivery on trials that on average have negative or positive performance errors, respectively.***

In addition to this experiment design *in fact* selecting for distinct PE distributions (contrary to the reviewer's understanding but hopefully cleared up now), it is crucial to recognize that simulations in Fig. 5 show clearly that this paradigm of stimulating on Lick+ vs Lick- trials **makes unique and strongly distinguishing predictions between multiple models**. Indeed, we find this design particularly useful because it uniquely competes many different model predictions against each other.

Other designs, for example stimulating during putative negative and positive reward prediction errors, are less straightforward to distinguish from standard models and thus are less useful. For one, they require some inference about when signed prediction errors occur that could be debated, whereas detection/no detection of licking is unambiguous across labs trying to replicate our work. Second, stimulating on positive prediction error trials makes identical predictions for an RPE/value learning framework and our adaptive rate framework and a behavioral reinforcement framework; thus, one half of the manipulation experiment would be uninformative with respect to our main conclusions. We do think that an essential next step will be to design new tasks that are specifically tailored to even better distinguish performance errors and decorrelate them from positive and negative prediction errors - but we

also see a great deal of power in taking a canonical task for the value learning interpretation and showing a surprising result.

In addition to previous comparisons in Fig 5, we also now include for comparison the predictions of “on-policy” TD learning and the commonly held notion of direct behavioral reinforcement by DA (as invoked by the reviewer previously). In the first round of review, the reviewer wondered (very reasonably) whether direct reinforcement of behavior by dopamine stim might explain our effects. We pointed out that 1) the pause right before reward was not true of the data, 2) that still would not explain stimLick- where there was no licking to be reinforced, 3) the stim++Lick+ condition (supraphysiological stimLick+) under the reviewer’s proposal would lead to a stronger reduction in licking over training rather than the observed increase. Thus, the design in question provided several lines of evidence to rule out the reviewer’s specific alternative model. To help readers more broadly appreciate this important aspect of the design we also now include those rejected predictions in revised Figure 5.

... hybrid actor-critic models are probably the best to encompass all data, which has long been thought in the field... Moreover, their comment in the rebuttal that DA isn’t traditionally thought to contribute to the policy learning in traditional actor-critic models isn’t true. DA projections to dorsal striatum are thought to help train the “actor” in actor-critic models, where the actor is classically placed in dorsal striatum and the critic in ventral striatum.

We agree that in actor-critic models of the basal ganglia that “DA projections to the dorsal striatum are thought to help train the actor” and regret any language that would have contradicted that statement. Below is the relevant passage from our rebuttal:

When DA is proposed to participate in actor-critic learning in the basal ganglia, what is specifically being proposed is the same as for critic-only learning: that DA signals an RPE that updates a value function. Thus, it is not “mainstream” to argue that dopamine participates in direct evaluation of policies rather than indirect evaluation through construction of value functions. Recent work from Yael Niv and colleagues¹² for example has clearly summarized that prior work had considered only relatively limited roles for dopamine in actor learning and had focused extensively on a primary role of dopamine in determining learning of the critic (i.e. value learning) (see also a prescient work from Loewenstein and colleagues¹³)

The reviewer agrees that direct policy learning (and REINFORCE algorithm in particular) is distinct from both value learning and actor-critic learning. We think the reviewer is then implying a few related, but different follow ups:

- 1) Is the function of DA in the ACTR model distinct from its function in value/actor-critic learning models?
- 2) Is the ACTR model itself meaningfully different from existing actor-critic models in the literature?
- 3) Does the ACTR model offer greater explanatory power than value learning models?

We have extensively addressed Point (1). Even when the field has considered DA in the context of actor-critic learning, DA was used as an RPE that can update both the critic and the actor, very clearly illustrated in¹⁴. Using a dopaminergic RPE to update or direct the actor is not necessarily direct policy learning (no explicit calculation of the policy gradient; our point in prior rebuttal) and is fundamentally distinct from using a dopaminergic adaptive learning rate to modulate the error signals that directly update the actor.

Point (2): The reviewer argues that actor-critic is a distinct, superior option preferred by the field, but the vast majority of primary work considers pure value learning models and prominent reviews consistently argue that mesolimbic dopamine and NAc are the ‘critic’ (value) component. See for just a few high profile examples of this:^{2,3,10,15–22}. As one example within this set “A Unified Framework for Dopamine across Timescales, Kim et al, Cell 2022” a recent paper from Gershman & Uchida labs only considers variants of value learning models (not actor-critic). By comparison, full actor-critic models are most often discussed in reviews e.g.^{14,23,24} rather than in primary literature modeling experimental results. For instance, we are unaware of a study measuring dorsal striatal dopamine signals and explaining them as updates or action selection commands to an actor (in contrast, RPEs have been discussed in relation to SNc dopamine activity extensively^{24,25}).

Furthermore, actor-critic AND pure value models universally place the critic in ventral striatum and the mesolimbic dopamine projections there. Our work explicitly considers those regions and projections as specifying and learning

the policy, and thus is fundamentally distinct from proposals that critic signals learned via error signaling by the mesolimbic DA circuit are broadcast to the dorsal striatum.

Point (3) Our approach has the distinct advantage of providing a more plausible neural network-based mechanistic account that includes explicit modeling of behavioral responses and accounts for empirical learning data (two dissociable learning components) that are not accounted for in any existing actor-critic (or other) model formulations. Moreover, this more mechanistic model makes connections to circuitry (e.g. DA receives inputs in parallel with sensorimotor pathways²⁶, the source of action initiation correlates in DA²⁷; dopamine responses are the summed combination of a sensory and action components¹) that can inform future work and provide a new perspective on past work.

The direct, rigorous model comparisons suggested by the reviewer and now included further support our claims here. Further, as discussed above, the closed-loop stimulation results from Fig. 5 are not explained by models that have VTA dopamine functioning as an RPE in critic learning.

One idea is DA in dorsal striatum could train the actor by serving as a “baseline” term in a REINFORCE-like actor. We would appreciate to know the citation that the reviewer has in mind here as we are not aware of any clear articulation or testing of such a hypothesis. Perhaps the reviewer is referencing proposals that tonic dopamine activity might represent a baseline reward rate term? In the absence of references from the reviewer, we are aware of the use of a dopaminergic baseline term has been proposed to control response vigor²⁸ or exploitation/exploration trade-offs²⁹ or inattention³⁰ or more speculative (without explicit modeling) ideas about relations to many behavioral functions³¹. However, none of these models referenced REINFORCE or considered explicit direct policy learning models or anything close to the implementation level detail that we develop here in the ACTR model. We would be happy to discuss this idea at some point in the paper (though again this idea relates to dorsal striatal, not nucleus accumbens dopamine) if the reviewer clarifies which study they have in mind.

In addition, in re-reading the new manuscript, another big concern arose for me. They clarify “Performance errors used to train the model were proportional to the difference between the policy output at the time of reward delivery and the latency to collect water reward (see Methods).” This error signal doesn’t make much sense as a learning rule for a naive agent. For the agent to calculate the difference in time between reward delivery and reward collection latency, and use that as an error signal, they would need to know when the reward has been delivered based. Isn’t knowing when the reward is delivered the entire point of learning a trace conditioning task?

In the first round of review comments the reviewer wrote: “*Relatedly, the objective function that was used to train the network was based on time to collect reward once available, which can be considered a discounted value function.*” The reviewer appreciated in their first reading how we used the objective function and that it was, in their words, like a standard discounted reward term²⁸.

It is standard to give RL models veridical knowledge of reward delivery timing. In some contexts this might be unfair/unphysiological, however in this case it is well appreciated that delivery of a water droplet under the nose of an expectant head-fixed rodent is a highly salient sensory event that is perceptible with subsecond timing (for instance, see³²). Thus the perception of water is not something we train the mice on, rather mice know how to perceive the presence of water and they also know that perception of water is a distinct percept from consumption of water (water touching the tongue). In Figure 1 we provide quite direct behavioral evidence that the animal is aware of reward delivery since they modify a number of behaviors at short latency after reward delivery even in the first 10 trials of training. Part of the improvement in learning, as we show in Figure 1, is to more rapidly react to the detected presence of reward and shorten the latency to a host of behavioral responses (nose movement ‘nosing’, whisking, body movement). We postulate that collection latency is computed as the latency between these distinct perceptual events much like latencies between sensory events are known to be calculated³³.

The mice can also learn to use novel information about presence of water (e.g. solenoid click) for such latency estimates, but as we showed in previous rebuttal with a cohort of mice trained without the solenoid noise this is not essential.

Is the idea that they have already learned the sound of the solenoid click means that reward is available, but haven't learned how to lick in the presence of the click?

Yes, this is part of the idea and strongly supported by the experimental data reported in Figure 1. A naive animal must learn to control the licking behavior - in the context of the ACTR model, the policy network must learn that sensory input around water delivery should be coupled to the output of the policy network and drive a rapid onset of licking (the "reactive" component of the behavior described in Figure 1). Recent papers underscore the importance of this aspect of the learning problem in rodents^{34,35}.

seems more like a supervised learning rule than a reinforcement learning rule. In fact the paper they reference (49) is a supervised learning paper.

It is clear that Miconi used a supervised objective and thus dating back to the first manuscript version we explain how we modified ACTR to be distinct from Miconi's formulation. Using the model's own current policy output as a comparison is not a labeled, veridical target output (i.e. a supervised label; by contrast, Miconi for example told the model that the correct output unit activity at the end of trial type A == 1 and B == -1). Minimizing an objective like latency to collect reward is also not a supervised learning problem. As the reviewer has pointed out in other arguments, this learning objective could be considered as broadly similar to RPEs and value discounting which also use a (albeit quite distinct) self-generated estimate (value function) to compare to veridical current trial reward value provided to the model; but this is not considered a supervised learning algorithm.

Minor points:

In general, the paper is hard to read. E.g. what does it mean for stimulation to have an "input-output property"? Many of the sentences are not stand-alone and they require going between the text and figures many times to understand the intended meaning.

We have edited the paper and endeavored to make a number of the confusing points raised above more clear and easy to understand - in some cases without reference to the figures. The reviewer is referring to the following clause: "In separate experiments, calibrated and uncalibrated VTA-DA stimulations had similar input-output properties across the medial prefrontal cortex and the dorsal-to-ventral axis of the striatum." An input-output property is the relationship between the strength of stimulation and magnitude of measured output. We found that the spatial profile of output magnitude was similar for calibrated and uncalibrated stimulation. Hopefully, this terminology of spatial profile is more clear.

Similarly, many of the figure legends seem incomplete, making it hard to understand the figures. For example Fig 4h legend doesn't seem to describe the right most panels, only the "data" panels

Thank you, the oversight on Fig 4i has been corrected and we have gone back over the rest of the figure legends to confirm they are complete.

The goal of the GLM in F1 to predict behavior from other behavior measures was not clear.

The goal was stated in the text: "*To assess whether changes in both preparatory and reactive components were correlated with improvements in reward collection performance, we built generalized linear models (GLM).*" Thanks to the reviewer's feedback, we now further re-confirm this goal of formally assessing the relationship between behavioral measures and the performance metric at the end of the relevant paragraph, saying: "*Thus each behavioral measure reflected at least partially distinct aspects of improving policies, and the significantly different time courses of preparatory and reactive learning (Fig. 1g) further confirm that these two learning components are dissociable processes.*"

To restate and elaborate: we are asking whether both reactive and preparatory components of behavior appeared to contribute to the overall learning (reduction in collection latency). One way to provide good statistical evidence for the dependence of one variable (latency) on another set of independent variables (reactive, preparatory) is to construct a GLM and ask whether the regression weights on the independent variables are non-zero and whether removing one or more independent variables reduces the explained variance (r-squared). We use the GLM in Figure 1 to formally evaluate these statistical dependencies.

I think they may mean "temporal difference" model, not "temporal discount" model.

Thank you for pointing out this typo. The only such reference we could find was in the legend for former Fig 3c, and it has been corrected.

Figure 4b- please include time axis units and clarify how many mice/sessions etc are being averaged

Thank you for the clarification. The n values that were listed next to the traces (n=5 and n=6) were referring to mice and the legend described the data as coming from “Mean NAc-DA reward responses across training”. We have now further clarified “Mean NAc-DA reward responses across training (trials 1-800) for each mouse” and added a time scale bar (“1 s” or 1 second) in the figure. Thus the traces visualize the average DA reward transient seen by each mouse on unstimulated (black) vs stimulated (colored) trials.

On the question of model comparison

We performed the requested model comparisons but would like to articulate two further general points about model comparison. As Maneesh Sahani (among others) has pointed out - model comparison has a clear limitation which is that it can only discover the best model out of those considered and the models may be brittle⁵ especially when one uses abstract models (like low parameter RL models) to compare to variable physiological or behavioral data. In his colorful example, it can allow us to determine if a banana is better modeled as an apple or an orange. For this reason, we do believe it is important to also articulate (1) novel, (2) generative, and (3) implementational models with additional explanatory power like the ACTR model described here. We also find it to be comforting and supportive of our general endeavor in this study that policy learning models are dramatically more stable and robust against parameter variation than value learning models over the bounded parameter range used for optimization (Ext. Fig. 1).

However, neural network based generative models are not always amenable to formal model comparison techniques especially when the dimensionality is large⁷. Indeed, as Breiman began to appreciate in the context of large scale genetic data and more recently in machine learning with algorithmic models and the huge dimensionality of parameters - performance is often considered rather than explicit model comparison. This is in part because the parameter space can be massive in generative network models and one cannot always guarantee the true global optimum parameterization has been discovered because the required simulations would be too extensive. For these reasons we focus on examining performance of ACTR (akin to much work in the machine learning literature) where appropriate and model comparison for low-parameter model equivalents. We now have endeavored to provide the best of both worlds thanks to the reviewers' feedback so we believe we have satisfied any potential concerns about model comparison.

Some important clarifications from Round 1:

We appreciate that the reviewer has raised additional questions in this round including new questions that were not raised in the first round. We believe we have addressed these new concerns above. We would also like to summarize a number of key contributions of the manuscript that have been clarified without further inquiry:

1. A new, principled, and detailed description of learning during the acquisition of trace conditioning that we connect to concepts of policy optimization in machine learning (Fig 1). There have been calls from prominent theoreticians to explore this space; ours is the first empirical+modeling study to do so and includes multiple points of unique explanatory power as well as confirmatory causal experiments.
2. Our careful examination of individual differences in learning reveal, for the first time, that the response of mesolimbic dopamine (mDA) to the first few naive rewards predicts the final learned state ~1000 trials in the future (Fig 2). The sign of this relationship is opposite to predictions of the dominant interpretations in the field while consistent with our model.
3. We demonstrate that the functional effects of mDA stimulation on learning depend upon the intensity and duration of stimulation. This potentially re-contextualizes many results in the field (Fig 3).
4. We articulate a more detailed and plausible implementation level model of reinforcement learning ('ACTR' model) and demonstrate that this model can account for well known, unexplained discrepancies between mDA responses and prevailing models (Fig 3-4).
5. we develop and test a novel closed loop stimulation experiment that is uniquely consistent with our model, but inconsistent with 3+ existing hypotheses about mDA function in learning (Fig 4)

Below we summarize several specific points that were clarified (no further questions asked):

“The network seemed to be trained to calculate value, and then the derivative of that signal was used to produce the dopamine signal (RPE) - which all in all seem exceedingly similar to traditional actor-critic models.”

We endeavored to clarify that we are claiming dopamine produces a signal critical for adaptive modulation of learning rate not a signed prediction error term. The reviewer now appreciates this distinction between policy and value learning and agrees that our main conclusion is that the function of DA reward signals, whatever quantities they may encode, are to set an adaptive rate term rather than a signed error term (the reviewer correctly notes this is our main conclusion reflecting why we used it as the title of our paper).

“I did think it was interesting that dopamine early on was so predictive of individual differences in behavior, and that their model recapitulated it. But I couldn’t find an explanation of why their model produced that effect. Also, could they show the learning trajectory of the two groups of mice?”

We clarified how the model explains this effect and this remains a crucial point because the sign of the effect on individual differences is **opposite** to that predicted from models that propose dopamine is a signed error term (generally in value learning models previously), and this is thus very useful for distinguishing between competing models.

My understanding of this explanation is that despite calling these trials lick+, they are actually licking less immediately before the reward. Can they show this directly in both model and mice? ... if the behavior immediately before the reward is “not licking” that will strengthen the policy in a policy model and reinforce the action in a value learning model (since most recent behavior is most eligible for modification given eligibility trace).

We disconfirmed the reviewer’s hypothesis about the behavior here (DA stim is not selecting for a pause in licking) and we further show that a value-based model does **not** make the same prediction and fails to account for data (revised Fig. 4). Moreover, standard actor-critic implementations use a signed error term and thus also fail to account for these effects (our experimental data go in the opposite direction of “reinforcement” as the reviewer proposes). As described below we now update the figure with this additional prediction from an actor-critic framework.

Is that because they are mostly learning about the solenoid click?

We presented a new experimental dataset without solenoid clicks and showed very comparable behavioral learning ruling out “mostly learning” about the solenoid click.

It would be helpful to provide a clear definition of performance prediction errors. It was extremely difficult to read & follow the Methods section on their model. I suggest the authors should separate an explanation of the equations and concepts of the model versus details of the implementation, simulations etc.

We extensively updated the methods section providing an explicit break out of all equations and terms as requested. No further clarification was requested.

A particularly important issue is lack of clarity on how the DA signal is calculated, as the dopamine “beta” signal doesn’t seem to have an equation in methods

This was much more explicitly and intuitively described. No further clarification was requested.

I don’t think it makes the effect of DA on producing a cue response at all uninteresting or unconvincing. They are calibrating to the DA signal immediately adjacent to the optical fiber in NAc. This should be conservative since not all dopamine neurons are in the immediate vicinity of the NAc fiber.

We added a new experimental dataset allowing a reader to explicitly evaluate the extent to which calibration was just for dopamine neurons in NAc. We also reiterate that even if conclusions were restricted to the mesolimbic DA pathway as the reviewer seems to be suggesting, this pathway is the most closely associated with value learning (or the critic in actor-critic) in the literature and thus of greatest interest to the arguments of the manuscript and those raised by the reviewer.

Also, does strong stimulation of DA in their model lead to a greater cue response?

We added new modeling and analysis that showed strong stimulation leads to greater cue responses both in experimental data and the ACTR model as requested (revised Figure 4g-h). No further clarification was requested.

1. Coddington, L. T. & Dudman, J. T. The timing of action determines reward prediction signals in identified midbrain dopamine neurons. *Nat. Neurosci.* **21**, 1563–1573 (2018).
2. Suri, R. E. & Schultz, W. A neural network model with dopamine-like reinforcement signal that learns a spatial delayed response task. *Neuroscience* **91**, 871–890 (1999).
3. Amo, R., Yamanaka, A., Tanaka, K. F., Uchida, N. & Watabe-Uchida, M. A gradual backward shift of dopamine responses during associative learning. *bioRxiv* 2020.10.04.325324 (2020) doi:10.1101/2020.10.04.325324.
4. Li, J. & Daw, N. D. Signals in Human Striatum Are Appropriate for Policy Update Rather than Value Prediction. *Journal of Neuroscience* vol. 31 5504–5511 (2011).
5. Chandrasekaran, C. *et al.* Brittleness in model selection analysis of single neuron firing rates. *bioRxiv* 430710 (2018) doi:10.1101/430710.
6. Ecoffet, A., Huizinga, J., Lehman, J., Stanley, K. O. & Clune, J. First return, then explore. *Nature* **590**, 580–586 (2021).
7. Breiman, L. Statistical Modeling: The Two Cultures (with comments and a rejoinder by the author). *Stat. Sci.* **16**, 199–231 (2001).
8. Williams, R. J. Simple statistical gradient-following algorithms for connectionist reinforcement learning. *Mach. Learn.* **8**, 229–256 (1992).
9. Schulman, J., Wolski, F., Dhariwal, P., Radford, A. & Klimov, O. Proximal Policy Optimization Algorithms. *arXiv [cs.LG]* (2017).
10. Schultz, W., Dayan, P. & Montague, P. R. A neural substrate of prediction and reward. *Science* **275**, 1593–1599 (1997).
11. Kingma, D. P. & Ba, J. Adam: A Method for Stochastic Optimization. *arXiv [cs.LG]* (2014).
12. Bennett, D., Niv, Y. & Langdon, A. J. Value-free reinforcement learning: policy optimization as a minimal model of operant behavior. *Current Opinion in Behavioral Sciences* **41**, 114–121 (2021).
13. Mongillo, G., Shteingart, H. & Loewenstein, Y. The Misbehavior of Reinforcement Learning. *Proc. IEEE* **102**, 528–541 (2014).
14. Niv, Y. Reinforcement learning in the brain. *J. Math. Psychol.* **53**, 139–154 (2009).
15. Eshel, N. *et al.* Arithmetic and local circuitry underlying dopamine prediction errors. *Nature* vol. 525 243–246 (2015).
16. Kim, H. R. *et al.* A Unified Framework for Dopamine Signals across Timescales. *Cell* **183**, 1600–1616.e25 (2020).
17. Hamid, A. A., Frank, M. J. & Moore, C. I. Wave-like dopamine dynamics as a mechanism for spatiotemporal

- credit assignment. *Cell* **184**, 2733–2749.e16 (2021).
18. Waelti, P., Dickinson, A. & Schultz, W. Dopamine responses comply with basic assumptions of formal learning theory. *Nature* **412**, 43–48 (2001).
 19. Soares, S., Atallah, B. V. & Paton, J. J. Midbrain dopamine neurons control judgment of time. *Science* **354**, 1273–1277 (2016).
 20. Mikhael, J. G., Kim, H. R., Uchida, N. & Gershman, S. J. The role of state uncertainty in the dynamics of dopamine. *Curr. Biol.* **32**, 1077–1087.e9 (2022).
 21. Bari, B. A. *et al.* Stable Representations of Decision Variables for Flexible Behavior. *Neuron* **103**, 922–933.e7 (2019).
 22. Rutledge, R. B. *et al.* Dopaminergic drugs modulate learning rates and perseveration in Parkinson's patients in a dynamic foraging task. *J. Neurosci.* **29**, 15104–15114 (2009).
 23. Chen, R. & Goldberg, J. H. Actor-critic reinforcement learning in the songbird. *Curr. Opin. Neurobiol.* **65**, 1–9 (2020).
 24. Schultz, W. Neuronal Reward and Decision Signals: From Theories to Data. *Physiol. Rev.* **95**, 853–951 (2015).
 25. Hikosaka, O., Kim, H. F., Yasuda, M. & Yamamoto, S. Basal Ganglia Circuits for Reward Value–Guided Behavior. (2014) doi:10.1146/annurev-neuro-071013-013924.
 26. Pan, W.-X., Coddington, L. T. & Dudman, J. T. Dissociable contributions of phasic dopamine activity to reward and prediction. *Cell Rep.* **36**, 109684 (2021).
 27. Coddington, L. T. & Dudman, J. T. Learning from Action: Reconsidering Movement Signaling in Midbrain Dopamine Neuron Activity. *Neuron* **104**, 63–77 (2019).
 28. Niv, Y., Daw, N. D., Joel, D. & Dayan, P. Tonic dopamine: opportunity costs and the control of response vigor. *Psychopharmacology* **191**, 507–520 (2007).
 29. Beeler, J. A., Daw, N., Frazier, C. R. M. & Zhuang, X. Tonic dopamine modulates exploitation of reward learning. *Front. Behav. Neurosci.* **4**, 170 (2010).
 30. Mikhael, J. G., Lai, L. & Gershman, S. J. Rational inattention and tonic dopamine. *PLoS Comput. Biol.* **17**, e1008659 (2021).
 31. Wang, Y., Toyoshima, O., Kunimatsu, J., Yamada, H. & Matsumoto, M. Tonic firing mode of midbrain dopamine neurons continuously tracks reward values changing moment-by-moment. *Elife* **10**, (2021).
 32. Galifianes, G. L., Bonardi, C. & Huber, D. Directional Reaching for Water as a Cortex-Dependent Behavioral Framework for Mice. *Cell Reports* vol. 22 2767–2783 (2018).
 33. Paton, J. J. & Buonomano, D. V. The Neural Basis of Timing: Distributed Mechanisms for Diverse Functions.

Neuron **98**, 687–705 (2018).

34. Bollu, T. *et al.* Cortex-dependent corrections as the tongue reaches for and misses targets. *Nature* **594**, 82–87 (2021).
35. Xu, D. *et al.* Cortical processing of flexible and context-dependent sensorimotor sequences. *Nature* **603**, 464–469 (2022).

Reviewer Reports on the Second Revision:

Referees' comments:

Referee #3 (Remarks to the Author):

I thank the reviewers for the revised comments. I think the manuscript is much clearer than before in terms of understanding what the main points and conclusions are. I think they are bringing up an interesting hypothesis, and have interesting results. While I still find several components unintuitive (from the very concept of using a Pavlovian conditioning task to distinguish value learning from policy learning, to the points about the learning rule that I brought up before, and finally the experimental predictions in Figure 5 remain unintuitive to me), I think this data and modeling work is extensive and the message is interesting, and it is time to publish this work & share the results broadly with the field.